# Scaling-Aware Adapter for Structure-Grounded LLM Reasoning

**Zihao Jing** [1]  **Qiuhao Zeng** [1]  **Ruiyi Fang** [1]  **Yan Yi Li** [2]  **Yan Sun** [1]  **Boyu Wang** [1]  **Pingzhao Hu** [1 2]

## Abstract

Large language models (LLMs) are enabling reasoning over 2D and 3D structures, yet existing methods remain modality-specific and typically compress structural inputs through sequence-based tokenization or fixed-length query connectors. Such architectures either omit the geometric grounding requisite for mitigating structural hallucinations, or impose inflexible modality fusion bottlenecks that concurrently over-compress and suboptimally allocate structural tokens, thereby impeding the realization of generalized all-atom reasoning. We introduce **Cuttlefish**, a unified multimodal LLM that grounds language reasoning in geometric cues while scaling modality tokens with structural complexity. First, **Scaling-Aware Patching** leverages an instruction-conditioned gating mechanism to generate variable-size patches over structural graphs, adaptively scaling the query token budget with structural complexity to mitigate fixed-length connector bottlenecks. Second, **Geometry Grounding Adapter** refines these adaptive tokens via cross-attention to modality embeddings and injects the resulting modality tokens into the LLM, exposing explicit geometric cues to reduce structural hallucination. Experiments across interdisciplinary all-atom benchmarks demonstrate that Cuttlefish achieves superior performance in heterogeneous structure-grounded reasoning. Code: github.com/zihao-jing/Cuttlefish.

## 1. Introduction

**Background.** Multimodal large language models (LLMs) increasingly extend to reason over non-linguistic modalities, including vision, video, and scientific structures. Beyond vision-language models (e.g., Flamingo (Alayrac et al.,

*Table 1.* Functional group hallucination test on 200 molecules and proteins from dataset GEO-AT. Metrics HR: hallucination rate; HPM: hallucinations per molecule; and AR: answer rate. (S) denotes the model's sequence-only variant (with tokenizer enhanced). The preferred variant is highlighted in pink for clarity.

| | MOLECULE | | | PROTEIN | |
|---|---|---|---|---|---|
| MODEL | HR (HPM) ↓ | AR ↑ | MODEL | HR (HPM) ↓ | AR ↑ |
| MOL-LLAMA(S) | 0.28 (0.91) | 0.95 | PROTCHATGPT(S) | 0.34 (0.99) | 0.99 |
| MOL-LLAMA | 0.12 (0.59) | 0.97 | PROTCHATGPT | 0.10 (0.55) | 0.99 |
| 3D-MOLM(S) | 0.59 (2.23) | 0.89 | PROT2CHAT(S) | 0.29 (2.30) | 1.00 |
| 3D-MOLM | 0.23 (1.15) | 0.83 | PROT2CHAT | 0.06 (0.23) | 0.99 |

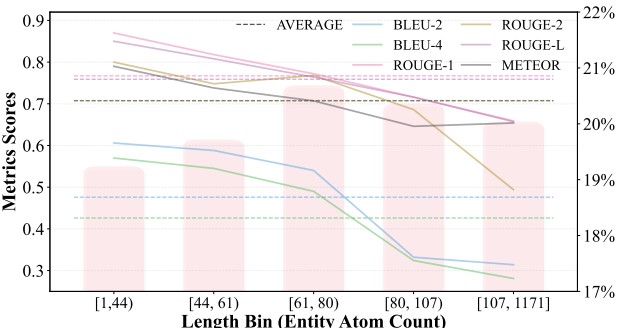

*Figure 1.* Mol-Llama performance on the Mol-Instructions captioning task, evaluated across five molecule length bins with 6 metrics (left y-axis, detailed in App D.3) plotted as curves with dashed overall averages, and the background bars indicate the proportion of samples in each length bin (right y-axis).

2022) and LLaVA (Lin et al., 2024)), recent efforts also target scientific domains, such as Galactica (Taylor et al., 2022) for scientific text, BioGPT (Luo et al., 2022) for biomedical text, and structure-aware models that operate on microscopic geometry, including Mol-Llama (Kim et al., 2025) for molecules, ProtLLM (Zhuo et al., 2024) for proteins, and ChatNT (de Almeida et al., 2025) for nucleic acids.

**All-atom modeling** has advanced rapidly, catalyzed by AlphaFold-3 (Abramson et al., 2024). Subsequent work, including BoltzGen (Stark et al., 2025) for coordinates generation and ATOMICA (Fang et al., 2025a) for representation, further suggests that deep models can operate directly on all-atom level structures. However, most structure-aware LLM efforts remain single modality (Wang et al., 2025b; Park et al., 2024) without a unified interface for heterogeneous all-atom entities. ChatNT (de Almeida et al., 2025) takes a step toward integrating nucleic acids and proteins, but is constrained to sequence inputs without geometric

[1]Department of Computer Science, Western University, London, Canada [2]Department of Biochemistry, Western University, London, Canada. Correspondence to: Pingzhao Hu <phu49@uwo.ca>.

*Proceedings of the $43^{rd}$ International Conference on Machine Learning*, Seoul, South Korea. PMLR 306, 2026. Copyright 2026 by the author(s).

evidence.

This gap motivates a rethinking of how all-atom structural evidence can be selected and exposed to the language model:

**Challenge 1: Budget Scaling in All-Atom Modalities.** All-atom graphs span a wide size range. As shown in Fig. 1, Q-Former-style (Li et al., 2023) fixed-length query connectors (Mol-Llama (Kim et al., 2025)) yield lower performance across all metrics for molecules in larger length bins, due to over-compressed geometric features. Conversely, increasing the budget over-allocates capacity to small entities, diluting attention. **Our solution: Scaling-Aware Patching** uses instruction-conditioned gating to select anchors and expand variable-size structural patches, letting the query token budget grow with structural complexity to avoid over-compression and context inefficiency.

**Challenge 2: Structural Hallucination.** Sequence-only inputs, such as SMILES, protein/DNA residue sequences, do not encode geometry or long-range spatial relations. As shown in Tab. 1, LLMs that lack verifiable structural evidence can produce geometric rationales that are not entailed by the conditioning context. **Our solution: Geometry Grounding Adapter** refines the adaptive tokens through cross-attention with modality embeddings, injecting the final modality tokens into the LLM to provide the explicit geometric grounding to suppress structural hallucinations.

Together, these designs address both the correctness and scalability of structure-aware reasoning: Scaling-Aware Patching adaptively allocates representational capacity to structurally informative regions, while the Geometry Grounding Adapter ensures that LLM outputs are explicitly conditioned on geometry-aware tokens. Our contributions are summarized as follows:

- We propose **Scaling-Aware Patching**, an instruction-conditioned anchor patch growing mechanism ensuring modality token count scales with structural complexity, mitigating fixed-length connector bottlenecks.
- We introduce **Geometry Grounding Adapter**, which injects verifiable geometric cues into the LLM via modality tokens, reducing structural hallucination.
- We present **Cuttlefish**, a unified structure-aware LLM that reasons over all-atom modalities, and achieves strong performance across all-atom understanding benchmarks.

## 2. Related Work

Molecular LLMs have progressed from contrastive representation learning toward generation and instruction following. MoleculeSTM (Liu et al., 2023b) and MoMu (Su et al., 2022) establish graph or structure and text alignment via contrastive co-embedding, while InstructMol (Cao et al., 2025) and GIT-Mol (Liu et al., 2023a) introduce projector

and adapter-based fusion with instruction tuning. Connector designs vary across Q-Former-style (Li et al., 2023) query tokens, as in 3D-MoLM (Li et al., 2024) and Uni-MoT (Guo et al., 2025b), and tokenization oriented bridges such as Graph2Token (Wang et al., 2025a), contrasting with sequence-only baselines like MolT5 (Edwards et al., 2022). Protein LLMs follow a parallel trajectory, from alignment in ProtST (Xu et al., 2023b) to open-ended generation in ProtLLM (Zhuo et al., 2024), with Prot2Text-V2 (Fei et al., 2025) and ProteinGPT (Xiao et al., 2024a) emphasizing captioning and dialogue under instruction tuning. For nucleic acids, ChatNT (de Almeida et al., 2025) unifies DNA, RNA, and protein through encoder coupling, while RNA-GPT (Xiao et al., 2024b) targets instruction-aligned RNA sequence understanding.

Despite strong empirical progress, existing systems exhibit recurring geometry bottlenecks. Sequence-only or shallow fusion, including MolT5 (Edwards et al., 2022), ProtST (Xu et al., 2023b), and RNA-GPT (Xiao et al., 2024b), often under-specifies geometry, which correlates with structural hallucinations in dialogue settings such as ProtChat-GPT (Wang et al., 2024). Embedding-only alignment, as in MoleculeSTM (Liu et al., 2023b), limits expressivity for multi-step reasoning and generation. Simple projection fusion and fixed-length query Q-Former-style (Li et al., 2023) connectors, as in InstructMol (Cao et al., 2025), GIT-Mol (Liu et al., 2023a), 3D-MoLM (Li et al., 2024), Uni-MoT (Guo et al., 2025b), and Chem3DLLM (Jiang et al., 2025), compress variable-size structures into a constant token budget, which degrades scalability with entity size and atom-level detail. Discretization-based bridges such as Graph2Token (Wang et al., 2025a) reduce this mismatch but still face information loss at high structural complexity. Several recent variants, including MolCA (Liu et al., 2023c), LLaMo (Park et al., 2024), DeepMolTex (Yan et al., 2025), and Prot2Chat (Wang et al., 2025b), remain within these fixed-capacity connector regimes.

All-atom LLMs must jointly address both structural representation and scaling under variable-length atomistic inputs. This setting demands long-range spatial relations modeling while avoiding information loss from fixed-capacity connectors. Progress likely requires a universal atom-level structural adapter with scaling-aware query length that preserves fine geometry for faithful reasoning. Precise definitions of scaling and modality scope are provided in App. A.3 with detailed related works in App. B.

## 3. Structure-Grounding All-Atom LLM: Cuttlefish

We introduce Cuttlefish, a structure-aware all-atom LLM. Mirroring the multi-appendaged versatility of its namesake, Cuttlefish leverages a unified connector that extends to het-

(a) Cuttlefish

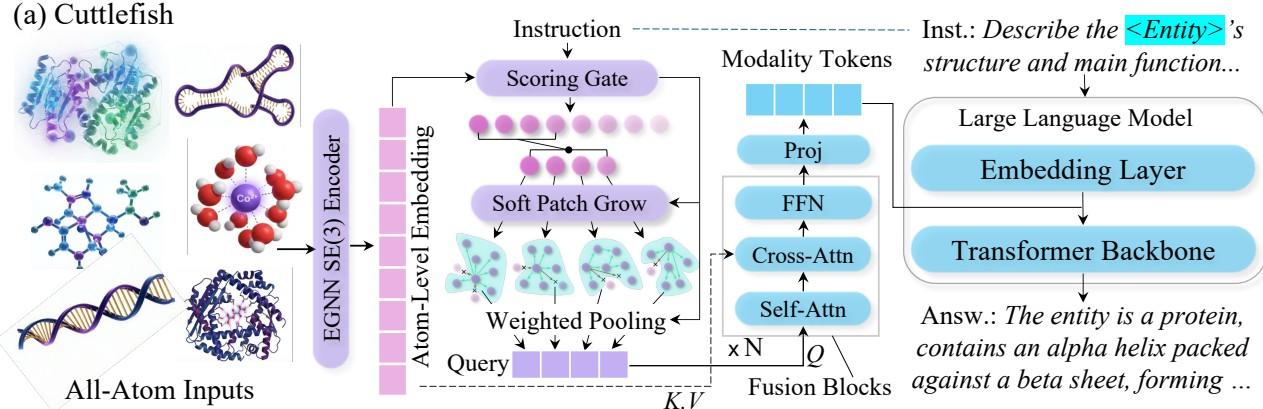

*Figure 2.* Architecture of Cuttlefish. The framework accepts all-atom inputs (spatial graph: atom features, coordinates, and spatial relations) processed by EGNN for modality embeddings. The model incorporates Scaling-Aware Patching through an instruction-conditioned gate and soft patch-growing mechanism. Then Geometry Grounding Adapter utilizes cross-attention to enrich adaptive tokens with granular geometric features derived from modality embeddings, subsequently projecting these modality tokens into the LLM's embedding space.

erogeneous all-atom modalities. Sec. 3.1 introduces the model architecture and the overall data flow, followed by a description of Scaling-Aware Patching in Sec. 3.2 and Geometry Grounding Adapter in Sec. 3.3. Finally, Sec. 3.4 outlines the training objectives and optimization protocol.

### 3.1. Architecture Overview

Cuttlefish is a multimodal LLM designed for structure-grounded reasoning by adaptively bridging microscopic geometry with linguistic context. As shown in Fig. 2, (1) input atomic coordinates and features are first encoded via an SE(3)-equivariant Equivariant Graph Neural Network (EGNN) to produce atom-level modality embeddings. (2) Then, an instruction-conditioned scoring gate identifies critical anchor atoms by selecting top-ranking nodes that satisfy a cumulative mass-based threshold. (3) A soft patch-growing operator determines atomic membership using the spatial proximity and gate logits, enabling in-patch weighted pooling to generate structural query tokens. (4) These queries retrieve geometric details through cross-attention to produce modality tokens, which are finally injected into the LLM embedding space at the designated locations.

### 3.2. Scaling-Aware Patching

The Scaling-Aware Patching compresses all-atom geometry into an adaptive number of tokens by identifying structurally and semantically significant regions. This mechanism addresses the fixed-budget bottleneck by scaling representational capacity relative to the scale of different entities.

**Mass-Based Anchor Selection.** Given instruction embeddings $z$, graph-batch assignments $b$, and SE(3)-equivariant node embeddings $X$, we compute per-node anchor logits $\ell = G_{\text{anc}}(z, X, b)$, which represent **instruction-conditioned relevance**: how much each atom contributes

to answering the current prompt (Algo. 1, Line 1). For each graph $g$, we then choose the number of anchors $k_g$ adaptively. Ranking nodes by $\boldsymbol{prob} = \text{Softmax}(\boldsymbol{\ell})$—where softmax serves purely as a normalization to enable the mass-based threshold, distinct from any GNN-internal operations—we select the top indices $\mathcal{S}_g$ such that their cumulative probability mass exceeds $\rho$:

$$k_g = \min\left\{ k \in \{1, \ldots, N_g\} \;\middle|\; \sum_{j=1}^{k} \boldsymbol{prob}_{\pi_j} \geq \rho \right\}. \quad (1)$$

where $\pi$ denotes the descending permutation of $\boldsymbol{prob}$ (Lines 7–8). This cumulative selection ensures further query tokens are adaptively allocated to match the entity scale.

**Soft Patch Growth.** The identified anchors $\mathcal{S}_g$ are expanded into structural patches via a soft assignment mechanism. We determine the membership $\boldsymbol{W}_{i,a}$ of atom $i$ to anchor $a$ by fusing spatial proximity with the semantic bias of the anchor (Line 12). The assignment weight is formulated as:

$$\boldsymbol{W}_{i,a} = \frac{\exp(-\|\mathbf{P}_i - \mathbf{P}_a\|_2^2 + \ell_a)}{\sum_{a' \in \mathcal{S}_g} \exp(-\|\mathbf{P}_i - \mathbf{P}_{a'}\|_2^2 + \ell_{a'})} \quad (2)$$

It ensures that anchors with high instruction relevance (large $\ell_a$) can "grow" to capture a wider receptive field, while spatial distances $\|\mathbf{P}_i - \mathbf{P}_a\|_2^2$ ($\mathbf{P}$ is the atom coordinates) maintain geometric locality.

**In-Patch Weighted Pooling.** Final patch tokens $\boldsymbol{t}_a$ are synthesized through membership-weighted aggregation of node features within each patch:

$$\boldsymbol{t}_a = \sum_{i \in \mathcal{I}_g} \frac{\boldsymbol{W}_{i,a}}{\sum_{j \in \mathcal{I}_g} \boldsymbol{W}_{j,a}} \boldsymbol{X}_i \quad (3)$$

The resulting tokens $\boldsymbol{t}_a$ effectively summarize the structural grounding in both geometry and textual intent, providing a

**Algorithm 1** Scaling-Aware Patching

**Input:** Instruction embeddings $\boldsymbol{z} \in \mathbb{R}^{G \times D_{LLM}}$, node embeddings $\boldsymbol{X} \in \mathbb{R}^{N \times D_{enc}}$, node coordinates $\mathbf{P} \in \mathbb{R}^{N \times 3}$, batch assignment $\boldsymbol{b} \in \{0, \dots, G-1\}^N$, anchor gate $g_{\text{anc}}(\cdot)$, maximum anchors $K_{\max}$, mass threshold $\rho$.

**Output:** Patch tokens $\boldsymbol{T} \in \mathbb{R}^{G \times K \times D_{enc}}$, patch mask $\boldsymbol{M} \in \{0,1\}^{G \times K}$, anchor indices $\boldsymbol{A} \in \{1, \dots, N\}^{G \times K}$.

/* Step1: Instruction-Conditioned Anchor Scoring */
1: $\boldsymbol{\ell} \leftarrow G_{\text{anc}}(\boldsymbol{z}, \boldsymbol{X}, \boldsymbol{b})$       $\triangleright \boldsymbol{\ell} \in \mathbb{R}^N$ *anchor logits*
2: Initialize $\boldsymbol{T} \leftarrow \boldsymbol{0}$, $\boldsymbol{M} \leftarrow \boldsymbol{0}$, $\boldsymbol{A} \leftarrow -1$
/* Step2: Mass-Based Anchor Selection */
3: **for** $g = 0$ **to** $G - 1$ **do**
4:     $\mathcal{I}_g \leftarrow \{i \mid \boldsymbol{b}_i = g\}$    $\triangleright$ *node indices in graph $g$*
5:     $\mathbf{p} \leftarrow Softmax(\boldsymbol{\ell}_{\mathcal{I}_g})$    $\triangleright$ *softmax for anchor scoring*
6:     $\pi \leftarrow Argsort(\mathbf{p}, descending)$    $\triangleright$ *sort by score*
7:     $k_g \leftarrow \min \left( K_{\max}, \min\{k : \sum_{j=1}^{k} \mathbf{p}_{\pi_j} \geq \rho\} \right)$
8:     $\mathcal{S}_g \leftarrow \{\mathcal{I}_g[\pi_1], \dots, \mathcal{I}_g[\pi_{k_g}]\}$    $\triangleright$ *selected anchors*
/* Step3: Soft Assignment Patch Growth */
9:     $\boldsymbol{D}_{i,a} \leftarrow \|\mathbf{P}_i - \mathbf{P}_a\|_2^2$   $\forall i \in \mathcal{I}_g,\ a \in \mathcal{S}_g$ $\triangleright$ *distance*
10:    $\boldsymbol{S}_{i,a} \leftarrow -\boldsymbol{D}_{i,a} + \boldsymbol{\ell}_a$   $\forall i \in \mathcal{I}_g,\ a \in \mathcal{S}_g$   $\triangleright$ *add bias*
11:    $\boldsymbol{W} \leftarrow Softmax_a(\boldsymbol{S})$    $\triangleright$ *to assignment weight*
/* Step4: Patch Token Pooling */
12:    $\boldsymbol{T}_a \leftarrow Norm_i(\boldsymbol{W})^\top \boldsymbol{X}_{\mathcal{I}_g}$   $\forall a \in \mathcal{S}_g$    $\triangleright$ *pooling*
13:    $\boldsymbol{T}[g, 1 : k_g] \leftarrow \{\boldsymbol{T}_a\}_{a \in \mathcal{S}_g}$, $\boldsymbol{A}[g, 1 : k_g] \leftarrow \mathcal{S}_g$
14:    $\boldsymbol{M}[g, 1 : k_g] \leftarrow 1$    $\triangleright$ *write back pooling results*
15: **end for**
16: **return** $\boldsymbol{T}, \boldsymbol{M}, \boldsymbol{A}$

---

solid query initialization in further retrieval (Sec. 3.3). Theoretical analysis of the instruction-weighted compression distortion bound is provided in App. A.1.

### 3.3. Geometry Grounding Adapter

The Geometry Grounding Adapter facilitates the explicit injection of geometric evidence into the LLM by transforming query tokens from patch pooling into modality tokens.

**Geometry Grounding Retrieval.** We first project the patch tokens $\boldsymbol{t}$ from Sec. 3.2 into patch queries $\mathcal{Q}$ (Algo. 2, Line 2). To capture detailed structural dependencies, these queries undergo a series of fusion blocks $\{f_\ell\}_{\ell=1}^{L_f}$ where they execute cross-attention with the full modality embeddings $\boldsymbol{X}$ (Lines 4–8). This enables queries to retrieve high-resolution geometric cues that are abstracted during patching. This retrieval-and-refinement stage complements anchor selection: while the anchor gate identifies instruction-relevant substructure positions, the fusion blocks recover full geometric detail at and around those positions rather than performing a second selection.

**Space Alignment and Injection.** The retrieved features are projected into the LLM's embedding space $D_{\text{LLM}}$ as final

---

**Algorithm 2** Geometry Grounding Adapter

**Input:** Instruction embeddings $\boldsymbol{E} \in \mathbb{R}^{B \times L \times D_{LLM}}$, modality (node) embeddings $\boldsymbol{X} \in \mathbb{R}^{N \times D_{enc}}$, patch tokens $\boldsymbol{T}$ and patch mask $\boldsymbol{M}$ (Alg. 1), batch assignment $\boldsymbol{b}$, fusion blocks $\{f_\ell\}_{\ell=1}^{L_f}$, modality placeholder token id $y_{\text{ins}}$.

**Output:** Updated LLM embeddings $\boldsymbol{E}'$ (and aligned attention/label masks).

/* Step1: Locate Injection Positions */
1: $\boldsymbol{p} \leftarrow FindPos(\mathbf{y} = y_{\text{ins}})$
/* Step2: Geometry Grounding Retrieval */
2: $\mathcal{Q} \leftarrow Proj_p(Norm(\boldsymbol{T}))$      $\triangleright$ *get patch queries*
3: $\mathcal{K}, \mathcal{V} \leftarrow Proj_n(Norm(\boldsymbol{X}))$    $\triangleright$ *modality embeddings*
4: **for** $\ell = 1$ **to** $L_f$ **do**
5:    $\mathcal{Q} \leftarrow Self\text{-}Attn(\mathcal{Q})$     $\triangleright$ *patch tokens themselves*
6:    $\mathcal{Q} \leftarrow Cross\text{-}Attn(\mathcal{Q}, \mathcal{K}, \mathcal{V})$   $\triangleright$ *retrieve full nodes*
7:    $\mathcal{Q} \leftarrow FFN(\mathcal{Q})$
8: **end for**
9: $\widehat{\boldsymbol{T}} \leftarrow Norm_{\text{out}}(Proj_{\text{out}}(\mathcal{Q})) \in \mathbb{R}^{G \times K \times D_{LLM}}$
/* Step3: Inject Geometry Tokens */
10: $\boldsymbol{E}' \leftarrow Insert(\boldsymbol{E}, \widehat{\boldsymbol{T}}, \boldsymbol{p}; \boldsymbol{M})$    $\triangleright$ *insert $K$ tokens at $\boldsymbol{p}$*
11: Update masks & labels to match $\boldsymbol{E}'$
12: **return** $\boldsymbol{E}'$

---

modality tokens $\widehat{\boldsymbol{T}}$ (Line 9). We locate the designated injection positions $\boldsymbol{p}$ corresponding to the modality placeholder $y_{\text{ins}}$ within the instruction sequence $\mathbf{E}$ (Line 1). The adapter then inserts the structure-aware tokens $\widehat{\boldsymbol{T}}$ into the instruction embedding (Lines 10–11). This integration allows Cuttlefish to ground its textual responses in specific structural regions. Analysis showing that geometry grounding reduces irreducible Bayes risk is discussed in App. A.2.

### 3.4. Training

**Encoder pretraining** (Fig. 3). While the EGNN encoder is not the focus of our novelty, the absence of a unified model capable of processing all-atom modalities necessitated a custom pretraining phase. We adopt an SE(3)-equivariant EGNN as the encoder to ensure the invariance to rotations and translations, which is essential for stable geometric representation. Concretely, for masked atoms and edges, the encoder predicts element identity, pairwise distances, and the injected noise on direction vectors, with total loss:

$$\mathcal{L}_{\text{enc}} = \mathcal{L}_{\text{type}} + \lambda_d \mathcal{L}_{\text{dist}} + \lambda_u \mathcal{L}_{\text{dir}}, \tag{4}$$

$$\mathcal{L}_{\text{type}} = - \sum_{i \in \mathcal{M}} \log p_\theta(a_i \mid \mathbf{h}_i), \tag{5}$$

$$\mathcal{L}_{\text{dist}} = \sum_{(i,j) \in \mathcal{E}_{\mathcal{M}}} \|\hat{d}_{ij} - d_{ij}\|_1, \quad \mathcal{L}_{\text{dir}} = \sum_{(i,j) \in \mathcal{E}_{\mathcal{M}}} \|\hat{\boldsymbol{\epsilon}}_{ij} - \boldsymbol{\epsilon}_{ij}\|_2^2, \tag{6}$$

where $\mathcal{M}$ denotes masked atoms, $\mathcal{E}_{\mathcal{M}}$ denotes edges incident to masked atoms, $a_i$ is the element type, $d_{ij} =$

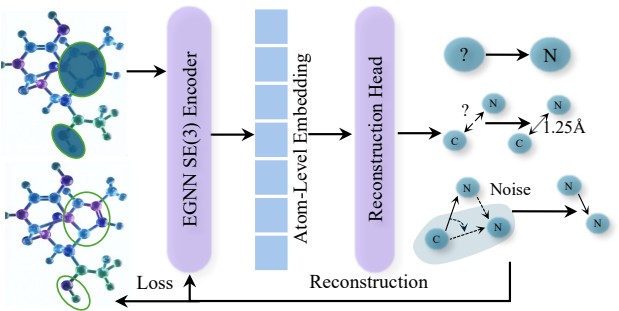

*Figure 3.* Schematic of all-atom encoder pretraining. Masked reconstruction on all-atom spatial graphs with multi-task heads. $\|\mathbf{r}_i - \mathbf{r}_j\|_2$ is the interatomic distance, and $\epsilon_{ij}$ is the synthetic noise added to a normalized direction vector.

**Modality Alignment Tuning (Connector Unfrozen).** After encoder pretraining, we freeze the EGNN encoder and the LLM, and optimize only the connector components (Geometry Grounding Adapter and Scaling-Aware Patching) on our all-atom instruction tuning corpus–GEO-AT. This isolates the optimization to the geometry-to-text interface, stabilizes training under all-atom inputs, and allows the connector to learn instruction-conditioned selection and injection without perturbing pretrained linguistic priors.

**LLM Adaptation Tuning (Connector & LLM Unfrozen).** We then unfreeze the LLM and continue end-to-end tuning with a relatively smaller learning rate to let the LLM accommodate the new modality injection distribution while preserving its general language capabilities. Reasoning-related strategy is described in App. C.4.

Unlike Q-Former (Li et al., 2023) style pipelines that require heavy contrastive pretraining to fit a fixed set of learnable query tokens as modality anchors, our structural queries are dynamically instantiated by Scaling-Aware Patching and carry explicit geometric meaning; thus, alignment is achieved directly via instruction supervised end-to-end tuning under variable length query token budgets. Detailed model and training settings are discussed in App. C.1.

# 4. Experiments

This section evaluates Cuttlefish on heterogeneous all-atom instruction datasets spanning molecules, proteins, and nucleic acids. Training and evaluation data are detailed in Sec. 4.1. Training and optimization performance are analyzed in Sec. 4.2. The performance gains beyond general LLMs are quantified in Sec. 4.3. Comparisons to modality-specific baselines are reported in Sec. 4.4. Scaling ability is discussed in Sec. 4.5, followed by core ablations in Sec. 4.6. A matched-budget connector comparison is in Sec. 4.7. Structure availability analysis is in Sec. 4.8. Backbone size ablation is in Sec. 4.9. Hallucination analysis is in Sec. 4.10. Coordinate-noise robustness is in Sec. 4.11 with

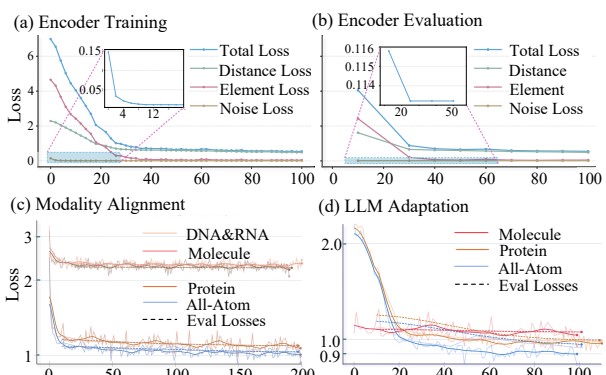

*Figure 4.* Loss (all atom vs. single modality) across training stages: encoder (a, training; b, evaluation) with objective-wise losses, then modality alignment (c) and LLM adaptation (d). Dashed lines denote evaluation loss. X-axis: global steps (scaled by 0.1).

data contamination analysis in Sec. 4.12.

## 4.1. Training & Benchmark Datasets

**Standard Benchmarks.** To fairly compare Cuttlefish with modality-specific baselines, we evaluated on widely adopted benchmarks: (1) **Mol-Instructions** (Fang et al., 2023) aggregates molecules, proteins into a unified instruction format comprising approximately 700K samples verified via human-in-the-loop quality control; (2) **DNA-Chat** (de Almeida et al., 2025) reformulates the Nucleotide Transformer benchmark (Dalla-Torre et al., 2025) into a 7.8M-sample corpus using multi-template question-answering; (3) **RNA-QA** (Xiao et al., 2024b) leverages a topic-modeling summarization pipeline and LLM-based decomposition to generate 400K+ literature-grounded QA pairs from RNA-central (Bateman et al., 2011) for functional annotation. Baselines and selection rationale are discussed in App. D.2.

**GEO-AT** (ours). Motivated by the absence of corpora with all-atom geometries for LLM tuning, we introduce GEO-AT, the first all-atom instruction dataset encompassing modalities including molecules, proteins, nucleic acids, with atom-level coordinates and language annotations. GEO-AT serves as the training and evaluation data for Cuttlefish, providing a community resource for a geometry-grounded instruction corpus, which will catalyze the next-generation all-atom LLMs. Detailed dataset construction, source, and count are discussed in App. C.2 & D.1. Text-output tasks (description, open-ended QA) are evaluated by METEOR and BERTScore; generation tasks (molecular design, reaction prediction) by validity and RDKit fingerprint similarity (FTS); closed-form QA tasks (true-or-false, multiple-choice) by accuracy. Full metric definitions are in App. D.3.

## 4.2. Training Performance

**Encoder pretraining.** Fig. 4 (a,b) shows the encoder training with three objectives with stable and quick convergence,

*Table 2.* Improvements compared to general LLM baselines on benchmark GEO-AT across four modalities (molecule, protein, DNA, RNA) measured by METEOR and BERTScore with average scores reported. The best (pink) and second-best (lightpink) are highlighted.

| Model | Molecule | | Protein | | DNA | | RNA | | Average | |
|---|---|---|---|---|---|---|---|---|---|---|
| | METEOR | BERT-S | METEOR | BERT-S | METEOR | BERT-S | METEOR | BERT-S | METEOR | BERT-S |
| *General LLM Baselines - Sequence Only - Non-Reasoning* | | | | | | | | | | |
| Qwen-2.5-7B-Instruct | 0.137 | 0.691 | 0.091 | 0.651 | 0.171 | 0.644 | 0.159 | 0.640 | 0.143 | 0.653 |
| Llama-3.1-8B-Instruct | 0.229 | 0.778 | 0.178 | 0.742 | 0.175 | 0.658 | 0.175 | 0.646 | 0.186 | 0.694 |
| Mistral-3-8B-Instruct-2512 | 0.185 | 0.732 | 0.134 | 0.714 | 0.156 | 0.665 | 0.192 | 0.652 | 0.172 | 0.683 |
| GLM-4-9B-0414 | 0.174 | 0.644 | 0.110 | 0.672 | 0.204 | 0.690 | 0.143 | 0.549 | 0.155 | 0.621 |
| *General LLM Baselines - Reasoning - Sequence Only* | | | | | | | | | | |
| Qwen3-8B | 0.103 | 0.715 | 0.044 | 0.679 | 0.131 | 0.665 | 0.128 | 0.655 | 0.107 | 0.674 |
| DeepSeek-R1-D-Qwen-7B | 0.160 | 0.721 | 0.141 | 0.707 | 0.206 | 0.659 | 0.170 | 0.687 | 0.169 | 0.692 |
| Mistral-3-8B-Reasoning-2512 | 0.147 | 0.753 | 0.108 | 0.811 | 0.264 | 0.806 | 0.180 | 0.674 | 0.176 | 0.744 |
| *General LLM Baselines - Reasoning - Sequence Only - Modality Enhanced Tokenizer* | | | | | | | | | | |
| Qwen3-8B | 0.158 | 0.709 | 0.027 | 0.573 | 0.177 | 0.792 | 0.095 | 0.773 | 0.110 | 0.724 |
| DeepSeek-R1-D-Qwen-7B | 0.204 | 0.673 | 0.219 | 0.617 | 0.147 | 0.644 | 0.283 | 0.688 | 0.227 | 0.662 |
| Mistral-3-8B-Reasoning-2512 | 0.185 | 0.823 | 0.192 | 0.755 | 0.149 | 0.756 | 0.288 | 0.579 | 0.220 | 0.698 |
| *Ours* | | | | | | | | | | |
| Cuttlefish + Qwen2.5-7B | 0.314 | 0.863 | 0.317 | 0.816 | 0.262 | 0.806 | 0.378 | 0.858 | 0.330 | 0.840 |
| Cuttlefish + Llama-3.1-8B-I | 0.391 | 0.875 | 0.417 | 0.896 | 0.529 | 0.816 | 0.403 | 0.868 | 0.428 | 0.864 |
| Cuttlefish + Mistral-3-8B-I | 0.415 | 0.653 | 0.333 | 0.858 | 0.358 | 0.852 | 0.220 | 0.694 | 0.310 | 0.750 |
| Cuttlefish + GLM-4-9B | 0.327 | 0.830 | 0.262 | 0.831 | 0.443 | 0.888 | 0.403 | 0.794 | 0.367 | 0.827 |
| Cuttlefish + Qwen3-8B | 0.389 | 0.853 | 0.377 | 0.888 | 0.391 | 0.860 | 0.491 | 0.890 | 0.428 | 0.876 |
| Cuttlefish + R1-7B | 0.342 | 0.812 | 0.327 | 0.801 | 0.330 | 0.802 | 0.422 | 0.799 | 0.369 | 0.803 |
| Cuttlefish + Mistral3-8B-R | 0.309 | 0.776 | 0.320 | 0.756 | 0.256 | 0.798 | 0.395 | 0.775 | 0.335 | 0.776 |

indicating that a huge amount of data is not needed for pre-training to let an encoder learn the spatial relations inside all-atom entities, in line with MuMo (Jing et al., 2025). For the mask strategy, we also considered masking an appropriate size (multiple residue-level) of graph regions that can capture meaningful long-range relations.

**End-to-end tuning.** Fig. 4 (c,d) illustrates the loss of two tuning stages. We report the single-modality and mixed-all-atom modality training loss, which shows that the mixed-modality converges better, indicating that the model benefits from cross-modality diversity in complexes and chemical features. Detailed training settings are in App. C.1.

## 4.3. Cuttlefish vs. General LLMs

To quantify Cuttlefish 's advantage on structural understanding and reasoning, representative open-source LLM backbones are compared, including reasoning and non-reasoning variants, on GEO-AT under identical split and finetuning settings, as shown in Tab. 2. Llama-3 (Grattafiori et al., 2024) and Qwen-3 (Yang et al., 2025a) exhibit relatively strong bio-modality understanding, motivating the adoption of Llama-3.1 as the default backbone in subsequent experiments. To mitigate sequence-level comprehension difficulty for general LLMs, a modality tokenizer is introduced that maps meaningful motifs or residues to single tokens, yielding consistent performance gains. Across different backbones, augmenting with Cuttlefish produces substantial improvements on each modality and the average, supporting **two conclusions**: (1) atom-level modalities require structure-enhanced understanding beyond sequence repre-

sentation, and (2) Cuttlefish enables stronger multimodal understanding and reasoning over heterogeneous all-atom modalities. These cross-backbone improvements confirm that explicit geometry grounding is essential for structure-conditioned understanding and reasoning (Challenge 2).

## 4.4. Cuttlefish vs. Modality-Specific Baselines

Cuttlefish is evaluated on all 17 Mol-Instruction (Fang et al., 2023) tasks, 20 tasks from DNA-Chat (de Almeida et al., 2025) and RNA-QA (Xiao et al., 2024b), comparing against modality-specific models.

**Small molecules** (Tab. 3). Performance is reported against benchmark baselines and strong molecular LLMs such as Mol-Llama (Kim et al., 2025). Tasks fall into two types: *description tasks* produce text output evaluated by METEOR, BERTScore, or accuracy; *generation tasks* (molecular design, forward/retro prediction, reagent prediction) produce molecular structures evaluated by validity and FTS. Full metrics results see App. D.4.

**Proteins** (Tab. 4). Evaluation uses five protein tasks from Mol-Instruction (Fang et al., 2023). Provided inputs are protein sequences; structures are retrieved by mapping UniProt IDs (Consortium, 2019) to PDB IDs (Burley et al., 2017). For targets missing PDB structures, AlphaFold2 (Jumper et al., 2021) is used to generate structures. Cuttlefish enables superior reasoning over both sequence-only and Q-Former style multimodal LLM baselines.

**Nucleic acids.** DNA evaluation follows the DNA-Chat (de Almeida et al., 2025) dataset, covering 18 tasks. As

*Table 3.* Performance on the Mol-Instructions molecule-related benchmark. Design, Forward, Retro, and Reagent denote molecular design, forward prediction, retrosynthesis, and reagent prediction, each reported by validity (Val) and RDKit fingerprint similarity (FTS). OpenQA reports BERTScore. T/F and MC denote True-or-False and Multi-Choice accuracy, and ER and IE denote Entity Recognition and Interaction Extraction F1. The best (pink) and second-best (lightpink) results are highlighted. Full metrics results see App. D.4. Default Cuttlefish backbone: Llama-3.1-8B-Instruct.

| Model | Captioning | | Design | | Forward | | Retro | | Reagent | | OpenQA | T/F | MC | ER | IE |
|---|---|---|---|---|---|---|---|---|---|---|---|---|---|---|---|
| | ROUGE-L↑ | METEOR↑ | Val↑ | FTS↑ | Val↑ | FTS↑ | Val↑ | FTS↑ | Val↑ | FTS↑ | BERT-S↑ | Acc↑ | Acc↑ | F1↑ | F1↑ |
| Alpaca-7B | 0.136 | 0.107 | 0.002 | 0.006 | 0.138 | 0.004 | 0.160 | 0.005 | 0.186 | 0.029 | 0.824 | 0.330 | 0.290 | 0.210 | 0.040 |
| Baize-7B | 0.148 | 0.106 | 0.002 | 0.000 | 0.097 | 0.004 | 0.112 | 0.025 | 0.099 | 0.022 | 0.811 | 0.480 | 0.240 | 0.010 | 0.000 |
| ChatGLM-6B | 0.166 | 0.129 | 0.005 | 0.005 | 0.108 | 0.050 | 0.046 | 0.056 | 0.074 | 0.017 | 0.795 | 0.180 | 0.220 | 0.150 | 0.020 |
| Llama-3-8B | 0.148 | 0.184 | 0.003 | 0.005 | 0.039 | 0.001 | 0.010 | 0.018 | 0.001 | 0.037 | 0.814 | 0.270 | 0.300 | 0.000 | 0.050 |
| Vicuna-v1.5-13B | 0.130 | 0.168 | 0.001 | 0.006 | 0.059 | 0.007 | 0.017 | 0.025 | 0.007 | 0.038 | 0.814 | 0.120 | 0.290 | 0.020 | 0.080 |
| Galactica-6.7B | 0.063 | 0.065 | 0.992 | 0.135 | 0.946 | 0.156 | 0.984 | 0.167 | 0.995 | 0.036 | 0.794 | 0.420 | 0.310 | 0.170 | 0.030 |
| Qwen-2.5-7B | 0.572 | 0.538 | 0.772 | 0.207 | 0.953 | 0.163 | 0.939 | 0.168 | 0.997 | 0.258 | 0.846 | 0.210 | 0.870 | 0.400 | 0.170 |
| MolT5 | 0.034 | 0.033 | 0.773 | 0.400 | - | - | - | - | - | - | 0.615 | 0.350 | 0.490 | 0.540 | 0.120 |
| Mol-Ins.-Llama-2 | 0.291 | 0.291 | 1.000 | 0.231 | 1.000 | 0.313 | 1.000 | 0.283 | 1.000 | 0.237 | 0.837 | 0.550 | 0.650 | 0.750 | 0.220 |
| Mol-Ins.-Llama-3.1 | 0.709 | 0.637 | 1.000 | 0.358 | 1.000 | 0.756 | 1.000 | 0.704 | 1.000 | 0.412 | 0.846 | 0.600 | 0.960 | 0.690 | 0.180 |
| Mol-LLama-2 | 0.750 | 0.701 | 0.876 | 0.254 | 1.000 | 0.724 | 1.000 | 0.362 | 1.000 | 0.256 | 0.823 | 0.560 | 0.850 | 0.740 | 0.210 |
| Mol-LLama-3.1 | 0.759 | 0.707 | 0.953 | 0.392 | 1.000 | 0.774 | 1.000 | 0.708 | 1.000 | 0.411 | 0.812 | 0.600 | 0.880 | 0.700 | 0.200 |
| Cuttlefish | 0.766 | 0.715 | 1.000 | 0.422 | 1.000 | 0.792 | 1.000 | 0.747 | 1.000 | 0.509 | 0.884 | 0.660 | 0.890 | 0.780 | 0.270 |

*Table 4.* Protein-oriented Mol-Instructions benchmarks. IE denotes interaction extraction. PF, FD, CA, and DM denote protein function, functional description, catalytic activity, and domain or motif prediction. The top-2 are highlighted in pink and lightpink. Default Cuttlefish backbone: Llama-3.1-8B-Instruct.

| Model | IE | PF | FD | CA | DM |
|---|---|---|---|---|---|
| | F1↑ | ROUGE-L↑ | ROUGE-L↑ | ROUGE-L↑ | ROUGE-L↑ |
| Alpaca-7B | 0.002 | 0.200 | 0.100 | 0.230 | 0.120 |
| Baize-7B | 0.004 | 0.200 | 0.150 | 0.220 | 0.130 |
| ChatGLM-6B | 0.003 | 0.150 | 0.140 | 0.130 | 0.100 |
| Llama-3-8B | 0.003 | 0.120 | 0.120 | 0.130 | 0.090 |
| Vicuna-v1.5-13B | 0.013 | 0.150 | 0.140 | 0.160 | 0.120 |
| Galactica-6.7B | 0.001 | 0.070 | 0.080 | 0.080 | 0.060 |
| Mol-Instructions | 0.224 | 0.430 | 0.440 | 0.520 | 0.460 |
| ProtLLM | 0.176 | 0.450 | 0.435 | 0.463 | 0.384 |
| ProteinGPT | 0.166 | 0.336 | 0.416 | 0.406 | 0.472 |
| ProtChatGPT | 0.055 | 0.355 | 0.510 | 0.306 | 0.452 |
| Cuttlefish | 0.273 | 0.486 | 0.520 | 0.551 | 0.495 |

*Table 5.* Performance on two data types from RNA-QA: Abstract Information Summary (AIS) and Divide and Conquer RNA Literature Summarization (D&C), evaluated by ROUGE-1/2/L. The best (pink) and second-best (lightpink) results are highlighted. Default Cuttlefish backbone: Llama-3.1-8B-Instruct.

| Baseline | RNA-QA (AIS) | | | RNA-QA (D&C) | | |
|---|---|---|---|---|---|---|
| | ROUGE-1↑ | ROUGE-2↑ | ROUGE-L↑ | ROUGE-1↑ | ROUGE-2↑ | ROUGE-L↑ |
| Llama3-8B | 0.236 | 0.094 | 0.204 | 0.247 | 0.096 | 0.218 |
| RNA-FM | 0.224 | 0.136 | 0.209 | 0.092 | 0.039 | 0.080 |
| RNA-GPT | 0.503 | 0.367 | 0.475 | 0.479 | 0.269 | 0.441 |
| Cuttlefish | 0.494 | 0.371 | 0.512 | 0.519 | 0.432 | 0.471 |

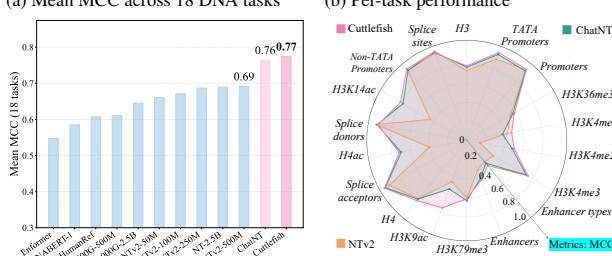

*Figure 5.* Evaluation on DNA-Chat. (a) Mean Matthews correlation coefficient (MCC) over 18 DNA tasks, comparing Cuttlefish with representative baselines (x-axis). (b) Per-task MCC on the same 18 tasks, overlaying Cuttlefish and top-2 baselines.

### 4.5. Scaling Stability and Token Efficiency

**Scaling Stability**. Figs. 6 (a–b) illustrate modality scaling results on two Mol-Instructions (Fang et al., 2023) generation tasks (molecular captioning and protein functional description), where test instances are stratified into graph node-count bins to evaluate performance under increasing structural complexity. Across these varied bins, Cuttlefish maintains remarkably stable generation quality and achieves its most significant relative gains in the high-size regime, signaling favorable scaling characteristics rather than the size-induced degradation often seen in fixed-capacity models. This trend is directly attributable to Scaling-Aware Patching: by implementing a variable query token budget rather than enforcing a rigid fixed-length bottleneck, the connector adaptively preserves critical fine-grind features regardless of the entity's scale.

**Token Efficiency.** Fig. 6 (c) reports the language/structure token fractions and Fig. 6 (d) shows the modality token usage as entity size increases. Compared to a strong baseline ChatNT (de Almeida et al., 2025), which scales token count linearly, Cuttlefish maintains a substantially lower structural token budget while scaling reasonably with the entity size.

shown in Fig. 5 (a) Cuttlefish achieves top-1 average performance, slightly exceeding ChatNT (de Almeida et al., 2025). Limited gains from atom-level inputs are consistent with substantial structural repetition in DNA. Per-task results are summarized in Fig. 5 (b). For RNA (Tab. 5), Cuttlefish is validated on benchmark RNA-QA (Xiao et al., 2024b).

Across modalities, Cuttlefish attains top-1 results on most tasks, indicating that unified all-atom structural grounding enables understanding that surpasses modality-specific models (Challenge 2).

*Table 6.* Ablation study on generation quality across modalities, evaluated by METEOR and BERTScore. DNA&RNA aggregates nucleic acids. Avg. Drop reports relative degradation from the full model. The best (pink) and second-best (lightpink) results are highlighted.

| Ablation | Molecule | | Protein | | DNA&RNA | | Average | | Avg. Drop (%) | |
|---|---|---|---|---|---|---|---|---|---|---|
| | METEOR↑ | BERTScore↑ | METEOR↑ | BERTScore↑ | METEOR↑ | BERTScore↑ | METEOR↑ | BERTScore↑ | METEOR | BERTScore |
| Cuttlefish | 0.391 | 0.875 | 0.417 | 0.896 | 0.466 | 0.842 | 0.425 | 0.871 | - | - |
| Single-modality encoder | 0.387 | 0.845 | 0.424 | 0.876 | 0.435 | 0.821 | 0.415 | 0.847 | 2.17 | 2.69 |
| Residue-level patching | 0.320 | 0.746 | 0.368 | 0.820 | 0.452 | 0.771 | 0.380 | 0.779 | 10.49 | 10.53 |
| w/o instruction-guided gating | 0.309 | 0.791 | 0.338 | 0.814 | 0.448 | 0.748 | 0.365 | 0.785 | 14.01 | 9.89 |
| Direct Projection | 0.310 | 0.796 | 0.342 | 0.815 | 0.355 | 0.678 | 0.336 | 0.763 | 20.96 | 12.38 |
| Q-Former variant (same max token length) | 0.332 | 0.856 | 0.218 | 0.774 | 0.174 | 0.687 | 0.241 | 0.772 | 43.15 | 11.31 |
| Sequence-only baseline | 0.229 | 0.778 | 0.178 | 0.742 | 0.175 | 0.652 | 0.194 | 0.724 | 54.32 | 16.83 |
| Seq-only + enhanced tokenizer | 0.203 | 0.699 | 0.141 | 0.494 | 0.218 | 0.731 | 0.187 | 0.701 | 55.89 | 19.46 |
| w/o soft patch growth | 0.136 | 0.675 | 0.123 | 0.686 | 0.163 | 0.662 | 0.141 | 0.674 | 66.83 | 22.57 |

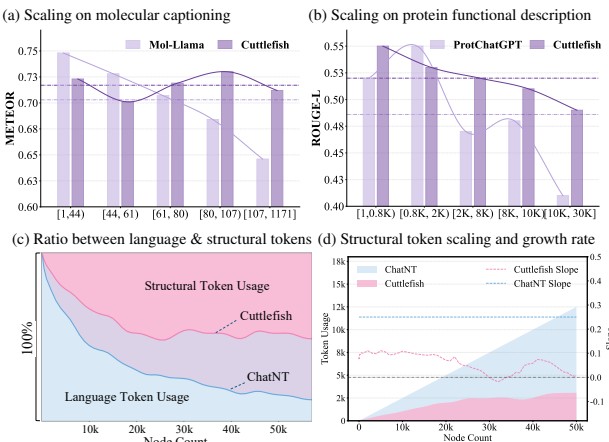

(a) Scaling on molecular captioning
(b) Scaling on protein functional description
(c) Ratio between language & structural tokens
(d) Structural token scaling and growth rate

*Figure 6.* Scaling analysis by entity size. (a–b) Mol-Instructions results for Cuttlefish versus task-specific strong baselines on molecular & protein captioning tasks, plotted over atom-count bins (x-axis); bars show bin-wise scores (METEOR / ROUGE-L), with dashed lines indicating averages and solid curves indicating trend lines. (c–d) Scaling versus ChatNT over the node-count axis: (c) language/structural token fractions, and (d) structural-token usage (area) with growth-rate indicators (dashed).

Detailed training, token and memory efficiency analysis see App. D.7. These results confirm that adaptive token allocation effectively addresses the fixed-budget bottleneck (Challenge 1).

### 4.6. Main Ablations

**Settings.** Tab. 6 reports the core ablations of Cuttlefish, including: (i) single-modality instead of mixed all-atom, (ii) residue-level chemically meaningful patches in place of the proposed patching, (iii) removal of instruction guidance in the anchor gate, (iv) replacement of fusion blocks with direct projection, (v) replacement of the connector with standard Q-Former (Li et al., 2023), (vi) a sequence-only baseline with the enhanced tokenizer, and (vii) removal of soft patch growth.

**Results.** Single-modality training yields the smallest degradation, indicating that the architecture remains effective under unimodal inputs, while mixed-modality training provides additional gains, consistent with cross-entity trans-

*Table 7.* Matched-budget comparison between fixed-length Q-Former connectors and Cuttlefish on GEO-AT with other parts use Cuttlefish's settings. BERTScore is reported (higher is better), together with the overall average.

| Connector | Token policy | Molecule↑ | Protein↑ | DNA&RNA↑ | Average↑ |
|---|---|---|---|---|---|
| Q-Former | Fixed 256 | 0.778 | 0.743 | 0.658 | 0.726 |
| Q-Former | Fixed 512 | 0.781 | 0.772 | 0.680 | 0.744 |
| Q-Former | Fixed 1024 | 0.809 | 0.784 | 0.712 | 0.768 |
| Q-Former | Fixed 2048 | 0.768 | 0.767 | 0.745 | 0.760 |
| Cuttlefish | Matched to 256 | 0.842 | 0.659 | 0.693 | 0.731 |
| Cuttlefish | Matched to 512 | 0.853 | 0.776 | 0.748 | 0.792 |
| Cuttlefish | Matched to 1024 | 0.875 | 0.798 | 0.750 | 0.808 |
| Cuttlefish | Matched to 2048 | 0.875 | 0.896 | 0.816 | 0.862 |

*Table 8.* Missing-structure analysis on GEO-AT. BERTScore (higher is better) is reported with the overall average.

| Variant | Molecule↑ | Protein↑ | DNA↑ | RNA↑ | Average↑ |
|---|---|---|---|---|---|
| Base LLM (no tuning) | 0.778 | 0.742 | 0.658 | 0.646 | 0.706 |
| Sequence-only | 0.752 | 0.820 | 0.710 | 0.698 | 0.745 |
| w/o Structure at inference | 0.867 | 0.788 | 0.730 | 0.860 | 0.811 |
| Full Cuttlefish | 0.875 | 0.896 | 0.816 | 0.868 | 0.864 |

fer of chemical relations. Residue-level patching incurs a ~10% drop, suggesting that chemically meaningful segmentation is not optimal under the target objectives and that lack of instruction conditioning limits focus allocation.

Combined ablations further substantiate the necessity of both proposed novelties. The Q-Former connector performs competitively on molecules but degrades on proteins and nucleic acids, consistent with a fixed-length bottleneck as structural complexity increases. Finally, the sequence-only baseline underperforms, supporting the claim that sequence-only inputs are insufficient. Extended ablation experiments are discussed in App. E.

### 4.7. Matched-Budget Connector Comparison

To isolate whether gains come from adaptivity or simply from using more tokens, we compare Cuttlefish against Q-Former connectors at matched budgets (Tab. 7). The adaptive policy consistently outperforms fixed-length counterparts across all budgets and modalities, with the largest margins on proteins and DNA&RNA, confirming that adaptivity mainly drives the improvement (Challenge 1).

*Table 9.* Backbone-size ablation on GEO-AT using the Qwen-2.5 family. Results are reported by modality with BERTScore (higher is better) used as the evaluation metric.

| MODEL SIZE | MOLECULE↑ | PROTEIN↑ | DNA&RNA↑ | AVERAGE↑ |
|---|---|---|---|---|
| 0.5B | 0.453 | 0.422 | 0.382 | 0.419 |
| 1.5B | 0.586 | 0.577 | 0.544 | 0.569 |
| 3B | 0.608 | 0.578 | 0.621 | 0.602 |
| 7B | 0.863 | 0.816 | 0.832 | 0.840 |
| 14B | 0.897 | 0.861 | 0.834 | 0.864 |
| 32B | 0.903 | 0.877 | 0.840 | 0.873 |

*Table 10.* Hallucination evaluation on GEO-AT across three failure modes: Func.-group (functional-group prediction), Mech.-expl. (mechanism explanation), and L.-range (long-range interaction). HR: hallucination rate (↓); AR: answer rate (↑).

| Model | Func.-group | | Mech.-expl. | | L.-range | | Avg. | |
|---|---|---|---|---|---|---|---|---|
| | HR | AR | HR | AR | HR | AR | HR | AR |
| *Molecule* | | | | | | | | |
| Mol-LLaMA | 0.12 | 0.94 | 0.23 | 0.98 | 0.19 | 1.00 | 0.18 | 0.97 |
| 3D-MoLM | 0.23 | 0.83 | 0.72 | 0.95 | 0.32 | 0.81 | 0.42 | 0.86 |
| Cuttlefish | 0.07 | 0.99 | 0.13 | 0.99 | 0.16 | 1.00 | 0.12 | 0.99 |
| *Protein* | | | | | | | | |
| ProtChatGPT | 0.10 | 0.94 | 0.20 | 0.96 | 0.24 | 1.00 | 0.18 | 0.97 |
| Prot2Chat | 0.06 | 0.95 | 0.71 | 0.89 | 0.33 | 0.76 | 0.37 | 0.86 |
| Cuttlefish | 0.04 | 0.97 | 0.16 | 0.98 | 0.20 | 0.98 | 0.13 | 0.98 |

### 4.8. Structure Availability Analysis

Tab. 8 examines performance when structural input is withheld at inference. A model trained with structure but evaluated without it still substantially outperforms sequence-only fine-tuning, showing that structure-aware training instills lasting geometric priors even when coordinates are absent at test time (Challenge 2).

### 4.9. Backbone Size Ablation

Tab. 9 scales the backbone from 0.5B to 32B within the Qwen-2.5 family. Performance improves monotonically across all modalities, with the largest gain between 3B and 7B, confirming that Cuttlefish's connector does not bottleneck backbone capacity (Challenge 2). This trend holds across both text-generation and closed-form QA tasks.

### 4.10. Hallucination Analysis

Tab. 10 evaluates hallucination across three failure modes (functional-group prediction, mechanism explanation, and long-range interaction) on molecules and proteins. Cuttlefish achieves the lowest hallucination rate and highest answer rate across all modes and modalities, confirming that geometry grounding reduces structural hallucination (Challenge 2). The advantage is most pronounced on the long-range interaction mode. Inference and Reasoning examples are shown in App. D.5.

### 4.11. Coordinate-Noise Robustness

Tab. 11 injects Gaussian coordinate perturbations of increasing magnitude into protein inputs to test structural noise

*Table 11.* Coordinate-noise robustness analysis on the GEO-AT protein tasks. Isotropic Gaussian noise with different standard deviations is added to the input coordinates, and ROUGE-L (higher is better) is reported for protein function (PF), functional description (FD), catalytic activity (CA), and domain or motif prediction (DM), together with the average.

| NOISE (Å) | PF↑ | FD↑ | CA↑ | DM↑ | AVERAGE↑ |
|---|---|---|---|---|---|
| 0 | 0.486 | 0.520 | 0.551 | 0.495 | 0.513 |
| 0.25 | 0.523 | 0.475 | 0.521 | 0.517 | 0.509 |
| 0.5 | 0.526 | 0.418 | 0.475 | 0.452 | 0.468 |
| 1 | 0.460 | 0.438 | 0.477 | 0.405 | 0.445 |
| 2 | 0.443 | 0.396 | 0.422 | 0.432 | 0.423 |

robustness. Performance degrades gracefully with noise level, confirming that Cuttlefish does not overfit to precise atomic coordinates and remains reliable under realistic structural uncertainty (Challenge 2).

### 4.12. Data Contamination Analysis

Data contamination is a common concern in LLM training. We argue that our encoder pretraining and alignment tuning are not highly data-dependent, as the model is trained to recognize structural patterns rather than memorize new factual knowledge (details in App. D.6). To ensure a fair comparison, we remove all benchmark test-set entities from our training set by entity ID and apply a 13-gram overlap filter (Brown et al., 2020). To isolate structural contributions from LLM pretrained knowledge, we train the same LLM backbone without Cuttlefish's structural components on the identical corpus; the resulting gap confirms that performance gains stem from structural grounding rather than data exposure (Tab. 2).

## 5. Conclusion

Cuttlefish establishes a unified framework for structure-grounded reasoning across all-atom modalities. By integrating Scaling-Aware Patching, the architecture enables instruction-conditioned query token scaling, bypassing fixed-length connector bottlenecks while preserving fine-grained structural signals. Building on these adaptive representations, Geometry Grounding Adapter injects verifiable geometric cues into the LLM to mitigate structural hallucinations. Empirical results across all-atom modality benchmarks show consistent gains, and support four observations: (1) sequence-only inputs are insufficient; (2) fixed-length connectors impose an information bottleneck for capturing geometry-dependent semantics; (3) structural-aware LLMs can internalize atom-level structural features with appropriate alignment; and (4) joint modality training improves per-modality learning. Overall, these findings indicate that Scaling-Aware Patching and Geometry Grounding Adapter are key to robust, unified multimodal scientific reasoning. Limitations and future works are discussed in App. F.

## Acknowledgements

This work was supported in part by the Canada Research Chairs Tier II Program (CRC-2021-00482), the Canadian Institutes of Health Research (PLL 185683, PJT 190272, PJT204042, CFA-205059), the Natural Sciences and Engineering Research Council of Canada (RGPIN-2026-07632, ALLRP 602759-24), and The Canada Foundation for Innovation (CFI) John R. Evans Leaders Fund (JELF) program (#43481).

## Impact Statement

This paper aims to advance machine learning by improving structure-aware LLMs for all-atom modalities. It introduces mechanisms that ground language reasoning in geometric evidence and scale modality tokens with structural complexity to improve faithfulness and scalability. The work is intended for general scientific modeling settings, and no broader societal impacts beyond those common to this research area are expected.

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

# Appendix Table of Contents

# A. Mechanistic Analysis

This appendix provides a mechanism-level analysis of the proposed connector. We focus on two operational properties that are central to the novelty of Cuttlefish. First, Scaling-Aware Patching adaptively allocates structural tokens to instruction-relevant regions, thereby mitigating the fixed-length bottleneck under variable all-atom complexity. Second, Geometry Grounding Adapter exposes geometry-conditioned evidence to the language model, thereby reducing ambiguity when the target depends on structural information that is not recoverable from the sequence alone.

## A.1. Mechanistic Analysis of Adaptive Structural Compression

We first analyze Scaling-Aware Patching as an adaptive structural compression mechanism. The goal of this subsection is to characterize how instruction-conditioned anchor selection and soft patch growth preserve relevant structural information while operating under a constrained modality-token budget.

**Setup.** Fix one graph $g$ with node set $I_g$, coordinates $\{P_i\}_{i \in I_g}$, encoder embeddings $\{X_i\}_{i \in I_g}$, and gate probabilities

$$p_i := \mathrm{Softmax}(\ell_{I_g})_i, \qquad i \in I_g.$$

Let $\pi$ be the permutation such that

$$p_{\pi_1} \geq p_{\pi_2} \geq \cdots \geq p_{\pi_{|I_g|}},$$

and define the adaptive anchor count and selected anchor set exactly as in Eq. (1):

$$k_g := \min\Big\{k \in \{1, \ldots, |I_g|\} : \sum_{j=1}^{k} p_{\pi_j} \geq \rho\Big\}, \qquad S_g := \{\pi_1, \ldots, \pi_{k_g}\}.$$

For each $i \in I_g$ and $a \in S_g$, let $W_{i,a}$ be the soft assignment from Eq. (2), so that

$$W_{i,a} \geq 0, \qquad \sum_{a \in S_g} W_{i,a} = 1.$$

Define the normalized pooling weights

$$\bar{W}_{j,a} := \frac{W_{j,a}}{\sum_{u \in I_g} W_{u,a}}, \qquad a \in S_g,$$

the patch token

$$t_a := \sum_{j \in I_g} \bar{W}_{j,a} X_j, \qquad a \in S_g,$$

and the patch-only reconstruction

$$\hat{X}_i := \sum_{a \in S_g} W_{i,a} t_a, \qquad i \in I_g.$$

Although the model does not explicitly reconstruct each node, $\hat{X}_i$ is the natural proxy for the structural information that remains available after compression into patch tokens.

**Proposition A.1** (Minimal-cardinality relevance coverage). *The selected anchor set $S_g$ has the following properties:*

$$\sum_{j=1}^{k_g} p_{\pi_j} = \sum_{i \in S_g} p_i \geq \rho,$$

$$\forall S \subseteq I_g \text{ with } |S| < k_g, \qquad \sum_{i \in S} p_i < \rho,$$

$$\forall S \subseteq I_g \text{ with } |S| = k_g, \qquad \sum_{i \in S} p_i \leq \sum_{i \in S_g} p_i.$$

*Hence $S_g$ is the* smallest-cardinality *set that attains relevance mass at least $\rho$, and among all sets of size $k_g$, it captures the largest total gate mass.*

(a) Smallest-Cardinality Budget Selection

(b) Bounded Distortion in Patch Growth

*Figure 7.* Illustration of the two components in the adaptive compression analysis. (a) Smallest-cardinality budget selection by sorting node relevance scores and choosing the minimal anchor set $S_g$ whose cumulative mass reaches the threshold $\rho$; (b) soft patch growth from relevant anchors to nearby candidate nodes, guided by node-to-anchor distance and edge-based scoring.

*Proof.* The first claim is immediate from the definition of $k_g$. For the second claim, let $m < k_g$ and let $S \subseteq I_g$ with $|S| = m$. Since $(p_{\pi_j})_{j=1}^{|I_g|}$ is sorted in descending order,

$$\sum_{i \in S} p_i \leq \sum_{j=1}^{m} p_{\pi_j}.$$

But $m < k_g$, and $k_g$ is the smallest index at which the cumulative mass reaches $\rho$, so

$$\sum_{j=1}^{m} p_{\pi_j} < \rho.$$

This proves the second claim.

For the third claim, let $S \subseteq I_g$ with $|S| = k_g$. Again by the sorting of $(p_{\pi_j})$,

$$\sum_{i \in S} p_i \leq \sum_{j=1}^{k_g} p_{\pi_j} = \sum_{i \in S_g} p_i.$$

Therefore $S_g$ is optimal among all subsets of size $k_g$.

Fig. 7 illustrates these two components of the adaptive compression analysis.

**Assumption A.2** (Local smoothness of structural signal). There exists a constant $L > 0$ such that

$$\|X_i - X_j\| \leq L\|P_i - P_j\|_2 \qquad \text{for all } i, j \in I_g.$$

**Theorem A.3** (Instruction-weighted compression distortion bound). *Under Assumption A.2,*

$$\sum_{i \in I_g} p_i \|X_i - \hat{X}_i\| \; \leq \; L \sum_{i \in I_g} p_i \sum_{a \in S_g} W_{i,a} \Big( \|P_i - P_a\|_2 + r_a \Big),$$

*where the average radius of patch $a$ is*

$$r_a := \sum_{j \in I_g} \bar{W}_{j,a} \|P_j - P_a\|_2.$$

*Therefore the information loss caused by the patch bottleneck is controlled by two explicit geometric quantities: the distance from each node to its assigned anchors, and the average radius of the pooled anchor patches.*

*Proof.* For any $i \in I_g$,

$$\hat{X}_i = \sum_{a \in S_g} W_{i,a} t_a = \sum_{a \in S_g} \sum_{j \in I_g} W_{i,a} \bar{W}_{j,a} X_j.$$

Since $\sum_{a \in S_g} W_{i,a} = 1$ and $\sum_{j \in I_g} \bar{W}_{j,a} = 1$ for each $a$, we have

$$\sum_{a \in S_g} \sum_{j \in I_g} W_{i,a} \bar{W}_{j,a} = 1.$$

Hence

$$X_i - \hat{X}_i = \sum_{a \in S_g} \sum_{j \in I_g} W_{i,a} \bar{W}_{j,a} (X_i - X_j),$$

and by the triangle inequality,

$$\|X_i - \hat{X}_i\| \leq \sum_{a \in S_g} \sum_{j \in I_g} W_{i,a} \bar{W}_{j,a} \|X_i - X_j\|$$
$$\leq L \sum_{a \in S_g} \sum_{j \in I_g} W_{i,a} \bar{W}_{j,a} \|P_i - P_j\|_2.$$

Applying the triangle inequality again in coordinate space,

$$\|P_i - P_j\|_2 \leq \|P_i - P_a\|_2 + \|P_j - P_a\|_2,$$

so

$$\|X_i - \hat{X}_i\| \leq L \sum_{a \in S_g} \sum_{j \in I_g} W_{i,a} \bar{W}_{j,a} \Big( \|P_i - P_a\|_2 + \|P_j - P_a\|_2 \Big)$$
$$= L \sum_{a \in S_g} W_{i,a} \|P_i - P_a\|_2 + L \sum_{a \in S_g} W_{i,a} \sum_{j \in I_g} \bar{W}_{j,a} \|P_j - P_a\|_2$$
$$= L \sum_{a \in S_g} W_{i,a} \Big( \|P_i - P_a\|_2 + r_a \Big).$$

Multiplying by $p_i$ and summing over $i \in I_g$ proves the claim.

**Corollary A.4** (Interpretation). *If the selected anchors stay close to instruction-relevant nodes and the resulting patches have small average radii, then the instruction-weighted distortion of the structural bottleneck is small. This formalizes the mechanism by which Scaling-Aware Patching preserves fine-grained geometry. Fig. 8 shows a detailed example of this soft patch growth process.*

**Theorem A.5** (Lower bound for fixed-length representative compression). *Let $v_1, \ldots, v_M \in \mathbb{R}^d$ denote $M$ instruction-relevant region descriptors, and assume they are pairwise separated:*

$$\|v_m - v_{m'}\| > 2\varepsilon \qquad \text{for all } m \neq m'.$$

*For any fixed set of $K < M$ representative tokens $c_1, \ldots, c_K \in \mathbb{R}^d$, define the covering distortion*

$$\mathcal{D}(c_1, \ldots, c_K) := \frac{1}{M} \sum_{m=1}^{M} \min_{1 \leq k \leq K} \|v_m - c_k\|.$$

*Then*

$$\mathcal{D}(c_1, \ldots, c_K) > \varepsilon \Big( 1 - \frac{K}{M} \Big).$$

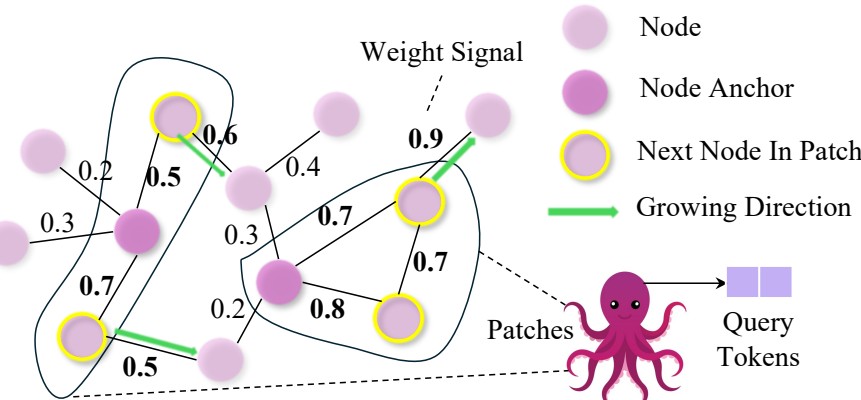

*Figure 8.* Detailed example of soft patch growth from selected node anchors. Edge weights indicate the growth signal between neighboring nodes, yellow circles mark the next nodes added into each patch, green arrows indicate the expansion direction, and the resulting patches are mapped to query tokens.

*Proof.* A single representative $c_k$ can lie within distance $\varepsilon$ of at most one descriptor $v_m$. Indeed, if for some $m \neq m'$,

$$\|v_m - c_k\| \leq \varepsilon \qquad \text{and} \qquad \|v_{m'} - c_k\| \leq \varepsilon,$$

then by the triangle inequality,

$$\|v_m - v_{m'}\| \leq \|v_m - c_k\| + \|c_k - v_{m'}\| \leq 2\varepsilon,$$

contradicting the pairwise separation assumption.

Therefore at most $K$ of the $M$ descriptors can be covered within distance $\varepsilon$ by the $K$ representatives. The remaining $M - K$ descriptors must each satisfy

$$\min_{1 \leq k \leq K} \|v_m - c_k\| > \varepsilon.$$

Hence

$$\mathcal{D}(c_1, \ldots, c_K) = \frac{1}{M} \sum_{m=1}^{M} \min_{1 \leq k \leq K} \|v_m - c_k\| > \frac{M - K}{M} \varepsilon = \varepsilon \left(1 - \frac{K}{M}\right).$$

**Corollary A.6** (Mechanistic meaning). *Theorem A.5 does* not *claim a lower bound for every possible neural connector. It establishes a lower bound for fixed-length representative-set compression under a standard covering-distortion surrogate. Under this surrogate, once the number of well-separated relevant regions exceeds the fixed token budget $K$, a nonzero compression error is unavoidable. Fig. 9 illustrates the contrast between adaptive and fixed-length compression strategies.*

## A.2. Geometry Grounding Reduces Irreducible Ambiguity

Let $S$ denote the sequence-level input, $G$ the geometry, $Z$ the instruction, and $Y$ the target answer. Let

$$T = T_\theta(S, G, Z)$$

denote the geometry-grounded tokens produced after the Geometry Grounding Adapter and injected into the LLM.

**Theorem A.7** (Bayes-risk reduction under log loss). *Under log loss, the Bayes-optimal sequence-only risk is*

$$\mathcal{R}_{\text{seq}}^* = H(Y \mid S, Z),$$

*whereas the Bayes-optimal geometry-grounded risk is*

$$\mathcal{R}_{\text{geom}}^* = H(Y \mid S, T, Z).$$

*Consequently,*

$$\mathcal{R}_{\text{geom}}^* \leq \mathcal{R}_{\text{seq}}^*,$$

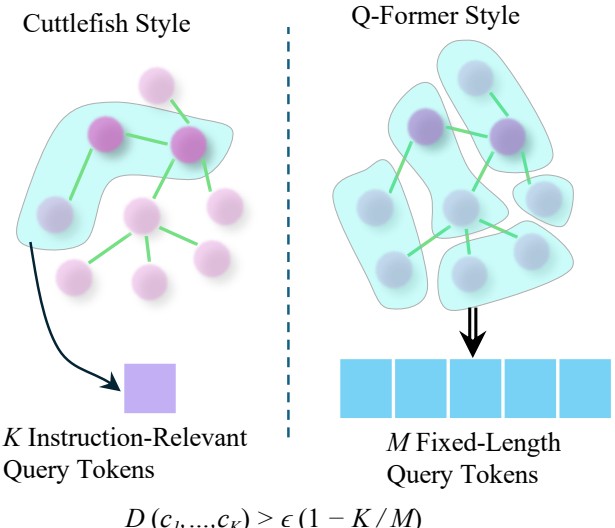

Cuttlefish Style      Q-Former Style

$K$ Instruction-Relevant Query Tokens     $M$ Fixed-Length Query Tokens

$$D\left(c_1, ..., c_K\right) > \epsilon\left(1 - K/M\right)$$

*Figure 9.* Comparison between adaptive instruction-relevant query construction and fixed-length query compression.

*and equality holds if and only if*

$$I(Y;T \mid S, Z) = 0.$$

*Equivalently,*

$$\mathcal{R}^*_{\text{seq}} - \mathcal{R}^*_{\text{geom}} = I(Y;T \mid S, Z).$$

*Proof.* For log loss, the Bayes-optimal predictor is the true conditional distribution, and the minimum expected loss equals the corresponding conditional entropy. Therefore

$$\mathcal{R}^*_{\text{seq}} = H(Y \mid S, Z), \qquad \mathcal{R}^*_{\text{geom}} = H(Y \mid S, T, Z).$$

Since conditioning cannot increase entropy,

$$H(Y \mid S, T, Z) \leq H(Y \mid S, Z).$$

Moreover, by the definition of conditional mutual information,

$$I(Y;T \mid S, Z) = H(Y \mid S, Z) - H(Y \mid S, T, Z).$$

Rearranging yields the stated equality and the inequality.

**Corollary A.8** (Interpretation). *Whenever geometry-grounded tokens $T$ contain answer-relevant information that is not identifiable from sequence alone, explicit geometry grounding strictly decreases the irreducible uncertainty of the task. This gives a precise uncertainty-level explanation for why geometry grounding can reduce hallucination. Fig. 10 illustrates this argument under three input conditions.*

### A.3. Definitions on Scaling and Modality Scope

**Scaling Definition (Entity Complexity).** In this paper, **scaling denotes all-atom entity complexity scaling, defined directly by the atom count $N_{\text{atom}}$ of the input entity.** We do not study backbone scaling (e.g., parameter count, depth) and do not attribute scaling effects to increasing LLM capacity. Scaling arises from larger structural entities containing more atoms and thus richer local environments and long-range relations. When structural inputs are tokenized, the resulting structural token count is a consequence of $N_{\text{atom}}$, not the definition of scaling.

**Scope of "Molecules".** In this work, the term **molecules refers to small-molecule chemical entities**, including drug-like compounds, ligands, metabolites, and related low–molecular-weight species. These entities are characterized by moderate atom counts and well-defined stereochemistry. Large macromolecules are explicitly excluded from this category; in particular, proteins and nucleic acids are not treated as molecules for scaling or evaluation purposes in this paper.

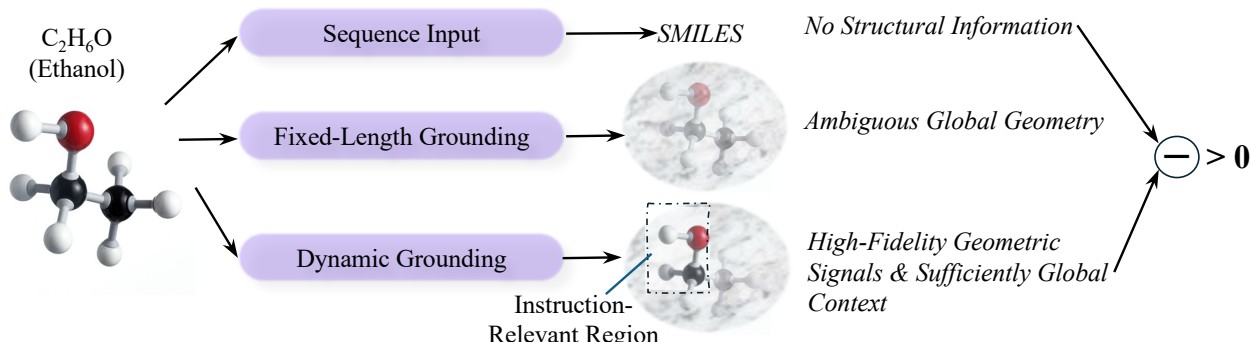

*Figure 10.* Illustration of the geometry-grounding argument under three input conditions. The top branch uses sequence alone, the middle branch uses fixed-length grounding, and the bottom branch uses dynamic grounding over the instruction-relevant region, showing the different levels of structural information retained before prediction.

*Table 12.* Comparison of All-Atom Models for Protein Structure and Molecular Design. Representative works are categorized by their architectural intent, including structural representation, generative design, and LLM-guided optimization.

| Model / Paper | What it does | Category |
|---|---|---|
| AlphaFold1 (Senior et al., 2020) | End-to-end protein structure prediction | Structural ML |
| AlphaFold2 (Jumper et al., 2021) | Near-experimental protein folding at scale | Structural ML |
| AlphaFold3 (Abramson et al., 2024) | Joint prediction of biomolecular interactions | Structural foundation model |
| ATOMICA (Fang et al., 2025a) | Universal atom-level interaction representations | Structural representation model |
| ODesign (Zhang et al., 2025a) | World-model for interaction-driven design | Structural generative model |
| BoltzGen (Stark et al., 2025) | Generative universal protein binder design | Binder generative model |
| All-atom Diffusion Transformers (Joshi et al., 2025) | Unified atomistic diffusion modeling | Structural foundation model |
| MolAct (Yang et al., 2025b) | Agent-based molecular editing and optimization | Molecular agent model |
| CIDD (Gao et al., 2025) | LLM-guided structure-based drug design | Structure-grounded LLM |

**Scope of "Proteins".** The **protein modality encompasses a broad granularity range**, including short peptides, partial protein sequences or domains, and full-length functional proteins. Scaling within this modality reflects variations in amino-acid length and corresponding all-atom representations, while remaining within the biochemical definition of proteins.

**DNA and RNA Structures.** For nucleic acids, we consider only **sequence-derived all-atom structures** that are directly displayed from linear DNA or RNA sequences. Higher-order folding phenomena occurring in the cellular nucleus or cytoplasm are not modeled. This design choice is motivated by: (i) the extreme complexity and context-dependence of in vivo nucleic-acid folding, which is beyond the scope of unified all-atom learning; (ii) the objective of maintaining a consistent structural abstraction across modalities; and (iii) the empirical observation that nucleic acids are not the primary performance driver of the proposed framework. The modest yet consistent improvements observed on DNA/RNA benchmarks support the feasibility of this simplified representation, while full nucleic-acid folding is left for future investigation.

## B. Related Literature Review

This appendix complements the main-text related work and summarizes it as Tab. 12 and Tab. 13. This section emphasizes two threads: structure-grounded biology LLMs organized by fusion strategy, and recent atom-level structural foundation and generative models that demonstrate feasible all-atom scaling.

### B.1. Structure LLMs: From Sequence Supervision to Structure Grounding

**Modality scope.** Prior biology LLMs span molecules, proteins, and nucleic acids, with parallel development patterns across modalities. Molecular systems cover representation alignment, captioning, reasoning, and instruction following; protein systems extend from sequence alignment to open-ended dialogue and structure-conditioned interaction; nucleic-acid systems include unified DNA and RNA modeling and RNA-focused instruction tuning.

**Integration strategies.** Existing approaches can be organized by how structural information is exposed to the language model. **(i) Sequence-only modeling.** Several baselines rely on sequence inputs without explicit structural grounding, including MolT5 (Edwards et al., 2022) for molecules, ProtST (Xu et al., 2023b) for proteins, RNA-GPT (Xiao et al., 2024b) for nucleic acids, and ChatNT (de Almeida et al., 2025) via sequence concatenation across complex. This line prioritizes language modeling convenience but leaves geometry implicit.

**(ii) Embedding-only alignment.** Contrastive co-embedding methods align molecular structure and text for retrieval and editing (MoleculeSTM (Liu et al., 2023b), MoMu (Su et al., 2022)), and similar embedding-only modeling appears for drug target interaction (LLM3-DTI (Zhang et al., 2025b)). These methods provide strong alignment primitives yet do not directly instantiate token-level structural reasoning.

**(iii) Fixed-length projection and feature injection.** A common design projects encoder outputs into a fixed token budget, as in InstructMol (Cao et al., 2025) and Chem3DLLM (Jiang et al., 2025), or injects encoder features into prompts, as in ProteinGPT (Guo et al., 2023). These designs are simple and effective, but they enforce constant-capacity structure summaries.

**(iv) Q-Former-style query connectors and variants.** Many models adopt fixed-length query tokens that attend to encoder representations, producing a small set of modality anchors. This connector pattern appears in molecular settings (GIT-Mol (Liu et al., 2023a), 3D-MoLM (Li et al., 2024), UniMoT (Guo et al., 2025b), MolCA (Liu et al., 2023c)), protein dialogue settings (ProtChatGPT (Wang et al., 2024)), and materials QA (MatterChat (Tang et al., 2025)), with instruction-conditioned variants (Prot2Chat (Wang et al., 2025b), LLaMo (Park et al., 2024)) and multi-stream extensions. In parallel, cross-attention bridges can be framed as tokenizing structural embeddings through attention, as in Graph2Token (Wang et al., 2025a). DeepMolTex (Yan et al., 2025) extends projection fusion with multi-scale projectors spanning atom, motif, and molecule representations, still under a fixed-length fusion regime.

### B.2. The Development of All-Atom Modeling

A distinct, rapidly maturing line demonstrates that atom-resolved modeling can scale to realistic biomolecular complexity (Tab. 12). Structural ML and structural foundation models progress from protein folding at scale (AlphaFold (Senior et al., 2020), AlphaFold2 (Jumper et al., 2021)) to joint biomolecular interaction prediction (AlphaFold3 (Abramson et al., 2024)). Atom-level representation and generative modeling further expand the capability envelope, including universal atom-level interaction representations (ATOMICA (Fang et al., 2025a)), interaction-driven world-model design (ODesign (Zhang et al., 2025a)), binder design (BoltzGen (Stark et al., 2025)), and unified atomistic diffusion modeling (All-atom Diffusion Transformers). Complementary directions connect atom-level structure to downstream design and optimization pipelines, including agent-based molecular editing (MolAct (Yang et al., 2025b)), LLM-guided structure-based drug design (CIDD (Gao et al., 2025)).

Collectively, these results motivate an all-atom perspective for language grounding: atom-resolved representations are now practical, and generative models can operate at atomistic resolution, suggesting a viable path to move beyond residue-level abstractions in structure-grounded LLMs.

### B.3. Q-Former Development and Widespread Use

Q-Former-style connectors operationalize cross-modal grounding by introducing a fixed set of learnable query tokens that attend to modality encoder outputs, producing a compact token sequence consumable by an LLM (Li et al., 2023). In the biology and chemistry literature summarized in Tab. 13, this mechanism is repeatedly adopted as a general-purpose fusion layer, spanning molecules (GIT-Mol (Liu et al., 2023a), 3D-MoLM (Li et al., 2024), UniMoT (Consortium, 2019), MolCA (Liu et al., 2023c), LLaMo (Park et al., 2024)), proteins (ProtChatGPT (Wang et al., 2024), Prot2Chat (Wang et al., 2025b)), and even materials (MatterChat (Tang et al., 2025)). The prevalence of this pattern reflects two practical advantages: architectural modularity and compatibility with instruction tuning while maintaining an LLM-native token interface.

### B.4. Recurring Limitations Across Prior Designs

The models in Tab. 13 expose three recurring limitations that align with the main-text discussion.

**Sequence-only under-specifies geometry.** Sequence-only and sequence-concatenation approaches do not explicitly represent 3D structure, limiting structure-faithful understanding and generation when geometry is essential (MolT5 (Edwards

*Table 13.* Comparison of Structure-Grounded Foundation Models categorized by Domain and Fusion Strategy. The models are organized into five primary domains based on the chemical entity they represent: Molecular LLMs, Protein LLMs, Nucleic Acid LLMs, and Materials LLMs focus on generative or reasoning tasks, while Embedding Models prioritize latent-space alignment for retrieval.

| Model | Core Idea | Category | Fusion |
|---|---|---|---|
| MoleculeSTM (Liu et al., 2023b) | Contrastive alignment between molecular structures and text for retrieval and text-guided molecular editing | Molecular Embedding | Embedding only |
| MoMu (Su et al., 2022) | Contrastive graph–text alignment supporting retrieval and molecular captioning | Molecular LLM | Fixed length projection fusion |
| MolT5 (Pei et al., 2024) | Sequence based molecular text generation without explicit structural grounding | Molecular LLM | Sequence only |
| InstructMol (Cao et al., 2025) | Instruction tuned molecular reasoning with graph encoders aligned to text | Molecular LLM | Fixed length projection fusion |
| GIT-Mol (Liu et al., 2023a) | Adapter-based instruction following over sequence, graph, and image modalities | Molecular LLM | Fixed-length Q-Former style connector |
| 3D-MoLM (Li et al., 2024) | Q-Former queries attending to 3D molecular representations from structure encoders | Molecular LLM | Fixed length Q-Former style connector |
| UniMoT (Guo et al., 2025b) | Unified molecular tokenizer for sequences with Q-Former based structural grounding | Molecular LLM | Fixed length Q-Former style connector |
| Graph2Token (Wang et al., 2025a) | Graph–text bridge using cross attention to convert graph embeddings into token representations | Molecular LLM | Fixed length cross attention fusion |
| MolCA (Liu et al., 2023c) | Joint modeling of molecular sequence and structure via Q-Former alignment | Molecular LLM | Fixed length Q-Former style connector |
| LLaMo (Park et al., 2024) | Multi-stream integration of sequence, graph tokens, and instructions for molecular reasoning | Molecular LLM | Fixed length multi-stream Q-Former connector |
| DeepMolTex (Yan et al., 2025) | Multi-scale graph projectors aligning atom, motif, and molecule representations with text | Molecular LLM | Fixed length multi-stream projection fusion |
| ProtST (Xu et al., 2023b) | Contrastive alignment between protein sequences and textual descriptions | Protein LLM | Sequence only |
| ProtLLM (Zhuo et al., 2024) | Protein sequence encoding followed by text fusion for language modeling | Protein LLM | Sequence concatenation fusion |
| Prot2Text-V2 (Fei et al., 2025) | Encoder-based protein sequence representation fused with text for generation | Protein LLM | Sequence concatenation fusion |
| ProteinGPT (Xiao et al., 2024a) | Injection of protein sequence and structure features into LLM prompts with instruction tuning | Protein LLM | Encoder feature injection |
| ProtChatGPT (Wang et al., 2024) | Instruction guided protein chat with structural embedding grounding | Protein LLM | Fixed length Q-Former style connector |
| Chem3DLLM (Jiang et al., 2025) | Structure–text fusion for protein reasoning using learned projections | Protein LLM | Fixed length projection fusion |
| Prot2Chat (Wang et al., 2025b) | Instruction guided adapters integrating protein structure and text | Protein LLM | Fixed length instruction conditioned Q-Former connector |
| ChatNT (de Almeida et al., 2025) | Unified grounding of DNA, RNA, and protein sequences for language modeling | Nucleic Acid LLM | Sequence concatenation fusion |
| RNA-GPT (Xiao et al., 2024b) | Instruction-tuned RNA sequence modeling without explicit structural input | Nucleic Acid LLM | Sequence only |
| LLM3-DTI (Zhang et al., 2025b) | Multimodal modeling of topology graphs and text for drug–target interaction | Embedding Model | Embedding only |
| DrugAgent (Inoue et al., 2025) | Multi-agent LLM framework for drug–target interaction with planning and tool use | Agent System | LLM as planner and evaluator |
| MatterChat (Tang et al., 2025) | Atom resolved structure grounded question answering for materials science | Materials LLM | Fixed length Q-Former style connector |

et al., 2022), ProtST (Xu et al., 2023b), RNA-GPT (Xiao et al., 2024b), ChatNT (de Almeida et al., 2025)).

**Fixed-capacity connectors induce structural bottlenecks.** Projection fusion and Q-Former-style connectors compress variable-size structures into a constant token budget (InstructMol (Cao et al., 2025), Chem3DLLM (Jiang et al., 2025), Prot2Chat (Wang et al., 2025b), MatterChat (Tang et al., 2025)). This design choice constrains scalability with entity size and atom-level detail.

**Tokenization bridges mitigate but do not remove capacity mismatch.** Cross-attention tokenization, exemplified by Graph2Token (Wang et al., 2025a), reduces the gap between structural embeddings and language tokens, but it remains bounded by a fixed-length interface under increasing structural complexity.

These limitations motivate all-atom language grounding mechanisms that preserve fine geometry while allocating token budget adaptively with structural complexity, consistent with the framing in the main related-work section. Note that Agent works such as DrugAgent (Inoue et al., 2025), LLMs usually behave as a judge or planner, which is not in our context.

### B.5. Connections to Graph Domain Adaptation, Prior Work, and Generative Modeling

**Graph learning and cross-distribution alignment.** Graph representation and domain adaptation research provides foundational insights for Cuttlefish's connector design. Structure-preserving graph learning (Fang et al., 2022) established that topology-aware encoding respects structural invariants, a property that carries forward into domain adaptation, where attribute-driven (Fang et al., 2025b), homophily-enhanced (Fang et al., 2025c), homophily-agnostic (Fang et al., 2026), and multiview structural aggregation (Fang et al., 2025d) methods address cross-distribution alignment under structural shift. These works motivate preserving instruction-relevant structural signals across the graph-to-language modality boundary.

**Fine-grained molecular graph representation.** Fine-grained molecular representation is a prerequisite for structure-grounded reasoning. MolGraph-xLSTM (Sun et al., 2025b) demonstrates that multi-scale graph encoding with mixture-of-experts attention yields interpretable, granular representations, motivating the patch-based aggregation in Cuttlefish rather than flat pooling.

**Relation to our prior molecular LLM work.** Cuttlefish extends our prior entropy-guided molecular LLM (Jing et al., 2026), which used a Q-Former-style fixed-length connector for single-modality molecular understanding. The present work replaces the fixed anchor with an instruction-conditioned gate that expands dynamically into soft patches, enabling budget scaling across modalities from small molecules to large nucleic acids. Fine-tuning the LLM is also necessary here: 3D Cartesian coordinates lie outside LLM pre-training distributions, and no connector design alone can substitute for exposure to coordinate-grounded supervision across the full range of all-atom entity sizes.

**Efficiency in multimodal LLM reasoning.** Efficiency is a first-class concern when incorporating additional modalities into LLMs. Dynamic parallel tree search (Ding et al., 2025) demonstrates that structured inference-time computation can dramatically reduce reasoning cost without sacrificing output quality, and difficulty-aware knowledge distillation (He et al., 2025) shows that model compression guided by sample difficulty yields compact yet capable models. Both works underscore that unchecked complexity growth — whether from search overhead or model size — limits practical deployment. This motivates the adaptive token budget in Cuttlefish: if structural token counts scaled linearly with entity size, the added modality would impose prohibitive context costs, making the system impractical for large biomolecules. Keeping the token budget sublinear preserves the efficiency that makes multimodal LLM reasoning viable at scale.

**Generative AI for molecules and proteins as experimental context.** The survey of generative AI for molecular and protein design (Sun et al., 2025a) contextualizes the interdisciplinary scope of Cuttlefish's evaluation. Molecules and proteins together constitute a natural testbed for structure-conditioned language understanding, and the connector advances demonstrated here are intended to generalize to broader domain-specific scientific modeling settings.

## C. Implementation Details

This section details Cuttlefish implementation, covering architecture and optimization hyperparameters, modality-specific atom-level preprocessing with aligned 3D coordinates plus filtering and sequence-only fallbacks on rare structure failures, and masked reasoning augmentation that injects teacher-generated latent deliberation spans while applying loss only to final-answer tokens.

### C.1. Hyperparameter and Settings

**Model Configuration.** Table 14 summarizes the core architecture hyperparameters of Cuttlefish, covering the graph encoder, the instruction-conditioned patching/assignment module, and the cross-attention fusion with the frozen LLM. The graph encoder hidden size and graph encoder depth specify the representation width and number of message-passing (equivariant)

*Table 14.* Hyperparameters and configuration settings for Cuttlefish. The parameters specify the architectural dimensions for the graph encoder, the multimodal fusion blocks, and the spatial logic for the all-atom anchor assignment mechanism.

| Parameter | Value | Parameter | Value | Parameter | Value |
|---|---|---|---|---|---|
| Graph encoder hidden size | 256 | Graph encoder depth | 8 | Graph encoder dropout | 0.1 |
| Coordinate updates | Enabled | Encoder LayerNorm | Enabled | Number of RBF bases | 32 |
| RBF cutoff distance | 10.0 | Fusion block count | 8 | Attention head count | 32 |
| Fusion model width | 4096 | Fusion MLP intermediate size | 16384 | Fusion dropout | 0.1 |
| Max anchors per graph | 2048 | Max nodes per anchor | None (soft pooling) | Mass-based anchor fraction | 0.1 |
| Assignment distance scale | 1.0 | Assignment temperature | 0.1 | Gate MLP hidden size | 256 |

*Table 15.* Summary of training parameters used for modality alignment tuning. This includes the learning rate schedules, gradient accumulation strategies, etc., employed to optimize the multimodal integration between structural representations and the language model.

| Parameter | Value | Parameter | Value | Parameter | Value |
|---|---|---|---|---|---|
| Base language model checkpoint | Llama-3.1-8B-Instruct | Optimizer | AdamW | Learning rate | 1e-4 |
| Training epochs | 4 | Per-device train batch size | 1 | Gradient accumulation steps | 16 |
| Warmup steps | 100 | Gradient clipping (max norm) | 1.0 | Weight decay | 0.01 |
| Adam beta1 | 0.9 | Adam beta2 | 0.999 | Adam epsilon | 1e-8 |
| Mixed precision (bf16) | Enabled | Gradient checkpointing | Enabled | Evaluation split ratio | 0.1 |
| Evaluation frequency (steps) | 100 | Logging frequency (steps) | 10 | DataLoader workers | 8 |

layers used to embed molecular/protein/nucleic-acid structures, with graph encoder dropout controlling regularization within this stack. Coordinate updates indicate whether the encoder performs equivariant coordinate refinement alongside feature updates, and encoder LayerNorm denotes the use of normalization for stabilizing deep message passing. The encoder's geometric featurization is parameterized by the number of Radial Basis Function (RBF) bases and RBF cutoff distance, which define the resolution and range of radial basis embeddings for interatomic distances. For multimodal integration, fusion block count, attention head count, fusion model width, and fusion MLP intermediate size define the capacity of the cross-attention transformer blocks that align graph-derived tokens with the LLM embedding space, with fusion dropout providing additional regularization. Finally, the patching/assignment module is governed by max anchors per graph and max nodes per anchor, which control how graph nodes are grouped into anchor-based patches (with "None" indicating soft pooling without a hard cap), and mass-based anchor fraction, assignment distance scale, and assignment temperature, which shape the softness and selectivity of node-to-anchor assignments. Gate MLP hidden size specifies the width of the gating network that modulates connector behavior, enabling adaptive routing of structural information into the fusion pathway.

**Training Configuration.** Table 15 reports the main optimization and data-loading hyperparameters used in our two-stage tuning pipeline. In the modality alignment stage, we train the multimodal connector components with AdamW using a base learning rate of $1 \times 10^{-4}$, warmup, gradient accumulation, and gradient clipping for stability, together with bf16 and gradient checkpointing for memory efficiency. In the subsequent LLM adaptation stage, we keep the overall setup the same for fair comparison, but use a smaller learning rate ($1 \times 10^{-5}$) than modality alignment to safely adapt the language model while avoiding destabilizing previously aligned multimodal representations.

### C.2. Data Process

**Structure generation and atom-level extraction.** All modalities output (i) an atom-ordered node feature matrix and (ii) an aligned coordinate matrix $\mathbf{P} \in \mathbb{R}^{N \times 3}$ in Å, with strict row-wise consistency between atom typing, graph construction, and Cartesian positions. Fig. 11 illustrates the structure generation flow, described in the following paragraphs:

**Molecules (SMILES → heavy atoms + 3D).** SMILES parsed into an RDKit (Landrum, 2013) "`Mol`"; atom types defined as chemical element symbols in RDKit internal atom ordering. 3D geometry construction: (i) explicit hydrogen addition for embedding robustness, (ii) one or multiple conformer candidates via RDKit distance-geometry embedding (ETKDG), (iii) MMFF force-field refinement, best-conformer selection (typically lowest-energy converged). Post-selection hydrogen removal; returned $\mathbf{P} \in \mathbb{R}^{N \times 3}$ corresponds to the heavy-atom set and preserves the same atom order used for typing and graph building.

**Proteins (residue sequence → matched PDB/mmCIF → atoms + 3D).** Residue sequence linked to an experimental structure (e.g., UniProt↔PDB mapping or sequence search); corresponding mmCIF processed with Gemmi. Polymer chains and residues iterated; per-atom records extracted in parser-defined order. Atom typing encodes (1) atomic number from the

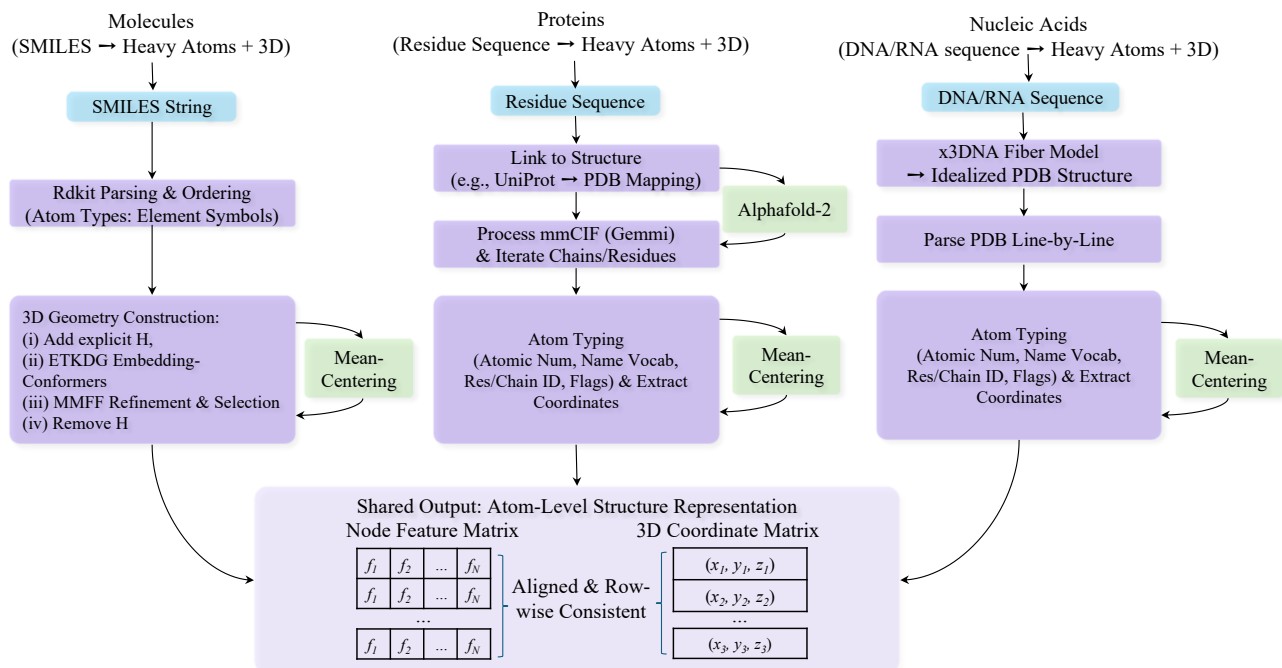

*Figure 11.* Multimodal structure generation and atom-level extraction. Three specialized streams process molecules (SMILES), proteins (residue sequences), and nucleic acids into a unified format. The pipeline ensures strict row-wise consistency between the atom-ordered node feature matrix and the aligned 3D coordinate matrix.

element field and (2) atom-name vocabulary (e.g., N, CA, C, O, sidechain heavy atoms; unknown→"misc"), augmented by parent residue identity (3-letter residue vocabulary), chain index, residue index, and binary flags (e.g., is_backbone, is_CA). Coordinates taken directly from mmCIF atom positions (Å); optional mean-centering by subtracting the global coordinate mean. Feature rows and $\mathbf{P}$ share identical atom ordering.

*Table 16.* Summary of proposed dataset GEO-AT: sources and counts by training stages and four modalities.

| Stage | Modality | Source | Count |
|---|---|---|---|
| Encoder | Molecule | PubChem | 1M |
| | Protein | UniProt | 300K |
| | DNA | DNA-Chat | 300K |
| | RNA | RNA-QA | 200K |
| End-to-End | Molecule | MolLlama-Instruct | 10K |
| | Protein | UniProt+PDB | 10K |
| | DNA | DNA-Chat | 5K |
| | RNA | RNA-QA | 5K |

**Nucleic acids (DNA/RNA sequence → X3DNA fiber PDB → atoms + 3D).** Sequence deterministically mapped to an idealized 3D structure using X3DNA fiber, written as PDB, then parsed line-by-line to collect atom metadata (atom name, residue name, chain id, residue id, element) and Cartesian coordinates. Atom typing uses a nucleic-acid-specific schema: atomic number (from element, with fallback inference from atom name), atom-name vocabulary spanning sugar–phosphate and base atoms (e.g., P/OP1/OP2/O5′... plus base atoms), nucleotide identity vocabulary (DNA: A/C/G/T; RNA: A/C/G/U), chain and residue indices, and binary backbone and phosphate-group indicators. RNA follows the same pipeline, with alphabet $\{A, C, G, U\}$ and optional single-strand generation (default). Output coordinates $\mathbf{P} \in \mathbb{R}^{N \times 3}$ mean-centered by subtracting the coordinate mean; one-to-one alignment maintained between feature rows and coordinate rows.

**Data filtering and fallbacks.** Entities with invalid primary inputs were removed, including SMILES strings not parsable by RDKit (Landrum, 2013) and sequences failing modality-specific parsing. When sequence inputs were valid but 3D structure generation failed, samples were retained as sequence-only inputs to preserve coverage and ensure consistent, fair evaluation across models. This fallback occurred in fewer than 0.5% of cases and did not materially affect reported conclusions, including claims about structure-conditioned reasoning.

### C.3. Encoder Pretraining Data

Following MuMo (Jing et al., 2025), samples are limited to the hundred-thousand scale, which suffices for representation convergence and feature alignment. For molecules, 1M samples from PubChem (Kim et al., 2023) are filtered by sequence length to cover structural complexity. 300K proteins from UniProt (Consortium, 2019) across key model organisms (Human, Mouse, Rat, A. thaliana, S. cerevisiae) are sampled. We include sequences with multiple conformers to capture structural flexibility. For nucleic acids, 300K DNA and 200K RNA samples from DNA-Chat (de Almeida et al., 2025) and RNA-QA (Xiao et al., 2024b). Data spans 30 functional categories across human and mouse genomes to ensure biological significance.

### C.4. Masked Reasoning Augmentation for Improved Latent Deliberation

**Motivation.** Instruction tuning corpora for scientific QA and structured reasoning often provide only a final response label, without explicit intermediate rationales. Directly optimizing for short answer templates (for example, "The answer is A") can bias training toward format memorization, reduce linguistic diversity, and degrade latent deliberation capacity. To mitigate this issue while avoiding the need for human reasoning annotations, masked reasoning augmentation is adopted.

**Construction.** As shown in Algo. 3, for each training instance with instruction $x$ and gold final answer $a$, a teacher model (OpenAI GPT-5) produces a harmless and general reasoning span $\tilde{r}$. The span length is variable and may include placeholder style statements that indicate reasoning progress without encoding task-specific private labels. The final training target sequence is constructed as

$$y = \big[\langle\texttt{think}\rangle, \tilde{r}, \langle/\texttt{think}\rangle, a\big], \tag{7}$$

where $\langle\texttt{think}\rangle$ and $\langle/\texttt{think}\rangle$ are retained to delimit the latent deliberation region.

**Masked objective.** Let $y_{1:T}$ be the tokenization of $y$, and let $m_t \in \{0, 1\}$ denote a loss mask. The mask is set to exclude all tokens inside the reasoning span, while retaining supervision on the final answer tokens (and optionally the delimiter tokens):

$$m_t = \mathbb{1}[y_t \in a] \quad \text{(optionally } m_t \leftarrow 1 \text{ for delimiter tokens)}. \tag{8}$$

Training minimizes a masked negative log likelihood:

$$\mathcal{L}(\theta) = -\mathbb{E}_{(x,a)\sim\mathcal{D}} \left[ \sum_{t=1}^{T} m_t \log p_\theta(y_t \mid x, y_{<t}) \right]. \tag{9}$$

Since the teacher forces conditions on the entire prefix $y_{<t}$, the model can attend to $\tilde{r}$ as privileged context during optimization, while gradients are applied only to the answer prediction. This reduces incentives to overfit to short answer templates and preserves expressive capacity, because the model is not penalized for mismatching the teacher-generated reasoning text.

**Usage at inference.** At inference time, decoding can either (i) omit the think region by prompting for answers only, or (ii) allow a think delimited region for internal deliberation, depending on the evaluation protocol. The core training mechanism remains the same, namely, reasoning tokens are never used as direct supervision targets.

**Why Masked Reasoning Augmentation Preserves Reasoning Ability.** We justify that masked reasoning augmentation (Sec. C.4) is a rational and effective surrogate for missing chain-of-thought (CoT) supervision: it avoids the degeneracy induced by short-answer-only tuning while not forcing imitation of teacher rationales, thereby preserving latent deliberation capacity.

**Setup.** Let $p_0$ denote the pre-trained (or instruction-tuned) initialization and $p_\theta$ the fine-tuned model. For each example $(x, a) \sim \mathcal{D}$, construct $y = [r, a]$ where $r = \langle\texttt{think}\rangle \tilde{r} \langle/\texttt{think}\rangle$ and $a$ is the gold answer (Eq. 7). Let $A$ be the set of answer-token positions, and $R$ the set of reasoning-token positions in the tokenization $y_{1:T}$. The training loss is the masked NLL (Eq. 9)

$$\mathcal{L}(\theta) = -\mathbb{E}_{(x,a)} \left[ \sum_{t\in A} \log p_\theta(y_t \mid x, y_{<t}) \right], \tag{10}$$

(optionally adding delimiter tokens to $A$ does not change the arguments below).

---

**Algorithm 3** Masked Reasoning Augmentation

---

**Input:** Dataset $\mathcal{D} = \{(x^{(i)}, a^{(i)})\}_{i=1}^{N}$, teacher generator $\mathcal{G}$ (GPT-5), model $p_\theta$, delimiter tokens $\langle\texttt{think}\rangle, \langle\texttt{/think}\rangle$
**Output:** Updated parameters $\theta$
 1: **for** each minibatch $\mathcal{B} \subset \mathcal{D}$ **do**
 2:     **for** each $(x, a) \in \mathcal{B}$ **do**
 3:        $\tilde{r} \leftarrow \mathcal{G}(x)$                                   ▷ *harmless, general, variable length, may include placeholders*
 4:        $y \leftarrow [\langle\texttt{think}\rangle, \tilde{r}, \langle\texttt{/think}\rangle, a]$
 5:        Tokenize $y \mapsto (y_1, \ldots, y_T)$
 6:        Build mask $m_{1:T}$ where $m_t = 0$ for tokens in $\tilde{r}$ and $m_t = 1$ for tokens in $a$ ▷ *optionally include delimiter tokens*
 7:     **end for**
 8:     $\mathcal{L} \leftarrow -\sum_{(x,a)\in\mathcal{B}} \sum_{t=1}^{T} m_t \log p_\theta(y_t \mid x, y_{<t})$
 9:     $\theta \leftarrow \theta - \eta\nabla_\theta\mathcal{L}$
10: **end for**

---

**Claim 1 (Masked augmentation reduces template overfitting).** Assume a decoder factorization $p_\theta(y \mid x) = \prod_{t=1}^{T} p_\theta(y_t \mid x, y_{<t})$. Consider the answer-only tuning objective without any reasoning prefix:

$$\mathcal{L}_{\text{ans-only}}(\theta) = -\mathbb{E}_{(x,a)}\Big[ \log p_\theta(a \mid x) \Big], \tag{11}$$

where the model learns $p_\theta(a \mid x)$ under a short and highly repetitive target format. If the target answers share a narrow template family (e.g., "The answer is A"), then $\mathcal{L}_{\text{ans-only}}$ admits low-loss solutions that allocate probability mass to a small set of high-frequency surface forms, yielding a biased decoder that collapses stylistic and intermediate-generation degrees of freedom.

Under masked augmentation, the supervised tokens $t \in A$ are predicted conditioned on a longer, variable prefix $r$:

$$p_\theta(a \mid x, r) = \prod_{t\in A} p_\theta(y_t \mid x, y_{<t}), \qquad y_{<t} \supseteq r. \tag{12}$$

Since $r$ is variable-length and lexically diverse, the conditional context distribution seen during training has higher entropy than in Eq. 11. This breaks the spurious shortcut of mapping $x \mapsto$ a fixed answer template independent of preceding generation state, because the answer head must remain stable across a family of diverse prefixes. Operationally, the model is optimized to output the same $a$ across many distinct $r$-contexts, which regularizes against brittle template memorization and maintains flexibility in generation dynamics before the answer.

**Claim 2 (No-imitation property: the method does not teach or enforce teacher CoT).** By construction, the mask sets $m_t = 0$ for all $t \in R$ (Eq. 8), hence the gradient decomposes as

$$\nabla_\theta\mathcal{L}(\theta) = -\mathbb{E}_{(x,a)}\left[\sum_{t\in A} \nabla_\theta \log p_\theta(y_t \mid x, y_{<t})\right], \qquad \frac{\partial\mathcal{L}}{\partial \log p_\theta(y_t \mid \cdot)} = 0 \;\; \forall t \in R. \tag{13}$$

Therefore, masked augmentation cannot directly push $p_\theta$ to reproduce $\tilde{r}$ or match the teacher's reasoning distribution; it only uses $\tilde{r}$ as conditioning noise/augmentation for predicting $a$. This avoids two common failure modes of distillation-style CoT training: (i) copying teacher-specific artifacts and (ii) penalizing valid alternative reasoning trajectories.

**Claim 3 (Latent-deliberation preservation via prefix invariance).** Define the desired robustness property: for a given instruction $x$, correct answering should be invariant to benign internal deliberation text $r$ that does not leak the label. Formally, for a distribution $\mathcal{R}(x)$ over harmless reasoning prefixes, we want

$$p_\theta(a \mid x, r) \approx p_\theta(a \mid x, r'), \qquad r, r' \sim \mathcal{R}(x), \tag{14}$$

while still allowing $r$ to carry computational substrate during generation. Masked augmentation is exactly empirical risk minimization over random $r$-contexts:

$$\min_\theta \;\; \mathbb{E}_{(x,a)} \, \mathbb{E}_{r\sim\mathcal{R}(x)}\Big[ -\log p_\theta(a \mid x, r) \Big]. \tag{15}$$

Optimizing Eq. 15 encourages $p_\theta$ to be stable under variations of the pre-answer hidden-state trajectory induced by diverse prefixes. Intuitively, the model is trained to remain competent after spending tokens in a think region, rather than being trained to emit the answer immediately. This directly targets the failure case of answer-only tuning, where decoding before the answer becomes a low-utility region and the model learns to shortcut.

**Claim 4 (Connection to dropout-style regularization over prefixes).** Masked augmentation can be viewed as a structured data augmentation on the conditioning context: $r$ is a sampled perturbation of the prefix distribution that leaves the label $a$ unchanged. As in classic augmentation, the supervised signal forces the predictor to rely on task-relevant information (the instruction $x$ and structural tokens) rather than fragile surface regularities of a fixed answer template. Unlike standard dropout, this perturbation operates at the sequence level and explicitly exercises the model's long-context attention and state evolution before producing $a$.

**Practical implication.** Claims 1–4 imply the method is rational under missing-CoT conditions: it (i) prevents collapse to short answer templates by training under diverse pre-answer prefixes, (ii) preserves freedom of internal reasoning because no gradients are applied to $\tilde{r}$, and (iii) encourages answer stability after variable-length latent deliberation, which is the minimal property needed to maintain reasoning-like behavior when explicit reasoning corpora are unavailable.

# D. Experiment Details

## D.1. Datasets and Benchmarks

**Mol-Instructions** (molecule and protein) (Fang et al., 2023). Mol-Instructions constructs instruction tuning data by aggregating licensed biomolecular sources and converting them into a unified instruction format, with additional task description diversification via human-written seeds expanded by an LLM and then manually reviewed, followed by explicit quality control to ensure reliability. It is organized into **three components:** molecule-oriented instructions covering molecular captioning, reactions, and design with 148.4K instructions across six tasks, protein-oriented instructions spanning five task categories for structure, function, activity, and design with 505K instructions, and biomolecular text instructions for information extraction and QA style NLP tasks with 53K instructions.

**DNA-Chat** (de Almeida et al., 2025). ChatNT converts existing DNA sequence benchmarks into instruction data by curating many English question answer templates per task and sampling a template per input sequence. For the Nucleotide Transformer benchmark, it converts 18 binary or multilabel classification datasets into diverse instructions while keeping the original train-test splits. Data correctness and evaluation consistency are supported by retaining original dataset splits, using different questions in train versus test to probe English generalization, and reporting each task metric as defined in the corresponding benchmark, typically on up to 5,000 sampled test instances per task. The resulting instructions corpus is large, reported as 605M DNA tokens and 273M English tokens, and model training over the 27 task set is described as 7.8M samples over 2B tokens.

**RNA-QA** (Xiao et al., 2024b). RNA-QA is constructed from RNAcentral (Bateman et al., 2011) by filtering RNAs with associated literature and then scraping and summarizing relevant papers to produce per RNA annotations, with a divide and conquer summarization pipeline that applies topic modeling to group papers and then summarizes each group before a final aggregation summary. The paper reports an initial filter of roughly 420,000 literature-linked RNAs, refined to 407,616 RNAs, with sequences truncated or filtered to a length of at most 1024 nucleotides to fit the sequence encoder input limit, and each RNA is paired with an abstract style description. For instruction tuning, the dataset decomposes these annotations into more targeted QA pairs using an LLM, producing multiple QA pairs per RNA and explicitly noting 5 to 14 QA pairs per RNA in the collection pipeline description.

**GEO-AT.** GEO-AT is a unified all-atom instruction dataset constructed for connector alignment and structure conditioned LLM adaptation under end-to-end finetuning, prioritizing compactness and high supervision density. It aggregates three modalities: (i) molecules, using 10K chemical to text captioning pairs from Mol-Llama-Instruct (Kim et al., 2025); (ii) proteins, using 10K high confidence UniProt (Consortium, 2019) entries (annotation score 5/5) with verified PDB (Burley et al., 2017) 3D structures, excluding multi conformer sequences to preserve unambiguous structural supervision; and (iii) nucleic acids, using 10K instruction following samples from ChatNT (5K DNA, 5K RNA) (de Almeida et al., 2025). For instances lacking experimentally resolved 3D structures, geometries are generated via the pipeline in App. C.2. Each modality reserves 200 held-out test samples, with the remaining examples used for training. All the baselines for this benchmark are tuned on this dataset. Table 16 summarizes data sources and counts across training stages and modalities.

## D.2. Baselines

Baselines include general instruction-tuned LLMs, domain-adapted chemistry and biomolecule LLMs, and explicit reasoning variants. General chat and backbone references include Alpaca (Taori et al., 2023), Baize (Xu et al., 2023a), ChatGLM (GLM et al., 2024), LLaMA-2 (Touvron et al., 2023), Vicuna-v1.5 (Zheng et al., 2023), Qwen2.5 (Bai et al., 2025), Llama-3 (Grattafiori et al., 2024), Mistral-3 (Liu et al., 2026), GLM-4-9B (GLM et al., 2024), and Qwen3-8B (Yang et al., 2025a), providing broad instruction-following capacity without structural specialization. Science and chemistry-oriented baselines include Galactica-6.7B (Taylor et al., 2022), Text+Chem T5 (Christofidellis et al., 2023), and MolT5 (Pei et al., 2024), which incorporate scientific or molecular pretraining objectives to improve chemistry-text alignment. Biomolecular instruction baselines include Mol-Instructions (Fang et al., 2023), which fine-tune Llama backbones on Mol-Instructions datasets, and Mol-LLama (Kim et al., 2025), which further targets molecule-centric alignment. Reasoning-focused general baselines include DeepSeek-R1-D-Qwen-7B (Guo et al., 2025a) and Mistral-3-8B-Reasoning (Liu et al., 2026), used to assess gains attributable to explicit reasoning post-training rather than modality grounding. For protein-specific models, ProtLLM (Zhuo et al., 2024), ProteinGPT (Xiao et al., 2024a), and ProtChatGPT (Wang et al., 2024) are selected.

**Rationale of Selection.** Baselines are selected to isolate gains from structure grounding under realistic compute and model-size constraints. General LLMs are restricted to recent, open-source instruction-tuned backbones, covering both reasoning and non-reasoning variants, with parameters largely aligned to the 6–8B regime to match the available training and evaluation budget. Modality-specific baselines prioritize strong, recent domain models that natively accept molecular or biomolecular structural inputs or molecule-aware tokenizations, and that report competitive performance on closely related instruction or QA benchmarks, enabling a fair comparison between sequence-only adaptation and explicit structure-conditioned modeling.

## D.3. Benchmark Metrics

**Text generation quality.** For free-form descriptions and open-ended instruction responses, we report n-gram overlap, sequence-level recall, and semantic matching metrics. **BLEU-2/4** ($\uparrow$) quantify modified n-gram precision with a brevity penalty. **ROUGE-1/2/L** ($\uparrow$) measure unigram/bigram overlap and longest-common-subsequence (LCS) similarity, capturing content coverage and ordering robustness. **METEOR** ($\uparrow$) emphasizes alignment with stemming/synonymy and includes a fragmentation penalty, improving sensitivity to paraphrases. For embedding-level semantic similarity, we report **BERTScore** ($\uparrow$), computed via token embedding cosine matches between hypothesis and reference:

$$\text{BERTScore}(y, \hat{y}) \;=\; \frac{1}{|y|} \sum_{t \in y} \max_{s \in \hat{y}} \cos(\mathbf{e}_t, \mathbf{e}_s), \tag{16}$$

with precision/recall/F1 variants reported per benchmark convention. We use these metrics when multiple valid phrasings exist and surface overlap and semantic faithfulness jointly.

**Regression error.** For scalar property prediction, we use **MAE** ($\downarrow$), the mean absolute error:

$$\text{MAE} \;=\; \frac{1}{N} \sum_{i=1}^{N} |\hat{z}_i - z_i|, \tag{17}$$

where $z_i$ and $\hat{z}_i$ denote the ground-truth and predicted targets.

**Classification.** For DNA tasks with class imbalance, we report **Matthews correlation coefficient (MCC)** ($\uparrow$), which correlates predictions and labels and remains informative under skewed marginals:

$$\text{MCC} \;=\; \frac{TP \cdot TN - FP \cdot FN}{\sqrt{(TP + FP)(TP + FN)(TN + FP)(TN + FN)}}. \tag{18}$$

**Exact-match and edit distance.** For structured-string outputs (e.g., canonical answers or sequence-like targets), we report **Exact** ($\uparrow$) accuracy and character-level **Levenshtein** distance ($\downarrow$):

$$\text{Exact} \;=\; \mathbb{1}[\hat{y} = y], \qquad \text{Lev}(y, \hat{y}) \;=\; \min_{\pi \in \mathcal{E}(y \to \hat{y})} |\pi|, \tag{19}$$

where $\mathcal{E}$ denotes the set of edit sequences (insert/delete/substitute) transforming $y$ into $\hat{y}$.

*Table 17.* Detailed results on molecular captioning and property prediction tasks from the Mol-Instruction benchmark. Official splits are used for testing Cuttlefish, Qwen, and Mol-Llama models; others are taken from the benchmark leaderboard.

| Model | Molecular Description | | | | | | Property |
| --- | --- | --- | --- | --- | --- | --- | --- |
| | BLEU-2↑ | BLEU-4↑ | ROUGE-1↑ | ROUGE-2↑ | ROUGE-L↑ | METEOR↑ | MAE↓ |
| Alpaca-7B | 0.068 | 0.014 | 0.178 | 0.041 | 0.136 | 0.107 | 322.109 |
| Baize-7B | 0.064 | 0.015 | 0.189 | 0.053 | 0.148 | 0.106 | 261.343 |
| LLaMA-2-7B | 0.059 | 0.014 | 0.164 | 0.066 | 0.148 | 0.184 | 5.553 |
| Vicuna-v1.5-13B | 0.052 | 0.011 | 0.151 | 0.055 | 0.130 | 0.168 | 860.051 |
| Galactica-6.7B | 0.024 | 0.008 | 0.074 | 0.015 | 0.063 | 0.065 | 0.568 |
| Qwen2.5-7B | 0.435 | 0.356 | 0.581 | 0.491 | 0.572 | 0.538 | 0.182 |
| Mol-Ins.-Llama-2-7B | 0.217 | 0.143 | 0.337 | 0.196 | 0.291 | 0.291 | 0.013 |
| Mol-Ins.-Llama-3.1-8.3B | 0.419 | 0.361 | 0.719 | 0.646 | 0.709 | 0.637 | 15.059 |
| Mol-LLaMA-2-7B | 0.478 | 0.425 | 0.761 | 0.698 | 0.750 | 0.701 | 0.0035 |
| Mol-LLaMA-3.1-8.3B | 0.476 | 0.426 | 0.767 | 0.708 | 0.759 | 0.707 | 0.0039 |
| Cuttlefish-8.3B | 0.481 | 0.442 | 0.785 | 0.681 | 0.766 | 0.715 | 0.0030 |

**Molecule validity and similarity (structure-aware).** For molecule generation, we evaluate both chemical validity and structure similarity. **Validity** (↑) is the fraction of outputs parsable into valid molecules (e.g., via RDKit). To quantify similarity between generated and reference molecules, we compute fingerprint Tanimoto similarity (FTS, ↑) using multiple representations, including **RDKit** fingerprints, **MACCS** keys, and **Morgan** fingerprints:

$$\mathrm{FTS}(m, \hat{m}) \;=\; \mathrm{Tan}\big(\phi(m), \phi(\hat{m})\big) \;=\; \frac{\langle \phi(m), \phi(\hat{m}) \rangle}{\|\phi(m)\|_1 + \|\phi(\hat{m})\|_1 - \langle \phi(m), \phi(\hat{m}) \rangle}, \tag{20}$$

where $\phi(\cdot) \in \{0,1\}^d$ is a binary fingerprint and $\mathrm{Tan}$ denotes Tanimoto similarity. We additionally report **BLEU** (↑) and **Levenshtein** (↓) for string-level similarity when the target is provided as a SMILES-like representation, but prioritize validity and fingerprint similarity to reflect chemically meaningful agreement.

### D.4. Detailed Benchmark Results

This section reports the full Mol-Instructions (Fang et al., 2023) benchmark results. The main paper Tab. 3 is restricted to at most two metrics per task for layout compactness, whereas the appendix exposes the complete metric suite. Across tasks, Cuttlefish attains top rank on essentially all reported metrics, indicating consistent gains rather than metric-specific overfitting.

**Molecular description and property** (Tab. 17). Cuttlefish achieves best performance across the lexical overlap metrics for molecular captioning and simultaneously yields the lowest error on property regression, indicating joint improvements in structure-conditioned text and property grounding. The only non-leading captioning metric is a mid-order ROUGE variant, where Cuttlefish remains competitive.

**Description-guided molecule design** (Tab. 18). Cuttlefish dominates generation quality and edit-distance metrics, and also leads across multiple fingerprint similarity measures, suggesting improved alignment between instruction semantics and generated molecular structure. Validity is saturated at the top tier, indicating that gains are driven by chemical plausibility and target matching rather than trivial validity filtering. The only top-2 outcome occurs on one fingerprint similarity, plausibly reflecting descriptor-specific bias toward particular scaffold statistics.

**Forward reaction prediction** (Tab. 19). Cuttlefish attains the best exact match and overall generation quality, while also leading on structure-based similarity and maintaining perfect validity, supporting improved mechanistic consistency in reaction outcome generation.

**Reagent prediction** (Tab. 20). Cuttlefish ranks top-1 on exact match and all structure-based similarity metrics while maintaining perfect validity, indicating superior condition inference and chemically consistent reagent set generation. A single text-overlap metric is top-2, consistent with synonymy and ordering non-identifiability in reagent lists that weakly correlate with correctness under n-gram scoring.

**Retrosynthesis** (Tab. 21). Cuttlefish achieves the best exact match and fingerprint similarities with saturated validity, suggesting improved decomposition of target products into plausible precursors rather than purely string-level matching. BLEU is top-2, plausibly driven by multiple valid precursor sets and canonicalization variability that penalize lexical overlap

*Table 18.* Detailed results on the description-guided molecule design task from the Mol-Instructions benchmark. Official splits are used for testing Cuttlefish, Qwen, and Mol-LLaMA models; other baselines are taken from the benchmark leaderboard.

| Model | Exact↑ | BLEU↑ | Levenshtein↓ | RDKit FTS↑ | MACC FTS↑ | Morgan FTS↑ | Validity↑ |
|---|---|---|---|---|---|---|---|
| Alpaca-7B | 0.000 | 0.004 | 51.088 | 0.006 | 0.029 | 0.000 | 0.002 |
| Baize-7B | 0.000 | 0.006 | 53.796 | 0.000 | 0.000 | 0.000 | 0.002 |
| ChatGLM-6B | 0.000 | 0.004 | 53.157 | 0.005 | 0.000 | 0.000 | 0.005 |
| LLaMA-2-7B | 0.000 | 0.003 | 59.864 | 0.005 | 0.000 | 0.000 | 0.003 |
| Vicuna-v1.5-13B | 0.000 | 0.006 | 60.356 | 0.006 | 0.001 | 0.000 | 0.001 |
| Galactica-6.7B | 0.000 | 0.192 | 44.152 | 0.135 | 0.238 | 0.088 | 0.992 |
| Text+Chem T5 | 0.097 | 0.508 | 41.819 | 0.352 | 0.474 | 0.353 | 0.721 |
| MolT5 | 0.112 | 0.546 | 38.276 | 0.400 | 0.538 | 0.295 | 0.773 |
| Mol-Ins.-Llama-2-7B | 0.002 | 0.345 | 41.367 | 0.231 | 0.412 | 0.147 | 1.000 |
| Mol-Ins.-Llama-3.1 | 0.025 | 0.521 | 38.742 | 0.358 | 0.520 | 0.221 | 1.000 |
| Mol-LLaMA-3.1-8.3B | 0.012 | 0.638 | 18.917 | 0.392 | 0.534 | 0.220 | 0.876 |
| Qwen2.5-7B | 0.001 | 0.434 | 306.144 | 0.207 | 0.348 | 0.121 | 0.772 |
| Cuttlefish-8.3B | 0.019 | 0.668 | 16.029 | 0.422 | 0.583 | 0.287 | 1.000 |

*Table 19.* Detailed results on the forward reaction prediction task from the Mol-Instructions benchmark. Official splits are used for testing Cuttlefish, Qwen, and Mol-LLaMA models; other baselines are taken from the benchmark leaderboard.

| Model | Exact↑ | BLEU↑ | Levenshtein↓ | RDKit FTS↑ | MACC FTS↑ | Morgan FTS↑ | Validity↑ |
|---|---|---|---|---|---|---|---|
| Alpaca-7B | 0.000 | 0.065 | 41.989 | 0.004 | 0.024 | 0.008 | 0.138 |
| Baize-7B | 0.000 | 0.044 | 41.500 | 0.004 | 0.025 | 0.009 | 0.097 |
| ChatGLM-6B | 0.000 | 0.183 | 40.008 | 0.050 | 0.100 | 0.044 | 0.108 |
| LLaMA-2-7B | 0.000 | 0.020 | 42.002 | 0.001 | 0.002 | 0.001 | 0.039 |
| Vicuna-v1.5-13B | 0.000 | 0.057 | 41.690 | 0.007 | 0.016 | 0.006 | 0.059 |
| Galactica-6.7B | 0.000 | 0.468 | 35.021 | 0.156 | 0.257 | 0.097 | 0.946 |
| Text+Chem T5 | 0.239 | 0.782 | 20.413 | 0.705 | 0.789 | 0.652 | 0.762 |
| Mol-Ins.-Llama-2-7B | 0.045 | 0.654 | 27.262 | 0.313 | 0.509 | 0.262 | 1.000 |
| Mol-Ins.-Llama-3.1 | 0.503 | 0.883 | 13.410 | 0.756 | 0.863 | 0.708 | 1.000 |
| Mol-LLaMA-3.1-8.3B | 0.440 | 0.912 | 17.120 | 0.724 | 0.859 | 0.665 | 1.000 |
| Qwen2.5-7B | 0.000 | 0.542 | 140.000 | 0.163 | 0.240 | 0.090 | 0.953 |
| Cuttlefish-8.3B | 0.551 | 0.964 | 19.280 | 0.792 | 0.879 | 0.722 | 1.000 |

*Table 20.* Detailed results on the reagent prediction task from the Mol-Instructions benchmark. Official splits are used for testing Cuttlefish, Qwen, and Mol-LLaMA models; other baselines are taken from the benchmark leaderboard.

| Model | Exact↑ | BLEU↑ | Levenshtein↓ | RDKit FTS↑ | MACC FTS↑ | Morgan FTS↑ | Validity↑ |
|---|---|---|---|---|---|---|---|
| Alpaca-7B | 0.000 | 0.026 | 29.037 | 0.029 | 0.016 | 0.001 | 0.186 |
| Baize-7B | 0.000 | 0.051 | 30.628 | 0.022 | 0.018 | 0.004 | 0.099 |
| ChatGLM-6B | 0.000 | 0.019 | 29.169 | 0.017 | 0.006 | 0.002 | 0.074 |
| LLaMA-2-7B | 0.000 | 0.003 | 28.040 | 0.037 | 0.001 | 0.001 | 0.001 |
| Vicuna-v1.5-13B | 0.000 | 0.010 | 27.948 | 0.038 | 0.002 | 0.001 | 0.007 |
| Galactica-6.7B | 0.000 | 0.141 | 30.760 | 0.036 | 0.127 | 0.051 | 0.995 |
| Text+Chem T5 | 0.000 | 0.225 | 49.323 | 0.039 | 0.186 | 0.052 | 0.313 |
| Mol-Ins.-Llama-2-7B | 0.044 | 0.224 | 23.167 | 0.237 | 0.364 | 0.213 | 1.000 |
| Mol-Ins.-Llama-3.1 | 0.101 | 0.648 | 18.326 | 0.412 | 0.521 | 0.375 | 1.000 |
| Mol-LLaMA-3.1-8.3B | 0.132 | 0.495 | 49.230 | 0.411 | 0.521 | 0.361 | 1.000 |
| Qwen2.5-7B | 0.039 | 0.428 | 64.420 | 0.258 | 0.378 | 0.257 | 0.955 |
| Cuttlefish-8.3B | 0.149 | 0.640 | 46.991 | 0.509 | 0.549 | 0.450 | 1.000 |

despite chemically aligned outputs.

**OpenQA** (Tab. 22). This task is predominantly text-only, lacking a specific molecular input, so structure-conditioning advantages are expected to be attenuated. Nevertheless, Cuttlefish remains competitive on lexical overlap and achieves the

*Table 21.* Detailed results on the retrosynthesis task from the Mol-Instructions benchmark. Official splits are used for testing Cuttlefish, Qwen, and Mol-LLaMA models; other baselines are taken from the benchmark leaderboard.

| Model | Exact↑ | BLEU↑ | Levenshtein↓ | RDKit FTS↑ | MACC FTS↑ | Morgan FTS↑ | Validity↑ |
|---|---|---|---|---|---|---|---|
| Alpaca-7B | 0.000 | 0.063 | 46.915 | 0.005 | 0.023 | 0.007 | 0.160 |
| Baize-7B | 0.000 | 0.095 | 44.714 | 0.025 | 0.050 | 0.023 | 0.112 |
| ChatGLM-6B | 0.000 | 0.117 | 48.365 | 0.056 | 0.075 | 0.043 | 0.046 |
| LLaMA-2-7B | 0.000 | 0.036 | 46.844 | 0.018 | 0.029 | 0.017 | 0.010 |
| Vicuna-v1.5-13B | 0.000 | 0.057 | 46.877 | 0.025 | 0.030 | 0.021 | 0.017 |
| Galactica-6.7B | 0.000 | 0.452 | 34.940 | 0.167 | 0.274 | 0.134 | 0.984 |
| Text+Chem T5 | 0.141 | 0.765 | 24.043 | 0.685 | 0.765 | 0.585 | 0.698 |
| Mol-Ins.-Llama-2-7B | 0.009 | 0.705 | 31.227 | 0.283 | 0.487 | 0.230 | 1.000 |
| Mol-Ins.-Llama-3.1 | 0.333 | 0.842 | 17.642 | 0.704 | 0.815 | 0.646 | 1.000 |
| Mol-LLaMA-3.1-8.3B | 0.340 | 0.877 | 38.324 | 0.708 | 0.822 | 0.601 | 1.000 |
| Qwen2.5-7B | 0.000 | 0.579 | 144.000 | 0.168 | 0.291 | 0.112 | 0.939 |
| Cuttlefish-8.3B | 0.395 | 0.854 | 26.310 | 0.747 | 0.839 | 0.681 | 1.000 |

*Table 22.* Detailed results on the OpenQA task from the Mol-Instructions benchmark. Official splits are used for testing Cuttlefish, Qwen, and Mol-LLaMA models; other baselines are taken from the benchmark leaderboard.

| Model | BLEU↑ | ROUGE-1↑ | BertScore↑ |
|---|---|---|---|
| Alpaca-7B | 0.003 | 0.088 | 0.824 |
| Baize-7B | 0.005 | 0.100 | 0.811 |
| ChatGLM-6B | 0.003 | 0.090 | 0.795 |
| LLaMA-2-7B | 0.003 | 0.100 | 0.814 |
| Vicuna-v1.5-13B | 0.004 | 0.097 | 0.814 |
| Galactica-6.7B | 0.000 | 0.039 | 0.794 |
| PMC_LLaMA | 0.007 | 0.788 | 0.625 |
| Mol-Ins.-Llama-2-7B | 0.024 | 0.221 | 0.837 |
| Mol-Ins.-Llama-3.1 | 0.010 | 0.198 | 0.846 |
| Mol-LLaMA-3.1-8.3B | 0.024 | 0.134 | 0.812 |
| Qwen2.5-7B | 0.793 | 0.204 | 0.846 |
| Cuttlefish-8.3B | 0.117 | 0.342 | 0.884 |

strongest semantic similarity, indicating preserved general instruction-following and paraphrase robustness under multimodal training. This pattern suggests that structure alignment acts as a complementary inductive bias that improves biomolecular semantics without sacrificing text-only competence.

### D.5. Completion and Reasoning Cases

**Examples of Reasoning.** Figs. 12 – 16 present qualitative reasoning examples across molecule, protein, DNA, and RNA tasks, illustrating how Cuttlefish integrates all-atom structural evidence into intermediate reasoning. In molecular captioning and molecular QA (Figs. 12 and Fig. 13), the model explicitly conditions its predictions on geometric and physicochemical cues (such as steric bulk, ionization state, and functional group-driven permeability) rather than relying on sequence-level correlations. This behavior reflects the effect of Geometry Grounding Adapter, which exposes verifiable geometric information to the LLM.

For macromolecular modalities, Cuttlefish exhibits domain-aware and task-specific reasoning grounded in structure. In protein captioning (Fig. 14), functional conclusions are tied to structurally localized domains and interaction partners, avoiding unsupported mechanistic extrapolation. Similarly, in DNA and RNA (Figs. 15 and Fig. 16), the model produces conservative, evidence-aligned judgments (e.g., promoter presence) without hallucinating precise motif locations or regulatory mechanisms. These examples indicate that geometry-grounded tokens encourage faithful abstraction, thereby preventing hallucinations.

Across all cases, the reasoning traces also demonstrate the scalability benefits of Scaling-Aware Patching. Complex all-atom inputs are selectively summarized through instruction-conditioned patching, enabling the model to retain structurally salient regions while suppressing redundant context. As a result, Cuttlefish maintains coherent reasoning across heterogeneous sequence lengths and structural complexities, supporting its effectiveness as a unified structure-aware LLM.

Sequence /C=C/C1=CC2=C(C=C1)OC(=C2C)C3=CC=C(C=C3)OC4=C(C=CC(=C4)C5=C(C6=C(O5)C=CC(=C6)/C=C/C)C)O

System You are a structure grounded chemistry assistant. You must reason before answering.\nRequirements:\n1. First produce <think> with a brief, substantive rationale grounded in the provided context and the molecular representation.\n2. Then produce <answer> containing the assistant response sentence.\n3. Then output a final line: Final answer: ..., where the text matches <answer> exactly.\n4. Do not add any extra text outside these three outputs.\n\nFormat strictly:\n<think>...</think>\n<answer>...</answer>\nFinal answer: ...",

Instruction Could you provide a description of this molecule?\nMolecule <mol>.

Response <think>The user asks for a description, but the provided context already supplies the key factual descriptor.\nTo avoid hallucinating functional or class annotations not supported by the dialogue, I will restate the provenance claim succinctly.\nI will keep the answer to a single sentence aligned with the given assistant cue.</think>\n<answer>The molecule is a natural product found in Piper aequale with data available.</answer>\n**Final answer: The molecule is a natural product found in Piper aequale with data available.**",

Ground Truth **The molecule is a natural product found in Piper aequale with data available** …

*Figure 12.* Reasoning example on the molecular captioning task. The model accepts a molecular input (along with its structure) and the instructions. The output will contain the reasoning part for the question.

Sequence CO[C@]12CC[C@@]3(C[C@@H]1[C@](C)(O)C(C)(C)C)[C@H]1Cc4ccc(O)c5c4[C@@]3(CCN1CC1CC1)[C@H]2O5

System You are a chemistry research assistant, and I'd like to test your professional ability on molecule understanding, please complete the following task:\n\nYou are provided with a molecule and asked a question about the molecule's property-related knowledge (property means physical, biological or chemical property in various environments or reactions), with four options given. Three of these options do not describe the given molecule, and you must select the correct option.\n\nNotice that here are some rules you need to follow:\n1. Your answer for each question should be one of A/B/C/D, which corresponds to the four options.\n2. Please give me your ANSWER for the given instances in the format 'Answer: ...'

Instruction Molecule: <mol>\nQuestion: What is the right information about this molecule's absorption?\n- Option A: It undergoes very little absorption following oral or topical administration.\n- Option B: It has poor gastrointestinal absorption and is therefore formulated as a sublingual tablet.\n- Option C: It has minimal oral absorption and its use has not been linked to instances of acute liver injury.\n- Option D: It has good absorption and enhanced oral bioavailability.\n"

Response <think>The structure is a **bulky, rigid polycyclic scaffold with multiple fused rings**, which tends to reduce passive diffusion through the gastrointestinal epithelium.\nIt includes a **tertiary amine**, so it will be protonated at physiological pH, increasing aqueous solubility but decreasing membrane permeability for oral uptake.\nIt also **contains phenolic and ether oxygens**, adding polarity and hydrogen bonding capacity, further penalizing gut permeability.\nA common mitigation for poor gastrointestinal absorption of ionizable, permeability limited compounds is non GI delivery such as sublingual administration.\nTherefore the option stating poor gastrointestinal absorption with sublingual formulation is most consistent.</think>\n<answer>B</answer>\n**Final answer: B**

Ground Truth **Answer: B**

*Figure 13.* Reasoning example on the molecular QA task. The model accepts a molecular input (along with its structure) and the instructions. The output will contain the reasoning part for the question.

**Entity type recognition.** Fig. 17 illustrates representative structure-only prompts demonstrating that the model reliably infers entity type without any auxiliary textual cues. Given only structural inputs, Cuttlefish correctly classifies the underlying entity as a small molecule, protein, DNA, or RNA. Notably, DNA and RNA are disambiguated solely from structural signals, without explicit sequence information. These examples validate that the model learns modality-specific structural priors and can perform accurate entity-type recognition from structure alone, establishing a prerequisite for downstream structure-grounded reasoning.

**Structure ensures faithful reasoning.** Fig. 18 illustrates a reliability gap driven by structural grounding. Cuttlefish extracts salient structure-conditioned cues and produces a coherent property-level rationale aligned with the correct absorption

**Sequence:** MAQGSGDQRAVGVADPEESSPNMIVYCKIEDIITKMQDDKTGGVPIRTVKSFLSKIPSVVTGTDIVQWLMKNLSIEDPVEAIHL GSLIAAQGYIFPISDHVLTMKDDGTFYRFQAPYFWPSNCWEPENTDYAIYLCKRTMQNKARLELADYEAENLARLQRAFARKWEFIFMQ AEAQVKIDRKKDKTERKILDSQERAFWDVHRPVPGCVNTTEMDIRKCRRLKNPQKVKKSVYGVTEESQAQSPVHVLSQPIRKTTKEDIR KQITFLNAQIDRHCLKMSKVAESLIAYTEQYVEYDPLITPAEPSNPWISDDVALWDIEMSKEPSQQRVKRWGFSFDEILKDQVGRDQFLRF LESEFSSENLRFWLAVQDLKKQPLQDVAKRVEEIWQEFLAPGAPSAINLDSHSYEITSQNVKDGGRYTFEDAQEHIYKLMKSDSYARFLR SNAYQDLLLAKKKGKSLAGKRLTGLMQSS",

**System** You are a helpful assistant specializing in protein analysis and biology. The instruction that describes a task is given, paired with protein structure. Write a response that appropriately completes the request."

**Instruction** Could you provide a detailed description of the protein structure <STRUCTURE> and its function?"

**Response** <think>The request asks for structure and function, and **the provided completion specifies the functional mechanism**, the responsible domain, and interaction partners that modulate activity and localization.\nI will restate these points with equivalent wording, avoid adding unsupported structural claims, and retain the experimental recommendations and isoform caveat.</think>\n<answer>This protein negatively regulates **GPCR signaling by functioning as a GAP for Gα subunits**. By accelerating the intrinsic GTP hydrolysis of Gα, it promotes the inactive GDP bound state and suppresses downstream signal propagation. The key structural determinant is the central RGS domain, which binds Gα with high affinity and catalytically enhances Gα mediated GTP hydrolysis.\n\nA major functional assembly is a heterodimer with the atypical Gβ5 subunit (GNB5), where the RGS6–GNB5 complex increases GNAO1 GTPase activity. Additional reported partners include GNAI1 and RGS7BP, with RGS7BP influencing subcellular distribution of the RGS–GNB5 complex. Association with GNB5 supports nuclear translocation, so GNB5 and RGS7BP are important determinants of both localization and effective GAP activity.\n\nMultiple isoforms may exist, and isoform variation can alter interaction specificity, targeting, and catalytic potency. Functional characterization should therefore quantify Gα GTP hydrolysis with and without GNB5 and RGS7BP and control for isoform dependent effects.</answer>

**Ground Truth** Answer: This protein functions as a regulator of **G protein-coupled receptor (GPCR) signaling by acting as a GTPase-accelerating protein (GAP) for Gα subunits**. It inhibits signal transduction by increasing the intrinsic GTPase activity of Gα proteins, thereby driving them into their inactive, GDP-bound state. The central RGS domain mediates high-affinity interaction with Gα and is responsible for accelerating Gα-catalyzed GTP hydrolysis.\n\nFunctionally relevant interactions include formation of a heterodimer with the atypical beta subunit Gβ5 (GNB5); the RGS6–GNB5 dimer specifically enhances GNAO1 GTPase activity. The protein also interacts with GNAI1 and with RGS7BP, the latter interaction modulating the subcellular disposition of the RGS–GNB5 heterodimer. Binding to GNB5 promotes translocation of the complex to the nucleus, so consider GNB5 and RGS7BP as key determinants of both activity and localization.\n\nBe aware that additional isoforms of this protein appear to exist; isoform diversity may influence interaction profiles, subcellular targeting, and GAP activity. When investigating its function, assay Gα GTP hydrolysis in the presence and absence of GNB5 and RGS7BP and account for possible isoform-specific effects."

*Figure 14.* Reasoning example on the GEO-AT protein captioning task. The model accepts a protein and its structure, and the instructions as input. The output will contain the reasoning part before the final captioning answer.

choice, whereas the Q-Former variant overgeneralizes from superficial motifs and thus selects an incorrect option, reflecting connector-bottleneck–induced distortion or hallucination. The sequence-only variant cannot access fine-grained structural evidence, leading to low-confidence reasoning that guesses the answer but lacks a verifiable structure-based justification, highlighting the importance of detailed structure for faithful inference.

**D.6. Data Contamination Analysis**

Table 23 quantifies potential train–test contamination between Mol-Instructions (Fang et al., 2023) and the training corpus of GEO-AT using both entity-level overlap and an n-gram lexical criterion. Concretely, for each benchmark task with test set $\mathcal{D}*\text{test}$ and GEO-AT training set $\mathcal{D}*\text{train}$, we measure the 13-gram overlap rate as

$$\text{OR}*13; =; \frac{1}{|\mathcal{D}*\text{test}|} \sum_{x \in \mathcal{D}*\text{test}} \mathbb{1}! \left( \mathcal{G}*13(x) \cap \mathcal{G}*13(\mathcal{D}*\text{train}) \neq \varnothing \right), \tag{21}$$

where $\mathcal{G}*13(x)$ denotes the set of contiguous 13-token substrings (13-grams) extracted from the text fields of example $x$ (e.g., instruction, input prompt, and/or target text, following the benchmark format), and $\mathcal{G}*13(\mathcal{D}*\text{train}) = \bigcup *x' \in \mathcal{D}*\text{train} \mathcal{G}* 13(x')$ (Brown et al., 2020). We adopt 13-grams because exact long-span matches are unlikely to occur by chance, making $\text{OR}_{13}$ a conservative indicator of near-duplicate textual leakage beyond exact entity identity. Entity-level overlap

---

Sequence:CTGGGGAGGTGGGAGGGAAGAGGTCAGCAGGGCTTAGGGAGTGAGCAGAAAGGTTTGGAGCACATCCTGGAGGTGGG
AAGGAAGGGATGGAAAGGAGGCTGACAAGCGGGCGGCGGGGGGGGGGGCGCCTTCCTCCTCTGAGAGGAGGCCCCGCCCCACGCT
ATCCGGGCTGTCAGCCCTGATCAACCCTGACAGAGTTCCCGGATGTGCTTTGCCGGATGTGCTTTCCCGGCGGCCATCTTGGGAGTC
TGAAGGACCTGAGGCATTTTGTGACGAGGATCGTCTCAGGTCAGCGGAGGGA

---

System You are a helpful assistant specializing in DNA analysis and biology. The instruction that describes a task is given, paired with nucleic acid sequences. Write a response that appropriately completes the request."

---

Instruction Can a promoter in human or mouse cells lacking a TATA box be found in the sequence <DNA>?

---

Response <think>The user asks whether a TATA-less promoter can be found in the provided DNA.\nThe given completion in the context asserts that **such a promoter was identified**.\nTo avoid over-claiming specific motif locations or promoter type beyond the provided statement, I will answer affirmatively and succinctly.</think>\n<answer>Yes, a promoter without a TATA sequence was identified.</answer>

---

Ground Truth Yes, a promoter without a TATA sequence was identified.

*Figure 15.* Reasoning example from benchmark DNA-Chat. The model accepts a DNA input with its 3D structure and the question. The output will contain the reasoning part before giving the final answer.

---

Sequence:TTTTTTCTCTACTAGCAGGCAACTGGGTCAGTTGGCTGGGAGGAAAGGGT
... (1388 chars omitted) ... GACTAACAGGAGACATAATTATTCAATTAAAAAGTTTACTTTAGACCACA

---

System You are a helpful assistant specializing in RNA analysis and biology. The instruction that describes a task is given, paired with nucleic acid sequences. Write a response that appropriately completes the request.

---

Instruction Determine the degradation rate of the given mouse RNA sequence <RNA> within the range of -5 to 5."

---

Response <think>The sequence is long and **contains extended U rich and AU rich segments**, which are often associated with reduced mRNA stability via RNA binding proteins and decay pathways.\nThere is also a prominent poly(A) like tail signal at the end, but the overall composition **suggests moderate degradation** rather than extreme stability.\nGiven these cues, a mildly positive degradation score within the requested range is most consistent.</think><answer>This RNA sequence demonstrates a degradation rate of 1.7.</answer>

---

Ground Truth This RNA sequence demonstrates a degradation rate of 1.78.

*Figure 16.* Reasoning example from benchmark RNA-QA. The model accepts an RNA input with its 3D structure and the instructions. The output will contain the reasoning part for the question.

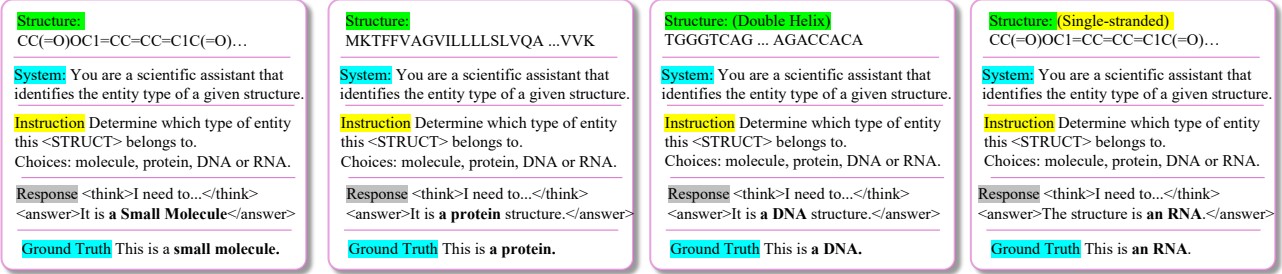

*Figure 17.* Examples of Cuttlefish that recognize entities solely based on structure (non-modality sequence inputs) of all-atom modalities include molecule, protein, DNA, and RNA.

complements this by detecting cases where the same underlying biological or chemical entity appears in both training and test splits.

Overall, the overlap remains low across tasks in Tab. 23. Protein-oriented tasks show relatively higher entity-level overlap, which is expected since both GEO-AT and several Mol-Instructions datasets source proteins from UniProt (Consortium, 2019). To ensure a fair comparison, we perform explicit decontamination: before fine-tuning, we remove from $\mathcal{D}_{\text{train}}$ any

**Sequence:**
CO[C@]12CC[C@@]3(C[C@@H]1[C@](C)(O)C(C)(C)C)[C@H]1Cc4ccc(O)c5c4[C@@]3(CCN1CC1CC1)[C@H]2O5

**System** You are a chemistry research assistant, and I'd like to test your professional ability on molecule understanding, please complete the following task:\n\nYou are provided with a molecule and asked a question about the molecule's property-related knowledge (property means physical, biological or chemical property in various environments or reactions), with four options given. Three of these options do not describe the given molecule, and you must select the correct option.\n\nNotice that here are some rules you need to follow:\n1. Your answer for each question should be one of A/B/C/D, which corresponds to the four options.\n2. Please give me your ANSWER for the given instances in the format 'Answer: ...'

**Instruction** Molecule: <mol>\nQuestion: What is the right information about this molecule's absorption?\n- Option A: It undergoes very little absorption following oral or topical administration.\n- Option B: It has poor gastrointestinal absorption and is therefore formulated as a sublingual tablet.\n- Option C: It has minimal oral absorption and its use has not been linked to instances of acute liver injury.\n- Option D: It has good absorption and enhanced oral bioavailability.

**Ground Truth Answer: B**

*Cuttlefish*

Response <think>\nThe molecule exhibits a large, rigid, polycyclic scaffold with substantial steric bulk and limited conformational flexibility. Although it contains several heteroatoms, the overall architecture remains highly lipophilic and sterically congested, which typically restricts passive diffusion across the gastrointestinal epithelium.\nCompounds with these characteristics often show poor oral absorption and are instead delivered through alternative routes to improve systemic exposure and avoid extensive first-pass metabolism. Such absorption behavior is consistent with formulations designed for sublingual administration rather than standard oral dosing. </think>The correct answer is **Answer: B**. ✔

*Q-Former variant*

Response <think>The molecule contains multiple oxygen atoms and appears chemically complex, suggesting the presence of polar functional groups. Polar functionalities are commonly associated with enhanced solubility and improved interaction with biological membranes. Based on these features, the compound is expected to be readily absorbed following oral administration and to exhibit favorable bioavailability without requiring specialized formulation strategies.</think>
**Answer: D** ✘

*Sequence-only variant*

Response <think> The molecule appears relatively large and structurally complex, which may influence its absorption properties. However, without a detailed understanding of how its structural features interact with biological barriers, the absorption behavior remains uncertain. Given that some large bioactive compounds are delivered using non-standard oral routes, a formulation that avoids conventional gastrointestinal absorption seems plausible. I am not fully certain; however, based on the available context, option B appears to be the most plausible.</think>
**Answer: B** ✔

*Figure 18.* Qualitative comparison of three Cuttlefish model variants on a molecular absorption multiple-choice question. The figure shows (top) the input small-molecule SMILES, the system prompt and instruction with four options, and the ground-truth label, and (bottom) the corresponding responses from Cuttlefish, a Q-Former variant, and a sequence-only variant, with highlighted text spans indicating the correctness of rationale cues.

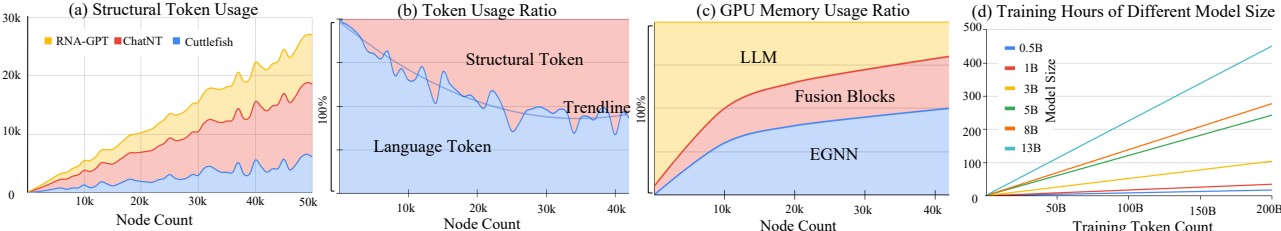

*Figure 19.* Training efficiency analysis of Cuttlefish versus ChatNT and RNA-GPT. (a) Structural token usage versus node count. (b) Structural token usage ratio (structural vs. language tokens) versus node count, with a fitted trendline. (c) GPU memory usage ratio decomposed into EGNN, fusion blocks, and LLM components as node count increases. (d) Training hours versus training token count for backbones from 0.5B to 13B parameters, with the y-axis regularized to H100-80G GPU hours (LoRA applied).

example that overlaps with a benchmark test instance under either criterion (entity match or 13-gram match). This uniform filtering ensures that reported performance reflects generalization rather than memorization or benchmark leakage.

## D.7. Training Efficiency Analysis

Figure 19 (a–d) characterizes how Cuttlefish improves scalability relative to sequence-centric baselines ChatNT (de Almeida et al., 2025) and RNA-GPT (Xiao et al., 2024b) as structural size and model scale increase. (a) shows that baseline structural token usage grows near-linearly with node count, reflecting weakly adaptive interfaces that directly expose expanding structures to the LLM context. In contrast, Cuttlefish maintains a substantially lower structural budget at large entities by

*Table 23.* Entity-level and 13-gram overlap rates across 3 types of tasks from benchmark Mol-Instructions (before filtered).

| Dataset / Task | Entity-Level Overlap Rate | 13-gram Overlap Rate |
|---|---|---|
| *Molecule-oriented tasks* | | |
| Molecule description generation | 5.57% | 6.85% |
| Description-guided molecule design | 0.78% | 0.88% |
| Forward reaction prediction | 5.02% | 2.59% |
| Retrosynthesis | 5.07% | 2.61% |
| Reagent prediction | 1.34% | 0.71% |
| Property prediction | 3.47% | 2.21% |
| *Protein-oriented tasks* | | |
| Protein design | 5.68% | 3.59% |
| Protein functional description | 4.97% | 3.20% |
| Catalytic activity prediction | 5.58% | 3.30% |
| Domain/motif prediction | 1.81% | 1.32% |
| Protein structural prediction | 2.14% | 1.79% |
| *Biomolecular text (NLP) tasks* | | |
| True or False question | 3.79% | 1.95% |
| Multiple-choice question | 2.86% | 1.94% |
| Chemical entity recognition | 1.61% | 1.07% |
| Chemical-disease interaction extraction | 2.74% | 2.24% |
| Chemical-protein interaction extraction | 2.96% | 2.09% |
| Open question | 3.66% | 2.64% |

*Table 24.* Ablation on different model sizes. Qwen-2.5 is used as the base model as it has the widest size coverage. Each variant is evaluated on GEO-AT with BERTScore reported.

| Model Size | Molecule | Protein | DNA&RNA | Average |
|---|---|---|---|---|
| 0.5B | 0.453 | 0.422 | 0.382 | 0.419 |
| 1.5B | 0.586 | 0.577 | 0.544 | 0.569 |
| 3B | 0.608 | 0.578 | 0.621 | 0.602 |
| 7B | 0.863 | 0.816 | 0.0.832 | 0.840 |
| 14B | 0.897 | 0.861 | 0.834 | 0.864 |

*Table 25.* Effect of training epochs on LLM adaptation tuning stage. Cuttlefish-8B is evaluated on GEO-AT, four tasks with BERTScore reported.

| Epochs | Molecule | Protein | DNA&RNA | Average |
|---|---|---|---|---|
| 2 | 0.875 | 0.896 | 0.816 | 0.862 |
| 3 | 0.881 | 0.886 | 0.820 | 0.863 |
| 4 | 0.877 | 0.856 | 0.803 | 0.845 |
| 5 | 0.848 | 0.823 | 0.789 | 0.820 |
| 10 | 0.758 | 0.770 | 0.739 | 0.756 |

*Table 26.* Ablation on four pooling methods (fine-designed weighted pooling, mean/max/min pooling), evaluated on GEO-AT.

| Pooling | Molecule | Protein | DNA&RNA | Average | Avg. Drop |
|---|---|---|---|---|---|
| Weighted | 0.875 | 0.896 | 0.816 | 0.862 | 0.00% |
| Mean | 0.848 | 0.812 | 0.802 | 0.821 | 4.14% |
| Max | 0.831 | 0.821 | 0.796 | 0.816 | 4.60% |
| Min | 0.851 | 0.777 | 0.786 | 0.804 | 5.76% |

*Table 27.* Ablation on maximum patch count from 256 to 4096, Cuttlefish is evaluated on four tasks from GEO-AT.

| Max Patch Count | Molecule | Protein | DNA&RNA | Average | Avg. Drop |
|---|---|---|---|---|---|
| 2048 | 0.875 | 0.896 | 0.816 | 0.862 | 0.00% |
| 4096 | 0.875 | 0.897 | 0.806 | 0.859 | 0.29% |
| 1024 | 0.875 | 0.798 | 0.750 | 0.808 | 5.45% |
| 512 | 0.853 | 0.776 | 0.748 | 0.792 | 6.98% |
| 256 | 0.842 | 0.659 | 0.693 | 0.731 | 13.08% |

activating Scaling-Aware Patching to select instruction-relevant anchors and grow to patches. (b) Further indicates that the structural-to-language token composition is acceptable as node count increases, with a clear controllable trend in the token usage ratio.

The remaining panels link token efficiency to practical computing. (c) decomposes GPU memory usage into EGNN encoder, fusion blocks, and the LLM, showing that memory growth is dominated by the LLM backbone, while the incremental cost of structure processing and fusion grows sublinearly with node count under Cuttlefish 's adaptive patching. This contrasts with baselines where increased structural tokens directly inflate LLM-side activation and attention costs. (d) reports training hours as a function of training token count for different backbone sizes (1B–13B), illustrating the superlinear time growth as tokens increase and highlighting why controlling structural tokenization is critical for feasible scaling. Together, (a–d) support that Geometry Grounding Adapter and Scaling-Aware Patching jointly decouple structural resolution from LLM context length, yielding more stable memory footprints and improved end-to-end training efficiency.

*Table 29.* Comparison of Cuttlefish against general LLM backbones on GEO-AT under reasoning and non-reasoning settings. Results are reported as the average METEOR and BERTScore (higher is better) across GEO-AT evaluation tasks, along with the corresponding absolute improvements over the backbone-only baseline. **B.**: backbone-only; **C.**: Cuttlefish.

| Backbone | Reasoning | B. METEOR↑ | B. BERT-S↑ | C. METEOR↑ | C. BERT-S↑ | (ΔMETEOR / ΔBERT-S)↑ |
|---|---|---|---|---|---|---|
| Qwen-2.5-7B-Instruct | No | 0.143 | 0.653 | 0.330 | 0.840 | +0.187 / +0.187 |
| Mistral-3-8B-Instruct-2512 | No | 0.172 | 0.683 | 0.310 | 0.750 | +0.138 / +0.067 |
| GLM-4-9B-0414 | No | 0.155 | 0.621 | 0.367 | 0.727 | +0.212 / +0.106 |
| Qwen3-8B | Yes | 0.107 | 0.674 | 0.428 | 0.876 | +0.321 / +0.202 |
| DeepSeek-R1-D-Qwen-7B | Yes | 0.169 | 0.692 | 0.369 | 0.803 | +0.200 / +0.111 |
| Mistral-3-8B-Reasoning-2512 | Yes | 0.176 | 0.744 | 0.335 | 0.876 | +0.159 / +0.132 |

# E. Extended Ablations

## E.1. Effect of Backbone Model Size

Tab. 24 analyzes the impact of LLM backbone size using Qwen-2.5 (Bai et al., 2025) variants from 0.5B to 14B parameters. Performance improves consistently with scale across all modalities, indicating that Cuttlefish effectively leverages increased language capacity without saturating early. Gains are most pronounced from 0.5B to 3B, while improvements beyond 7B become incremental, suggesting diminishing returns once sufficient language expressivity is reached. Importantly, strong DNA and RNA performance at 7B–14B confirms that geometry-grounded tokens remain effective at scale rather than being overshadowed by larger text priors.

## E.2. Effect of LLM Adaptation Epochs

Tab. 25 studies the number of epochs used in the LLM adaptation stage. Performance peaks at 2–3 epochs across modalities, while additional training leads to consistent degradation. This trend indicates that moderate adaptation is sufficient for aligning the frozen LLM with structure-grounded tokens, whereas over-training induces overfitting and harms generalization. These results motivate the use of early stopping in the adaptation phase and validate the efficiency of Geometry Grounding Adapter in rapidly injecting geometric information.

## E.3. Pooling Strategy for Patch Aggregation

Tab. 26 compares pooling strategies used to aggregate node-level features within each patch. The proposed weighted pooling consistently outperforms mean/max/min pooling, yielding the highest average score and zero performance drop. This confirms that learned, instruction-conditioned weighting is critical for preserving structurally salient signals, whereas uniform or extreme-value pooling discards fine-grained geometric cues required for accurate reasoning.

## E.4. Maximum Patch Count

Tab. 27 evaluates the effect of the maximum allowable patch count. Performance is stable between 2048 and 4096 patches, indicating that Scaling-Aware Patching can effectively utilize additional capacity when available. However, reducing the budget below 1024 patches causes sharp degradation, particularly for proteins and nucleic acids, where structural complexity is higher. This demonstrates a clear trade-off between efficiency and representational sufficiency, and validates the default choice of 2048 as a balanced operating point for scaling-aware patching.

## E.5. Effect of Training Stages

*Table 28.* Ablation on three training stages, evaluated on GEO-AT.

| Training Stage | Molecule | Protein | DNA&RNA | Average | Avg. Drop |
|---|---|---|---|---|---|
| Cuttlefish | 0.875 | 0.896 | 0.816 | 0.862 | 0 |
| w/o LLM Adaptation | 0.854 | 0.889 | 0.817 | 0.853 | 0.88% |
| w/o Alignment Tuning | 0.742 | 0.787 | 0.801 | 0.777 | 8.54% |
| Random Encoder | 0.376 | 0.382 | 0.565 | 0.441 | 42.11% |

Tab. 28 indicates that performance is dominated by structure-side representation learning and alignment. Replacing the pretrained encoder with a random encoder collapses the performance, confirming the necessity of modality pretraining. Removing alignment tuning yields a large drop as well, showing that initialized alignment is critical. LLM adaptation provides a smaller gain, suggesting it mainly refines instruction following after structural grounding is established.

*Table 30.* Prompt-setting ablation on GEO-AT using the same reasoning-capable backbones under non-reasoning and reasoning inference settings. Results are reported as average METEOR and BERTScore (higher is better), together with the relative improvements brought by Cuttlefish under each prompt setting. **B.**: backbone-only; **C.**: Cuttlefish.

| Backbone | Prompt setting | B. METEOR↑ | B. BERT-S↑ | C. METEOR↑ | C. BERT-S↑ | Improvement (ΔMETEOR / ΔBERT-S)↑ |
|---|---|---|---|---|---|---|
| Qwen3-8B | Non-reasoning | 0.051 | 0.634 | 0.055 | 0.689 | 7.84% / 8.68% |
| Qwen3-8B | Reasoning | 0.107 | 0.674 | 0.428 | 0.876 | 300.00% / 29.97% |
| DeepSeek-R1-D-Qwen-7B | Non-reasoning | 0.201 | 0.649 | 0.321 | 0.703 | 59.70% / 8.32% |
| DeepSeek-R1-D-Qwen-7B | Reasoning | 0.227 | 0.662 | 0.369 | 0.803 | 62.56% / 21.30% |
| Mistral-3-8B-Reasoning-2512 | Non-reasoning | 0.113 | 0.687 | 0.298 | 0.703 | 163.72% / 2.33% |
| Mistral-3-8B-Reasoning-2512 | Reasoning | 0.176 | 0.744 | 0.335 | 0.776 | 90.34% / 4.30% |
| **Average gain** | **Non-reasoning** | | | | | **77.09% / 6.44%** |
| **Average gain** | **Reasoning** | | | | | **150.97% / 18.52%** |

*Table 31.* Results on the reasoning-oriented benchmarks ChemIQ and MolecularIQ. Non-reasoning and reasoning backbones are compared before and after adding Cuttlefish, and relative gains are reported for each benchmark. Metrics are success rate and accuracy (higher is better) for ChemIQ and MolecularIQ, respectively.

| Backbone | Type | ChemIQ Base↑ | ChemIQ Cuttlefish ↑ | Gain↑ | MolecularIQ Base↑ | MolecularIQ Cuttlefish ↑ | Gain↑ |
|---|---|---|---|---|---|---|---|
| Qwen-2.5-7B-Instruct | Non-reasoning | 0.051 | 0.055 | 7.84% | 0.101 | 0.105 | 3.96% |
| Llama-3.1-8B-Instruct | Non-reasoning | 0.077 | 0.081 | 5.19% | 0.077 | 0.081 | 5.19% |
| Qwen3-8B | Reasoning | 0.066 | 0.091 | 37.88% | 0.155 | 0.188 | 21.29% |
| DeepSeek-R1-D-Qwen-7B | Reasoning | 0.232 | 0.312 | 34.48% | 0.234 | 0.277 | 18.38% |
| Mistral-3-8B-Reasoning-2512 | Reasoning | 0.112 | 0.134 | 19.64% | 0.123 | 0.163 | 32.52% |
| **Avg. Gain (non-reasoning)** | | | | **6.50%** | | | **4.58%** |
| **Avg. Gain (reasoning)** | | | | **30.67%** | | | **24.06%** |

## E.6. Backbone and Prompt-Setting Ablation on GEO-AT

Tab. 29 quantifies the absolute improvement Cuttlefish brings over six backbone-only baselines across reasoning and non-reasoning settings on GEO-AT. Gains are consistently positive for every backbone, confirming that structural grounding is complementary to language capacity regardless of inference mode. Reasoning-capable backbones benefit more: Qwen3-8B achieves the largest METEOR gain (+0.321), suggesting that structural tokens synergize with chain-of-thought deliberation.

Tab. 30 further probes the same reasoning-capable backbones under both non-reasoning and reasoning inference settings. Cuttlefish brings substantial relative METEOR gains in both modes, with an average of 77.09% under non-reasoning prompts and 150.97% under reasoning prompts. The larger gains under reasoning prompts indicate that structural tokens synergize with chain-of-thought: geometric priors guide multi-step reasoning, amplifying the benefit of both structural grounding and deliberation simultaneously.

## E.7. Reasoning Benchmark Results: ChemIQ and MolecularIQ

Tab. 31 reports performance on ChemIQ (chemical problem-solving success rate) and MolecularIQ (molecular property reasoning accuracy). Cuttlefish yields consistent gains on both benchmarks, with reasoning backbones benefiting substantially more than non-reasoning ones (average gain of 30.67% vs. 6.50% on ChemIQ). This asymmetry suggests that structural grounding amplifies the model's ability to leverage extended deliberation for chemistry tasks, where geometric cues guide multi-step reasoning.

## E.8. Same-Backbone Connector Comparison

Tab. 32 provides a controlled comparison by fixing the backbone to Llama-3.1-8B-Instruct and varying only the connector, directly isolating the architectural contribution of Cuttlefish. Cuttlefish outperforms both Mol-Instructions and Mol-LLaMA connectors on every modality under both METEOR and BERTScore. The margins are most pronounced on DNA and RNA, where the adaptive token budget better accommodates longer and more repetitive nucleic acid sequences that challenge fixed-length connectors.

## E.9. Structural Token-Usage by Entity Size

Tab. 33 contrasts structural token consumption across entity size bins for four approaches. The sequence-only baseline uses no structural tokens; Q-Former enforces a rigid 256-token budget at all sizes; ChatNT-style connectors scale linearly with entity size, quickly consuming thousands of tokens for large structures; Cuttlefish adapts its budget, using far fewer tokens

*Table 32.* Same-backbone comparison on GEO-AT using Llama-3.1-8B-Instruct. METEOR and BERTScore are reported (higher is better) by modality and as overall averages.

| Method | Backbone | Molecule↑ | Protein↑ | DNA↑ | RNA↑ | Average↑ |
|---|---|---|---|---|---|---|
| *METEOR* | | | | | | |
| Backbone Only | Llama-3.1-8B-Instruct | 0.229 | 0.178 | 0.175 | 0.175 | 0.189 |
| Mol-Instructions connector | Llama-3.1-8B-Instruct | 0.291 | 0.315 | 0.364 | 0.330 | 0.325 |
| Mol-LLaMA connector | Llama-3.1-8B-Instruct | 0.319 | 0.328 | 0.381 | 0.306 | 0.334 |
| Cuttlefish | Llama-3.1-8B-Instruct | 0.391 | 0.417 | 0.529 | 0.403 | 0.435 |
| *BERTScore* | | | | | | |
| Backbone Only | Llama-3.1-8B-Instruct | 0.778 | 0.742 | 0.658 | 0.646 | 0.706 |
| Mol-Instructions connector | Llama-3.1-8B-Instruct | 0.789 | 0.789 | 0.786 | 0.738 | 0.775 |
| Mol-LLaMA connector | Llama-3.1-8B-Instruct | 0.830 | 0.758 | 0.754 | 0.731 | 0.768 |
| Cuttlefish | Llama-3.1-8B-Instruct | 0.875 | 0.896 | 0.816 | 0.868 | 0.864 |

*Table 33.* Structural token-usage comparison by entity size on GEO-AT. Variants include sequence-only, fixed-token Q-Former, sequence-exposed structural-token, and Cuttlefish, reporting the atom-count range and the corresponding structural token-usage range for each method.

| Model | Token policy | Atom-count range | Token-usage range |
|---|---|---|---|
| Sequence-only baseline | No structural tokens | [1, 256) | 0 |
| Q-Former variant | Fixed | [1, 256) | 256 |
| Q-Former variant | Fixed | [256, 1K) | 256 |
| Q-Former variant | Fixed | [1K, 8K) | 256 |
| Q-Former variant | Fixed | [8K, 30K) | 256 |
| ChatNT-style baseline | Sequence-exposed structural tokens | [1, 256) | (0, 75) |
| ChatNT-style baseline | Sequence-exposed structural tokens | [256, 1K) | [75, 250) |
| ChatNT-style baseline | Sequence-exposed structural tokens | [1K, 8K) | [250, 2000) |
| ChatNT-style baseline | Sequence-exposed structural tokens | [8K, 30K) | [2000, 7500) |
| **Cuttlefish** | Adaptive | [1, 256) | (0, 17) |
| **Cuttlefish** | Adaptive | [256, 1K) | [18, 85) |
| **Cuttlefish** | Adaptive | [1K, 8K) | [86, 816) |
| **Cuttlefish** | Adaptive | [8K, 30K) | [217, 2116) |

*Table 34.* Structure-source analysis on the GEO-AT protein tasks. ROUGE-L (higher is better) is reported for protein function (PF), functional description (FD), catalytic activity (CA), and domain or motif prediction (DM).

| STRUCTURE SOURCE | PF↑ | FD↑ | CA↑ | DM↑ | AVERAGE↑ |
|---|---|---|---|---|---|
| PDB | 0.486 | 0.520 | 0.551 | 0.495 | 0.513 |
| ALPHAFOLD2 | 0.481 | 0.498 | 0.577 | 0.448 | 0.501 |
| DELTA (PDB − AF2) | 0.005 | 0.022 | −0.027 | 0.047 | 0.012 |

than ChatNT for small entities while scaling gracefully for larger ones. This demonstrates that the adaptive policy avoids over-allocation for simple structures while preserving representational capacity for complex ones.

### E.10. Protein Structure Quality Analysis

Tab. 34 compares performance when protein structures are sourced from experimental PDB entries versus AlphaFold2 predictions. The average delta is only 0.012 ROUGE-L, with PDB marginally better on most sub-tasks and AF2 slightly better on catalytic activity. This parity confirms that Cuttlefish does not overfit to crystallographic data quality and remains fully applicable to proteins that lack experimental structures.

Tab. 35 further stratifies AF2-derived structures by pLDDT confidence bins. Performance degrades gracefully from 0.513 at pLDDT≥90 to 0.393 at pLDDT<70. Notably, the 80–90 bin slightly outperforms the highest-confidence bin on catalytic activity and domain/motif prediction, suggesting moderate structural flexibility can benefit activity-related reasoning. Together, these results confirm Cuttlefish remains effective across both structure sources and confidence levels, substantially broadening its practical scope.

*Table 35.* AlphaFold2 confidence analysis on the GEO-AT protein tasks. AlphaFold2-derived structures are stratified by pLDDT confidence bins, and ROUGE-L (higher is better) is reported for protein function (PF), functional description (FD), catalytic activity (CA), and domain or motif prediction (DM), together with the average.

| AF2 CONFIDENCE BIN (PLDDT) | PF↑ | FD↑ | CA↑ | DM↑ | AVERAGE↑ |
|---|---|---|---|---|---|
| ≥ 90 | 0.486 | 0.520 | 0.551 | 0.495 | 0.513 |
| 80–90 | 0.479 | 0.491 | 0.576 | 0.537 | 0.521 |
| 70–80 | 0.434 | 0.439 | 0.552 | 0.458 | 0.471 |
| < 70 | 0.354 | 0.394 | 0.461 | 0.364 | 0.393 |

*Table 36.* Functional-group grounding evaluation on GEO-AT. Top-1 and Top-3 accuracy (higher is better) are reported.

| METHOD | TOP-1 ACC.↑ | TOP-3 ACC.↑ |
|---|---|---|
| RANDOM SELECTION | 0.02 | 0.05 |
| GATE W/O SOFTMAX | 0.46 | 0.85 |
| OUR GATE | 0.49 | 0.93 |

*Table 37.* Sensitivity analysis of the maximum injected structural-token budget on GEO-AT. BERTScore is reported (higher is better).

| MAX INJECTED STRUCTURAL TOKENS | MOLECULE↑ | PROTEIN↑ | DNA&RNA↑ | AVERAGE↑ | AVG. DROP↓ |
|---|---|---|---|---|---|
| 2048 | 0.875 | 0.896 | 0.816 | 0.862 | 0.00% |
| 4096 | 0.875 | 0.897 | 0.806 | 0.859 | 0.29% |
| 1024 | 0.875 | 0.798 | 0.750 | 0.808 | 5.45% |
| 512 | 0.853 | 0.776 | 0.748 | 0.792 | 6.98% |
| 256 | 0.842 | 0.659 | 0.693 | 0.731 | 13.08% |

### E.11. Functional-Group Grounding Accuracy

Tab. 36 evaluates the instruction-conditioned anchor gate by its ability to concentrate attention on relevant functional groups. The full gate achieves Top-1 accuracy of 0.49 and Top-3 accuracy of 0.93, substantially above both random selection (0.02 / 0.05) and the ablated variant without softmax normalization (0.46 / 0.85). This confirms that softmax normalization is essential for calibrating gate weights, not merely the presence of gating.

### E.12. Hyperparameter Sensitivity Analysis

Tab. 37 evaluates sensitivity to the maximum injected structural-token budget. Performance is stable between 2048 and 4096 tokens (average drop ≤0.29%), but degrades sharply below 1024 (5.45%–13.08%). This confirms that 2048 is a well-chosen default providing sufficient representational capacity without unnecessary overhead.

Tab. 38 varies the mass threshold $\rho$ and maximum anchor count $K_{\max}$ in Scaling-Aware Patching. Both knobs show stable performance around their defaults ($\rho$=0.10, $K_{\max}$=2048). For $\rho$, performance peaks at 0.10 and degrades symmetrically on either side. For $K_{\max}$, increasing to 4096 provides a marginal gain (average 0.884 vs. 0.869), while reducing below 1024 causes degradation especially on DNA&RNA.

Tab. 39 evaluates sensitivity to the assignment temperature $\tau$, distance scale $s$, and fusion block count $L_f$. All three knobs produce stable performance within a neighborhood of their defaults ($\tau$=0.10, $s$=1.0, $L_f$=8). Across all three tables, the default configurations are near-optimal within reasonable ranges, confirming that Cuttlefish does not require fine-grained hyperparameter tuning.

## F. Limitation and Future Work

### F.1. Limitations

Cuttlefish integrates (i) geometry-grounded cross-modal alignment and (ii) instruction-conditioned scaling-aware patching to expose all-atom structure to a frozen LLM under an adaptive token budget. Despite improved structural faithfulness, Cuttlefish remains limited by extremely large entities, upstream structure quality, and weak reasoning supervision (Tab. 40).

*Table 38.* Sensitivity analysis of the anchor-selection hyperparameters on GEO-AT. The table varies the mass threshold $\rho$ and maximum anchor count $K_{max}$ in Scaling-Aware Patching, and reports BERTScore (higher is better) on molecule, protein, and DNA&RNA tasks, together with the overall average.

| KNOB | SETTING | MOLECULE↑ | PROTEIN↑ | DNA&RNA↑ | AVERAGE↑ |
|------|---------|-----------|----------|----------|----------|
| MASS THRESHOLD $\rho$ | 0.05 | 0.850 | 0.805 | 0.782 | 0.812 |
| MASS THRESHOLD $\rho$ | 0.10 (DEFAULT) | 0.875 | 0.896 | 0.836 | 0.869 |
| MASS THRESHOLD $\rho$ | 0.2 | 0.826 | 0.918 | 0.780 | 0.841 |
| MASS THRESHOLD $\rho$ | 0.3 | 0.852 | 0.903 | 0.763 | 0.839 |
| $K_{max}$ | 256 | 0.820 | 0.903 | 0.734 | 0.819 |
| $K_{max}$ | 512 | 0.824 | 0.929 | 0.710 | 0.821 |
| $K_{max}$ | 1024 | 0.773 | 0.878 | 0.752 | 0.801 |
| $K_{max}$ | 2048 (DEFAULT) | 0.875 | 0.896 | 0.836 | 0.869 |
| $K_{max}$ | 4096 | 0.915 | 0.871 | 0.866 | 0.884 |

*Table 39.* Sensitivity analysis of soft patch growth and adapter-side hyperparameters on GEO-AT. BERTScore (higher is better) on four tasks is reported, together with the overall average.

| KNOB | SETTING | MOLECULE↑ | PROTEIN↑ | DNA&RNA↑ | AVERAGE↑ |
|------|---------|-----------|----------|----------|----------|
| ASSIGNMENT TEMPERATURE $\tau$ | 0.05 | 0.843 | 0.915 | 0.827 | 0.862 |
| ASSIGNMENT TEMPERATURE $\tau$ | 0.10 (DEFAULT) | 0.875 | 0.896 | 0.836 | 0.869 |
| ASSIGNMENT TEMPERATURE $\tau$ | 0.2 | 0.897 | 0.869 | 0.830 | 0.865 |
| ASSIGNMENT DISTANCE SCALE $s$ | 0.5 | 0.845 | 0.828 | 0.873 | 0.849 |
| ASSIGNMENT DISTANCE SCALE $s$ | 1.0 (DEFAULT) | 0.875 | 0.896 | 0.836 | 0.869 |
| ASSIGNMENT DISTANCE SCALE $s$ | 2 | 0.817 | 0.881 | 0.846 | 0.848 |
| FUSION BLOCKS $L_f$ | 4 | 0.897 | 0.862 | 0.827 | 0.862 |
| FUSION BLOCKS $L_f$ | 8 (DEFAULT) | 0.875 | 0.896 | 0.836 | 0.869 |
| FUSION BLOCKS $L_f$ | 12 | 0.887 | 0.818 | 0.820 | 0.842 |

*Table 40.* Key limitations of Cuttlefish and corresponding future directions (three most reviewer-critical items).

| Limitation | Why this is acceptable / common | Future work |
|------------|--------------------------------|-------------|
| No support for extremely large entities (e.g., > 50K nodes) | Scientific workflows and human interpretation typically focus on functional subregions (domains, active sites, loci) rather than a single-pass global readout; in our setting, LLM context and compute budgets further motivate segment-level reasoning. | Retrieval-augmented hierarchical reasoning (coarse-to-fine chunking, region proposals, iterative zoom-in) with memory-based summarization across passes. |
| Sensitivity to structure correctness / pre-processing choices | Structure-grounded modeling intrinsically depends on geometric fidelity; we therefore prioritize high-quality inputs (validated coordinates; controlled conformers) to avoid conflating model limitations with upstream structural noise. | Uncertainty-aware tokens (confidence/quality conditioning), multi-conformer / ensemble supervision, and structure-quality estimation are integrated into training and inference. |
| Reasoning supervision is coarse (no native all-atom reasoning corpus) | Large-scale, structure-grounded reasoning traces are scarce; our supervision is primarily format-stable and regularizes outputs, while most reasoning capacity is inherited from the backbone LLM. | Construct domain-grounded reasoning datasets and enforce evidence-based rationale learning with verifiable geometric attribution. |

Beyond the three core limitations retained in the table, we note two additional constraints that are currently out of scope: (a) we do not explicitly model DNA folding or higher-order genome organization (e.g., chromatin-scale 3D ensembles), since most available all-atom nucleic-acid benchmarks emphasize local, structure-conditioned tasks; and (b) residual non-structural LLM hallucinations (factual/interpretive) may still occur in open-ended generation, even when structural hallucination is reduced.

## F.2. Future Work

Tab. 40 suggests three actionable extensions: (i) extreme-scale structure understanding via hierarchical segmentation, retrieval, and iterative multi-pass summarization; (ii) robustness to upstream structure noise through uncertainty-aware tokens and multi-conformer/ensemble training; and (iii) domain-grounded reasoning supervision by building all-atom

reasoning corpora with evidence-tethered rationales. In addition, two orthogonal directions remain important for broadening scope and trustworthiness: folding-aware, multi-scale nucleic-acid modeling (secondary/tertiary constraints and long-range contacts) and post-hoc verification (structure-consistency checks, tool-assisted validation, and uncertainty calibration) to further reduce residual hallucinations in open-ended generation.

## G. Reproducibility

We release the full codebase, Cuttlefish checkpoints, and the proposed dataset GEO-AT at https://github.com/zihao-jing/Cuttlefish. The repository includes comprehensive documentation and scripts to reproduce training, evaluation, and inference, including environment setup, preprocessing, command-line entry points, and configuration files for all reported experiments. All hyperparameters and implementation details not fully enumerated (e.g., LoRA settings) in the paper are specified in the released code and configs.

