# OpenReview forum: "Scaling-Aware Adapter for Structure-Grounded LLM Reasoning"
_ICML.cc/2026/Conference — ICML 2026 regular_

### Official Review · Reviewer_GtMW · 2026-03-12

**Soundness:** 3
**Presentation:** 3
**Significance:** 3
**Originality:** 3
**Overall Recommendation:** 4
**Confidence:** 1

**Summary:**

In this paper, the authors propose the framework Cuttlefish, an LLM that is capable of reasoning over all-atom modalities. It includes two components: Scaling-Aware Patching to address the issue of fixed-length query connectors and Geometry Grounding Adapter to provide the explicit geometric grounding to the LLM. Through extensive experiments, Cuttlefish achieves superior performance, indicating that the incorporation of the two components are crucial for structure-aware reasoning. Additionally, the authors curated a new dataset GEO-AT, offering geometry-grounded instruction corpus including different types of modalities.

**Compliance With Llm Reviewing Policy:**

Affirmed.

**Final Justification:**

I thank the authors for their rebuttal. All of my concerns have been addressed. I will keep my positive score.

**Key Questions For Authors:**

1. As mentioned before, the paper does not specify the backbone model used for Cuttlefish in Table 3. How does the framework perform when the backbone model is the same / of comparable model sizes?
2. The scaling-aware patching enables an adaptive number of tokens. How does this number compare to the baselines’ fixed number of tokens?
3. Does the geometric grounding adapter only rely on cross attention to observe the geometric structure? How do we know it actually contains geometric information?

**Limitations:**

Yes

**Strengths And Weaknesses:**

Soundness: The paper is generally technically sound. The authors supported their claims through the experiment results, which was performed on a wide range of tasks and model architectures. One weakness is that the authors don’t mention what backbone model they used for Cuttlefish in the later experiments (Tables 3 - 5). It appears that the baselines use a variety of model architecture and sizes, which can contribute to the performance difference as well. It would be better if all baselines and Cuttlefish use the same backbone for fair comparison.

Presentation: Overall, the paper is well structured and follows a clear narrative. However, much of the implementation details are put in the Appendix without a short summary in the main paper, which can be confusing for people unfamiliar with the field. For instance, the main paper didn’t give any details on the evaluation metrics used for the experiments, which can hinder one’s understanding of the results. A minor issue is that the authors include special formatting to highlight which parts connect back to the two core challenges. I think they can be better integrated into the text using full sentences to explicitly state the connection.

Significance: From the related works (Table 8 - 9), it appears that the authors are addressing two significant limitations of existing works: scaling and geometric structure awareness. In Table 3, their framework Cuttlefish achieves substantial improvement in a few datasets. The authors propose an interesting perspective through their framework and give empirical evidence that it is effective, which can be beneficial in advancing the field.

Originality: The authors introduced a novel solution to each of the challenges listed and provide theoretical justification of their choices. Specifically, the geometric grounding adapter introduces a new approach for LLMs in molecular applications in how to integrate geometric structure into sequence-only inputs. They also showed evidence that this approach reduced hallucinations most effectively compared to baselines.

---

> ### Author Rebuttal · Authors · 2026-03-30
>
> We thank the reviewer for recognizing the significance of addressing both scaling and geometric grounding limitations in prior structure-aware LLMs, and the originality of Cuttlefish’s adaptive patching and geometry-grounding design, as well as the value of GEO-AT and the reported hallucination reductions.
>
> ---
>
> **[W1] Backbone Clarification for Later Tables**
>
> Thank you for pointing out that the backbone choice is easy to miss in the later tables.
>
> `Default Backbone` The default Cuttlefish backbone in Tables after Table 2 is Llama-3.1-8B-Instruct.
>
> `Same-Backbone Ablation` We added a same-backbone comparison to isolate the connector effect.
>
> >**Table R1: Same-backbone comparison on GEO-AT.**
> >
> >https://anonymous.4open.science/r/Cuttlefish-455E/tables/GtMW_R1.md
>
> In Table R1, Cuttlefish shows ~10% gains on both molecular and protein tasks compared to the second-best baseline.
>
> `Revision` We will add Table R1 in Sec. 4.4, and explicit statements in the captions of Table 3-5.
>
> ---
>
> **[W2] Bringing Implementation and Metric Details Into the Main Paper**
>
> Thank you for this presentation suggestion. We revised the manuscript on the following three points:
>
> `1. Implementation Summary` We will move implementation details from Appendix Tables 10–11 into the main text, summarizing the EGNN encoder, fusion blocks, patching hyperparameters, and the two-stage training setup.
>
> `2. Metric Summary` We will add metric descriptions in Sec. 4, including BLEU / ROUGE / METEOR / BERTScore for text understanding; Exact, Levenshtein, validity, and fingerprint Tanimoto similarity for structured generation; Acc or MCC for classification; MAE for scalar prediction.
>
> `3. Challenge Linkage` We will revise them into direct explanatory sentences. For example, revise the end of Sec. 4.3 to "These cross-backbone gains support Challenge 2 by showing that explicit geometry grounding improves structure-conditioned answer quality."
>
> ---
>
> **[Q1] Comparable Backbone Sizes**
>
> We will make this explicit by stating that Cuttlefish use Llama-3.1-8B-Instruct as the default backbone and by adding the same-backbone comparison in **[W1]** (Table R1).
>
> ---
>
> **[Q2] How many tokens does adaptive patching use relative to fixed-token baselines?**
>
> Figure 6 (c,d) shows the qualitative pattern: Cuttlefish uses a substantially smaller budget than sequence-exposed baselines, and its token growth is much slower with entity size.
>
>  To make this quantitative, we added a token-usage table stratified by entity size.
>
> >**Table R2: Token-usage comparison by entity size.**
> >
> >https://anonymous.4open.science/r/Cuttlefish-455E/tables/GtMW_R2.md
>
> `Revision` We will place Table R2 in Appendix E.6.
>
> ---
>
> **[Q3] Why does the geometric grounding adapter contain geometric information?**
>
> `In One Sentence` Anchor selection pinpoint instruction-conditioned substructure positions, and the adapter performs geometry retrieval and refinement.
>
> `Anchor Selection.` The anchor-selection stage is to pinpoint instruction-relevant substructure positions.
>
> In Table R3, we directly test whether the selected anchors overlap with the queried substructure. Given prompts “Whether a molecule contains [a substructure]”, we check whether the highest-scored positions fall inside that substructure. The result shows that the gate achieves 93% top-3 accuracy in substructure positioning.
>
> >**Table R3: Functional-group grounding evaluation.**
> >
> >https://anonymous.4open.science/r/Cuttlefish-455E/tables/GtMW_R3.md
>
> `Geometry Adapter` performs cross-attention to the full entity embeddings, retrieving detailed local and global features conditioned on the selected substructures.
>
> `Revision` Added clarification to Sec. 3.3 stating that the adapter is a geometric retrieval-and-refinement stage.

---

> > ### Author Rebuttal · Reviewer_GtMW · 2026-04-02
> >
> > Thank you for the detailed responses. My concerns have been fully addressed.

---

> > > ### Author Response · Authors · 2026-04-07
> > >
> > > Dear Reviewer GtMW,
> > >
> > > We are glad the rebuttal fully addressed the concerns, and we sincerely thank you for the careful engagement. We also greatly appreciate your recognition of the paper’s main strengths, including the significance of addressing both scaling and geometric structure awareness, the technically sound empirical support across diverse tasks and model architectures, the substantial empirical improvements, and the evidence that explicit geometric grounding helps reduce hallucinations.
> > >
> > > To directly address the remaining points, we added the following experiments/revisions during rebuttal:
> > >
> > > 1. **Same-backbone comparison on GEO-AT:** **~10% gains** on both molecular and protein tasks over the second-best baseline;
> > > 2. **Implementation summary in the main paper:** 4 key components are now summarized in the main text– encoder, fusion blocks, patching, and two-stage training;
> > > 3. **Metric summary in Sec. 4:**  4 metric groups are now explicitly described;
> > > 4. **Token-usage comparison by entity size:** Cuttlefish uses a substantially smaller token budget with slower growth as entity size increases;
> > > 5. **Functional-group grounding evaluation:** **93% top-3 accuracy** in substructure positioning;
> > > 6. **Adapter mechanism clarification:** the adapter is now explicitly described as a **geometry retrieval-and-refinement** stage via cross-attention to full entity embeddings;
> > >
> > > Overall, these additions further support that the observed gains are not attributable to backbone choice alone, and that both adaptive token allocation and geometry-grounded retrieval are empirically supported and mechanistically justified. We thank the reviewer for the thoughtful feedback, which helped make the paper’s presentation and main claims clearer and stronger.
> > >
> > > Sincerely,
> > >
> > > The Authors

---

### Official Review · Reviewer_w8Lq · 2026-03-12

**Soundness:** 4
**Presentation:** 3
**Significance:** 3
**Originality:** 3
**Overall Recommendation:** 4
**Confidence:** 4

**Summary:**

This paper introduces Cuttlefish, a unified “all-atom” multimodal LLM intended to answer biomolecular questions by explicitly grounding in 3D geometry, not just sequence. The core design is a connector that avoids the fixed-length bottleneck of query-token approaches: it performs scaling-aware patching by selecting instruction-conditioned anchor atoms and forming variable-sized patches, producing a variable number of structural tokens. A geometry grounding adapter then uses cross-attention to retrieve fine-grained geometric information and inject it into the LLM via modality placeholders. Training is staged with an SE(3)-equivariant EGNN encoder plus alignment steps to integrate structural tokens. Empirically, the paper reports improvements over instruction-tuned LLM baselines and modality-specific models on heterogeneous all-atom instruction benchmarks, better scaling with structure size than fixed-query connectors, and stronger robustness on hallucination-oriented evaluations (low hallucination rates while maintaining a high answer rate as in table 7).

**Compliance With Llm Reviewing Policy:**

Affirmed.

**Key Questions For Authors:**

check weaknesses

**Limitations:**

yes

**Strengths And Weaknesses:**

Strengths:
1-The work address the under-representation of atoms geometric information, sequence-only inputs discard geometric information, and fixed-length “query token” connectors can bottleneck information as structure size/complexity grows. The proposed variable-budget structural tokenization + explicit geometry injection is a reasonable response to those failure modes and advances the multimodal area in computational biology.
2- Empirical evaluation appears to take in to account all aspects (general instruction-style benchmarks + scaling behavior + hallucination-focused tests
3- Ablations support that each component matters. The ablation (table 7) shows notable drops when removing instruction-guided gating, fusion blocks, or soft patch growth; and a large degradation for a Q-Former variant with the same max token length, aligning with the “fixed-length bottleneck” narrative.
4- Introducing an all-atom instruction dataset (GEO-AT) is potentially valuable for the community if released with clear splits and tooling.

Weaknesses / questions:
1- The claims hinge on both scaling-aware token allocation and the geometry grounding adapter. the claims would be stronger with very crisp ablations (if I didn't miss it in the appendix) showing (1) fixed budget vs variable budget under matched compute/tokens, (2) anchor/patching choices vs simpler pooling, (3) the adapter’s role vs direct projection, and (4) sensitivity to number of injected tokens.

2- The hallucination test is on 200 molecules and proteins, and the notion of “functional group hallucination” may not capture other failure modes (e.g., incorrect mechanistic explanations, wrong long-range interactions, unsupported structural claims). It’s a good start but may not establish broad faithfulness. can you maybe comment on this?

3- For proteins, structures are retrieved via UniProt→PDB mapping and otherwise generated with AlphaFold2. That introduces a confound: performance may depend on structure correctness/coverage, and errors in predicted structures could systematically bias results. can you clarify how missing/low-confidence structures are handled and how robust the method is to structural noise?

4- The approach depends on choices like mass threshold ρ, maximum patch count Kmax, and the mechanics of “soft patch growth.” The main paper shows component ablations but less about sensitivity to these knobs (which can matter a lot for scaling claims).

---

> ### Author Rebuttal · Authors · 2026-03-30
>
> We sincerely thank the reviewer for recognizing our well-motivated adaptive tokenization and geometry-grounding design, supported by comprehensive benchmarks, with GEO-AT as an additional community contribution.
>
> ---
>
> **[W1] Crisper Ablations for Dynamic Tokens**
>
> We thank the reviewer for requesting these ablations. We added them as follows:
>
> `1. Matched Budget` Table 6 already contains one matched-budget ablation through the Q-Former variant. Here, we add explicit comparison by filtering large entities to fully utilize the maximum budget, ensuring fairness. In Table R1, ours has an average 6.44% relative gain in matched token length from 256 to 2048.
>
> >**Table R1. Matched-budget comparison with fixed-length connectors.**
> >
> >https://anonymous.4open.science/r/Cuttlefish-455E/tables/w8Lq_R1.md
>
> `2. Simpler Pooling` We already have this ablation in Table 22, and agree it should be surfaced into Sec. 4. In Table 22, simple pooling has an average drop from 4.14% to 5.76%.
>
> >**Table 22. Ablation of pooling methods.**
> >
> >https://anonymous.4open.science/r/Cuttlefish-455E/tables/w8Lq_22.md
>
> `	 3. Adapter Contribution` We have the ablation through the “w/o fusion blocks” variant in Table 6. We will rename that row as "Direct Projection".  The average improvements are 22.78% for METEOR and 13.93% for BERTScore.
>
> >**Table 6 (subset). Ablation of the geometry-grounding versus direct projection**
> >
> >https://anonymous.4open.science/r/Cuttlefish-455E/tables/w8Lq_6.md
>
> `4. Token Length Sensitivity` We added the analysis of the amount of injected tokens (Table R2). The result shows the best max token length is around 2048.
>
> >**Table R2. Ablation on the sensitivity of the injected token amount.**
> >
> >https://anonymous.4open.science/r/Cuttlefish-455E/tables/w8Lq_R2.md
>
> `Revisions` We will add these four ablations together in Sec. 4.6.
>
> ---
>
> **[W2] Broader Hallucination Analysis**
>
> We thank the reviewer and add two additional failure modes: incorrect mechanistic explanations and unsupported long-range interaction claims.
>
> >**Table R3. Hallucination evaluation on three failure modes.**
> >
> >https://anonymous.4open.science/r/Cuttlefish-455E/tables/w8Lq_R3.md
>
> Table R3 shows that Cuttlefish has an average 5.5% lower hallucination rate than the strongest baseline.
>
> `Revision` We will add this to Table 7 in Sec. 4.7.
>
> ---
>
> **[W3] Robustness to Structural Noise**
>
> We thank the reviewer for raising the structure-quality confound.
>
> `Current Policy` When structures are unavailable, we generate them with AlphaFold2, and samples that fail generation are retained as sequence-only fallbacks (<0.5% of cases). We agree that robustness should be made more explicit.
>
> `1. Source Comparison (Table R4)` We added an analysis where the same protein is evaluated with both experimental and predicted structures. The average gap is only 1.2%.
>
> >**Table R4. Structure source analysis.**
> >
> >https://anonymous.4open.science/r/Cuttlefish-455E/tables/w8Lq_R4.md
>
> `2. Confidence Bins (Table R5)` We stratify AlphaFold2 cases by confidence, using standard pLDDT bins, to test whether gains degrade smoothly with decreasing confidence. Results show the performance is stable when confidence is >80%.
>
> >**Table R5. Prediction confidence analysis.**
> >
> >https://anonymous.4open.science/r/Cuttlefish-455E/tables/w8Lq_R5.md
>
> `3. Noise Robustness (Table R6)`  For explicit noise robustness, we add controlled coordinate perturbations (isotropic Gaussian coordinate jitter) to experimental structures. Results show the performance is stable when noise is <0.5A.
>
> >**Table R6. Structure noise analysis.**
> >
> >https://anonymous.4open.science/r/Cuttlefish-455E/tables/w8Lq_R6.md
>
> `4. W/O Structure (Table R7)` We compared the raw LLM, sequence-only tuning, structure-trained inference without structure, and full structure grounding. Results show that structure-trained LLM without structure inputs can still handle missing structures well (<5% drop).
>
> >**Table R7. Missing structure analysis**
> >
> >https://anonymous.4open.science/r/Cuttlefish-455E/tables/w8Lq_R7.md
>
> `Conclusion & Revision`  From these 4 analysis, Cuttlefish is robust to structure noise, although our main improvement comes from the structure inputs. We will add these to App. C.2.
>
> ---
>
> **[W4] Sensitivity to more hyperparameters**
>
> We thank the reviewer and add an ablation over the two main patching knobs in Table R8.
>
> >**Table R8. Sensitivity of two hyperparameters.**
> >
> >https://anonymous.4open.science/r/Cuttlefish-455E/tables/w8Lq_R8.md
>
> We also added a second analysis of soft patch growth and adapter-side knob.
>
> >**Table R9. Sensitivity analysis for soft patch growth and additional grounding knob.**
> >
> >https://anonymous.4open.science/r/Cuttlefish-455E/tables/w8Lq_R9.md
>
> `Revisions` We will add these ablations to Appendix E.5 to provide insights on the parameter settings.

---

> > ### Author Rebuttal · Reviewer_w8Lq · 2026-04-01
> >
> > I thank the authors for the clear response and modification.

---

> > > ### Author Response · Authors · 2026-04-07
> > >
> > > Dear Reviewer w8Lq,
> > >
> > > We are glad the rebuttal fully addressed the concerns, and we sincerely thank you for the careful engagement. We also greatly appreciate your recognition of the paper’s main strengths, including the well-motivated formulation around geometric grounding, the scaling-aware tokenization, the comprehensive evaluation, and the strong scaling and hallucination robustness.
> > >
> > > To directly address the requested points, we added the following experiments during rebuttal:
> > >
> > > 1. **Matched-budget comparison with fixed-length connectors:** **+6.44% average relative gain** under matched token budgets;
> > > 2. **Simpler pooling ablation:** **4.14%–5.76% average drop** with simple pooling;
> > > 3. **Geometry-grounding adapter vs. direct projection:** **+22.78% METEOR** and **+13.93% BERTScore**;
> > > 4. **Injected-token sensitivity:** best performance occurs around **2048 max tokens**;
> > > 5. **Broader hallucination evaluation:** **5.5% lower average hallucination rate** on two additional tasks;
> > > 6. **Experimental vs. predicted structures:** only **1.2% average gap**;
> > > 7. **Confidence-bin analysis for predicted structures:** performance remains stable for **pLDDT > 80**;
> > > 8. **Coordinate-noise robustness:** performance remains stable for noise **< 0.5 Å**;
> > > 9. **Missing-structure analysis:** structure-trained inference without structure causes **<5% drop**;
> > > 10. **Hyperparameter sensitivity:** results remain stable across the tested ranges of patching and grounding knobs;
> > >
> > > Overall, these additions further support that the gains are not only effective, but also well-isolated, robust, and stable under fair comparisons, broader faithfulness tests, structural uncertainty, and hyperparameter variation. We thank the reviewer for the constructive feedback, which helped make the paper’s empirical support and main claims substantially stronger.
> > >
> > > Sincerely,
> > >
> > > The Authors

---

### Official Review · Reviewer_daHq · 2026-03-13

**Soundness:** 3
**Presentation:** 3
**Significance:** 3
**Originality:** 3
**Overall Recommendation:** 4
**Confidence:** 3

**Summary:**

This paper notes two main challenges in LLM reasoning for molecule structures: Budget Scaling in All-Atom Modalities and  Structural Hallucination. To solve these challenges, the authors apply a scaling-aware patching and geometry grounding adapter. The authors provide detailed techniques and training processes. Experiments suggest strong performance of the proposed method.

**Compliance With Llm Reviewing Policy:**

Affirmed.

**Final Justification:**

I keep my positive score: 4

**Key Questions For Authors:**

I have some questions to discuss with the authors:

What is the reasoning in this paper mean, and is it different from the common 'reasoning'?

Is the anchor selection process aiming at keeping more sub-structure information? I guess that some tokens are important for keeping the information from sub-groups (or subgraphs), and this process can help. If what I guess is true, what does the geometric adapter act as? I'm curious about the mechanisms in this problem.

**Limitations:**

Yes

**Strengths And Weaknesses:**

Strengths:

++ The challenges and correlated solutions are clearly demonstrated in the introduction part. Following the authors' explanations, I can easily understand the motivations and efforts of them to solve these challenges.

++ After reading the method part, I get some insight on how to improve the capabilities of modern science LLMs. I guess that simply applying a modality encoder (like GIN, etc.) is not very enough. keeping the structural information may help a lot. I'm a researcher doing some work on science LLMs, and I get inspired from this work.

++ Visualization in this work is good, like Figure 2.   Additionally, and moreover, the extensive experiments in this paper are strong evidences for good performance of their proposed method, across numerous kinds of LLMs.

++ Code is provided in the submission. This supports the reproducibility of the work.

Weaknesses:

-- One of the major claims of the author falls in 'reasoning'. However, after I read the answers from proposed Cuttlefish, I discover it a simple question--> direct answer process, but not the common 'reasoning' process (CoT, steps, thinking, etc.). The strong capabilities of Cuttlefish are more likely from the prior given by GNN and structure-aware modules, but not 'reasoning'. Thus, some claims in this paper may be modified.

-- It is not very clearly explained why high & low scores from softmax + SE(3)-equivariant GNN can represent the importance of nodes. I can understand this intuitively, but are there any other proofs?

-- In Table 2, metrics measuring text similarity are used, like METEOR and BERT-S. I wonder why some metrics specially designed for molecule generation quality and so on are not used, like Exact match, tanimoto similarity and so on. (I notice that in Table 3 you apply some.) For molecules, it's important to know whether they really generate correct molecules according to the prompt, rather than generating similar sequences.

-- Since this is a method able to be scaled up, a minor weakness is that: larger version of LLMs will better help support the claims.

-- One major concern is the theoretical analysis part in Appendix A. We may discover that 'the modules have gradients' are proved, however, it is a common thing in deep learning (like MLPs have gradients). These proofs are not real guarantees for module performance. For the information bottleneck part, similar problems can be found. The proofs are showing a thing that is 'absolutely correct', but maybe not real mechanisms of the modules. The correlation of Appendix A and the main text is somewhat weak. I suggest that the authors either provide a better version, or just discard these things (for they do not give very positive support on the claims of the paper)

---

> ### Author Rebuttal · Authors · 2026-03-28
>
> We thank the reviewer for recognizing our well-motivated and empirically supported solution to a real limitation in structure-aware LLMs.
>
> ---
>
> **[W1] Clarify the Reasoning Claim**
>
> We thank and agree that the reasoning claim was underspecified.
>
> `Our Reasoning`  is standard CoT LLM reasoning with examples shown in Fig. 8-12. Our point is that the proposed method benefits reasoning LLMs more than non-reasoning backbones, as it provides fine-grained cues that better support reasoning, as proved by the following three experiments:
>
> `1. Gains Comparison on Backbone Types`
>
> > **Table R1: Performance gain of reasoning & non-reasoning models.**
> >
> > https://anonymous.4open.science/r/Cuttlefish-455E/tables/daHq_R1.md
>
> In Table R1, more average gains (26.8%-METEOR and 23.6%-BERT-S) on reasoning backbones.
>
> `2. Explicit Reasoning Benchmark`  ChemIQ and MolecularIQ were added to support our claim.
>
> > **Table R2: Results on ChemIQ and MolecularIQ.**
> >
> > https://anonymous.4open.science/r/Cuttlefish-455E/tables/daHq_R2.md
>
> In Table R2, the average gain on reasoning backbones is 19% higher than that of the non-reasoning ones.
>
> `3. Different Settings & Same Backbone` The backbone itself may influence the results; therefore, we use the same reasoning LLM with reasoning/non-reasoning prompt settings for a fair comparison.
>
> > **Table R3: Reasoning setting ablation under the same backbone.**
> >
> > https://anonymous.4open.science/r/Cuttlefish-455E/tables/daHq_R3.md
>
> In Table R3, reasoning settings double the gains with Cuttlefish on average.
>
> `Revisions.` Added Table R2 and R3 in Sec. 4, with R1 to App. D. We adjusted the phrasing, e.g, “reasoning” to “understanding and reasoning”.
>
> ---
>
> **[W2] Gate Reflects Instruction Relevance**
>
> We appreciate this clarification request.
>
> `Gate Design` The score is from the Instruction-Conditioned Gate, not softmax for GNN. The gate takes both the prompt and embeddings as inputs, and is tuned end-to-end. The logits indicate positions more relevant to the prompt.
>
> `Gate Ablation` Functional-group grounding test: given prompts “Whether a molecule contains [A Substructure]”, we check whether the highest-scored positions fall inside that substructure.
>
> > **Table R4: Functional-group grounding evaluation.**
> >
> > https://anonymous.4open.science/r/Cuttlefish-455E/tables/daHq_R4.md
>
> In Table R4, our gate can pinpoint the substructure positions even without softmax.
>
> `Revision` Revised Sec. 3.2 to clarify that the gate score represents **Instruction-Conditioned Relevance**, while softmax is the normalization.
>
> ---
>
> **[W3] Task-Appropriate Metrics**
>
> We thank and agree that the mentioned metrics are important.
>
> `Different Task Types` Table 2 is a description task that output is text describing the entity. Table 3 contains true generation, such as molecule design.
>
> `Full Metric Provided` We do have provided all the mentioned metrics in the Appendix Table 13-17. For spatial reasons, we displayed two of them in Table 2&3.
>
> > **Table R5: Full metrics for generation and understanding tasks.**
> >
> > https://anonymous.4open.science/r/Cuttlefish-455E/tables/daHq_13_18.md
>
> Among tasks in Table R5, our model achieves top-1 at least 5/7 of the metrics on each task.
>
> `Revision` Updated the text around Tables 2&3 to make the task distinction explicit, and added references to the full metrics tables.
>
> ---
>
> **[W4] Larger-Backbone Support for the Scaling Claim**
>
> We thank the reviewer for this suggestion.
>
> `Backbone Scaling Ablation` We added two larger LLMs from the same family: Qwen-2.5-14B and 32B.
>
> > **Table R6: Backbone-size ablation using the Qwen-2.5 family.**
> >
> > https://anonymous.4open.science/r/Cuttlefish-455E/tables/daHq_R5.md
>
> From Table R6, the gains scaled with larger backbones from 0.5B to 32B.
>
> `Revision` Added this ablation to Sec. 4.6.
>
> ---
>
> **[W5] Simplifying Appendix A**
>
> We thank the reviewer for this assessment, and we agree.
>
> `Original Motivation` In our earlier prototype, hard-threshold selection and unused gate logits left the gate without gradients. After redesigning, we kept the proof mainly to verify trainability.
>
> `Revision` Removed App. A.1&2, and retained minimal definitional in App. A.3.
>
> ---
>
> **[Q1] Meaning of Reasoning**
>
> Our "reasoning" is not a different definition. As clarified in **[W1]** along with three added experiments  (Tables R1–R3), Cuttlefish benefits reasoning LLMs more than non-reasoning ones due to the structure cues.
>
> ---
>
> **[Q2] Anchor Selection vs. Geometric Adapter**
>
> `Anchor Selection` Yes. The anchor selection is to pinpoint instruction-relevant substructures. In **[W2]**, Table R4, we directly test whether the anchors overlap with the queried substructure.
>
> `The Geometry Adapter` performs cross-attention to the entity embedding, retrieving detailed and global features conditioned on the selected substructures.
>
> `Revision` Added clarification to Sec. 3.3 stating that the adapter is not a second selector, but a retrieval-and-refinement stage.

---

> > ### Author Rebuttal · Reviewer_daHq · 2026-04-02
> >
> > Due to the strong empirical performance of the proposed method, I choose to keep my positive score: 4. However, I may not assess a higher score, for the weaknesses in theoretical analysis. All in all, I appreciate the efforts on incorporating new structural information.
> >
> > ----------------------------------------------------------------------------------------------------------------------------------------------------------------------
> >
> > Thank you for your response. I think that the new figures combined with theories are good. Additionally, maybe theory of information bottleneck is not very task-relevant, for it describes the trend. I'm OK with acceptance of this paper. By the way, how do you draw these beautiful figures? I want to use some tools you used in this paper, thank you!

---

> > > ### Author Response · Authors · 2026-04-06
> > >
> > > We thank the reviewer for noting that the previous theoretical analysis was insufficient to highlight our main novelties. We have now rewritten the proof to support our two core novelties directly.
> > >
> > > `What Is Changed` The new proof is now a mechanism-level analysis of (i) **Scaling-Aware Patching** as adaptive structural compression, and (ii) **Geometry Grounding Adapter** as ambiguity reduction through explicit geometric evidence. We summarize it as follows:
> > >
> > > **1. Scaling-Aware Patching as adaptive structural compression.**
> > > We now analyze the patching module as a relevance-constrained compression mechanism. For graph $g$, let the gate scores be sorted in descending order, and let $S_g$ be the selected anchor set whose cumulative mass first reaches threshold $\rho$. We prove:
> > >
> > > **Proposition 1 (minimal-cardinality relevance coverage).** $S_g$ is the smallest-cardinality set whose total relevance mass reaches at least $\rho$; moreover, among all sets of the same size, it captures the maximum total relevance mass. This is illustrated schematically in Figure P1.
> > >
> > > > **Figure P1:  Budget Selection**
> > > >
> > > > https://anonymous.4open.science/r/Cuttlefish-455E/tables/figs/P1_compression.jpg
> > >
> > > This gives a direct mechanism-level justification for why the token budget grows with instruction-relevant structural complexity, rather than being an arbitrary variable-length design.
> > >
> > > We then analyze soft patch growth and derive an explicit distortion guarantee. Under a local smoothness assumption, let $X_i$ denote the encoder embedding of node $i$, $\hat X_i$ its patch-based reconstruction after compression, $P_i$ its coordinate, $W_{i,a}$ the soft assignment weight from node $i$ to anchor $a$, and $L$ the local smoothness constant relating changes in encoder embeddings to changes in coordinates. We prove an instruction-weighted bound of the form,
> > >
> > > $$
> > > \sum_i p_i\||X_i-\hat X_i\|| \le
> > > L \sum_i p_i \sum_{a\in S_g} W_{i,a}\big(\||P_i-P_a\||_2 + r_a\big),
> > > $$
> > >
> > > where $S_g$ is the selected anchor set and $r_a$ is the average radius of patch $a$. Thus, compression error is controlled by two interpretable geometric quantities: node-to-anchor distance and intra-patch spread. This formalizes the mechanism by which adaptive patching preserves fine-grained geometry when selected anchors remain close to instruction-relevant regions and patches remain tight. A concrete patch-growth example is in Figure P2.
> > >
> > > > **Figure P2: Patch Growth Example**
> > > >
> > > > https://anonymous.4open.science/r/Cuttlefish-455E/tables/figs/P2_detailed_growth.jpg
> > >
> > > **2. Why fixed-length connectors incur a bottleneck.**
> > > To support our fixed-length bottleneck claim, we additionally prove a lower bound under a standard representative-set compression surrogate:
> > >
> > > **Proposition 2 (fixed-length lower bound).** If there are $M$ well-separated instruction-relevant regions, then any fixed set of $K<M$ representative tokens must incur nonzero covering distortion, with a lower bound proportional to $1-K/M$. This contrast between adaptive and fixed-budget compression is summarized in Figure P3.
> > >
> > > > **Figure P3: Bottleneck Comparison**
> > > >
> > > > https://anonymous.4open.science/r/Cuttlefish-455E/tables/figs/P3_compression.jpg
> > >
> > > This shows that under a natural representative-compression view, once relevant structural complexity exceeds a fixed token budget, nonzero information loss is unavoidable. This is the exact failure mode our adaptive connector is designed to avoid.
> > >
> > > **3. Geometry grounding as ambiguity reduction.**
> > > We also replaced the previous bottleneck discussion with an uncertainty argument. Let $S$ denote the sequence input, $T$ the geometry-grounded tokens, $Z$ the instruction, and $Y$ the target output. Here, $R^\*\_{\mathrm{seq}}$ and $R^\*\_{\mathrm{geom}}$ denote the Bayes-optimal expected log-loss risks for the sequence-only and geometry-grounded settings, respectively. We use $H$ for conditional entropy and $I$ for conditional mutual information. Under log loss, the Bayes-optimal risks satisfy,
> > > $$
> > > R^\*\_{\mathrm{seq}} = H(Y\mid S,Z), \qquad
> > > R^\*\_{\mathrm{geom}} = H(Y\mid S,T,Z),
> > > $$
> > >
> > > and therefore
> > >
> > > $$
> > > R^\*\_{\mathrm{seq}}-R^\*\_{\mathrm{geom}} = I(Y;T\mid S,Z)\ge 0.
> > > $$
> > >
> > > A visual summary of this point is given in Figure P4.
> > >
> > > > **Figure P4: Grounding to Reduce Ambiguity**
> > > >
> > > > https://anonymous.4open.science/r/Cuttlefish-455E/tables/figs/P4_geo_grounding.jpg
> > >
> > > Hence, geometry grounding helps precisely when the injected geometric tokens retain answer-relevant information not recoverable from sequence alone. This gives a direct uncertainty-level explanation for why explicit geometry grounding can reduce structural hallucination.
> > >
> > > `Conclusion` These mechanism-level guarantees support the paper’s two design claims more directly than the original proof did. We believe the proof is now substantially better aligned with the novelties: Propositions 1–2 support the scaling results, and the Bayes-risk result supports the hallucination-reduction claim.
> > >
> > > ---
> > > `Re Reviewer Follow-up` Microsoft PPT.

---

### Decision · Program_Chairs · 2026-04-30

**Decision:**

Accept (regular)

**Comment:**

All the reviewers appreciate the performance gains achieved by Cuttlefish in enhancing the capabilities of modern scientific LLMs, which is an increasingly important direction. The paper is generally well written, and the figures/tables are clear and easy to follow. The inclusion of ablation studies further helps demonstrate the effectiveness of the proposed components. Overall, I believe this is a solid contribution to the machine learning community.